# Simple and Effective Masked Diffusion Language Models

**Subham Sekhar Sahoo**
Cornell Tech, NYC, USA.
`ssahoo@cs.cornell.edu`

**Marianne Arriola**
Cornell Tech, NYC, USA.
`ma2238@cornell.edu`

**Yair Schiff**
Cornell Tech, NYC, USA.
`yzs2@cornell.edu`

**Aaron Gokaslan**
Cornell Tech, NYC, USA.
`akg87@cs.cornell.edu`

**Edgar Marroquin**
Cornell Tech, NYC, USA.
`emm392@cornell.edu`

**Justin T Chiu**
Cornell Tech, NYC, USA.
`jtc257@cornell.edu`

**Alexander Rush**
Cornell Tech, NYC, USA.
`ar459@cornell.edu`

**Volodymyr Kuleshov**
Cornell Tech, NYC, USA.
`kuleshov@cornell.edu`

## Abstract

While diffusion models excel at generating high-quality images, prior work reports a significant performance gap between diffusion and autoregressive (AR) methods in language modeling. In this work, we show that simple masked discrete diffusion is more performant than previously thought. We apply an effective training recipe that improves the performance of masked diffusion models and derive a simplified, Rao-Blackwellized objective that results in additional improvements. Our objective has a simple form—it is a mixture of classical masked language modeling losses— and can be used to train encoder-only language models that admit efficient samplers, including ones that can generate arbitrary lengths of text semi-autoregressively like a traditional language model. On language modeling benchmarks, a range of masked diffusion models trained with modern engineering practices achieves a new state-of-the-art among diffusion models, and approaches AR perplexity. We provide the code[1], along with a blog post and video tutorial[2] on the project page:
https://s-sahoo.com/mdlm

## 1 Introduction

Diffusion models excel at producing realistic, high-quality images and have received significant attention as potential tools for generating discrete data, such as text [1, 31, 33], biological sequences [2, 47], and graphs [60, 63]. Unlike autoregressive (AR) approaches, diffusion-based methods are not constrained to generate data sequentially, and therefore have the potential to improve long-term planning, controllable generation, and sampling speed. However, discrete diffusion methods exhibit a performance gap relative to AR models [1, 23, 26, 33], especially in language modeling. The standard measure of language modeling performance is log-likelihood: when controlling for parameter count, prior work reports a sizable log-likelihood gap between AR and diffusion models.

In this work, we show that simple masked diffusion language modeling (MDLM) combined with effective training recipes is more performant than previously thought [1, 26]. We develop a well-engineered MDLM implementation that significantly improves discrete diffusion log-likelihood; we

---

[1]code: https://github.com/kuleshov-group/mdlm
[2]tutorial: http://youtu.be/WjAUX23vgfg

38th Conference on Neural Information Processing Systems (NeurIPS 2024).

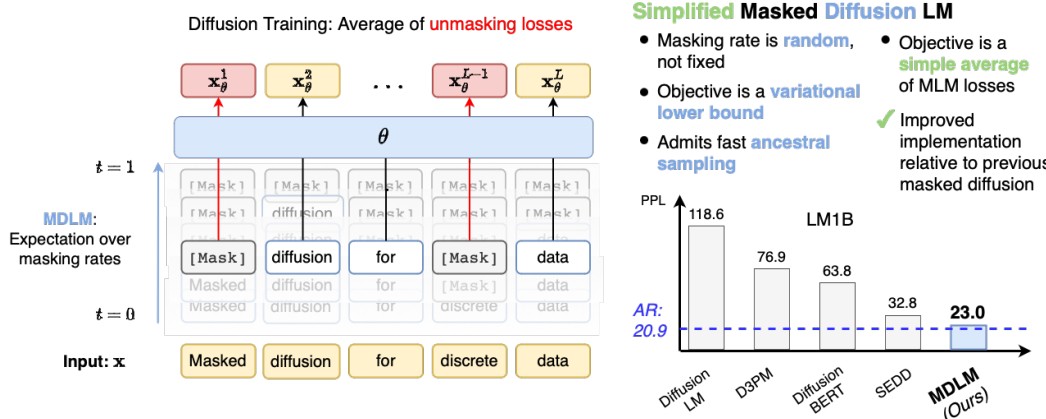

Figure 1: *(Left)* Our proposed masked diffusion language model (MDLM) is trained using a weighted average of masked cross entropy losses. *(Top Right)* In comparison to masked language models (MLM), MDLM's objective correspond to a principled variational lower bound, and supports generation via ancestral sampling. *(Bottom Right)* Perplexity (PPL) on One Billion Words (LM1B) benchmark.

further improve likelihood using a simple substitution-based parameterization of the reverse diffusion process that enables deriving a Rao-Blackwellized continuous-time variational lower bound (ELBO) with improved tightness [49]. Interestingly, our objective has a simple form: it is a weighted average of masked language modeling (MLM) losses [15], and can be used to endow BERT-style, encoder-only models with principled generation capabilities. We complement this framework with efficient samplers—including ones that can generate semi-autoregressively like a typical language model.

Our masked diffusion models achieve a new state-of-the-art among diffusion models on language modeling benchmarks and approach the perplexity of AR models within 15-25%. Surprisingly, simple engineering choices significantly improve performance in both our models and simple baselines that were previously thought to perform poorly. Our framework also extends to non-language domains, including biological sequence modeling. We pre-train DNA sequence models and observe similar or higher downstream performance compared to classical BERT-style training, while also introducing generative capabilities that classical masked DNA language models lack.

**Contributions** We describe (1) a simple masked diffusion language modeling (MDLM) framework with a well-engineered implementation that outperforms all existing diffusion models across language modeling benchmarks (LM1B [8], OWT [18], DNA [12]), and that significantly improves the performance of existing baselines [1, 26]. Our MDLM framework implements (2a) a substitution-based parameterization (SUBS) of the reverse unmasking diffusion process; SUBS allows us to derive (2b) a simple, continuous-time, Rao-Blackwellized objective that improves tightness and variance of the ELBO, further increasing performance. We complement MDLM with (3) fast samplers that support semi-autoregressive (SAR) generation and outperform previous SAR models.

## 2 Background

### 2.1 Diffusion Models

Diffusion models are trained to iteratively undo a forward corruption process $q$ that takes clean data $\mathbf{x}$ drawn from the data distribution $q(\mathbf{x})$ and defines latent variables $\mathbf{z}_t$ for $t \in [0,1]$ that represent progressively noisy versions of $\mathbf{x}$ [27, 54, 56, 66, 48, 19]. The standard forward process for continuous $\mathbf{x}$ is

$$\mathbf{z}_t = \sqrt{\alpha_t}\mathbf{x} + \sqrt{1-\alpha_t}\boldsymbol{\epsilon} \tag{1}$$

where $\boldsymbol{\epsilon} \sim \mathcal{N}(\mathbf{0},\mathbf{I})$ and $(\alpha_t)_{t \in [0,1]}$ is a noise schedule, monotonically decreasing in $t$. The parameterized reverse diffusion model $p_\theta$ over $\mathbf{x}$ and $\mathbf{z}_t$ is trained to maximize a variational lower bound on log-likelihood (ELBO). Given a number of discretization steps $T$, defining $s(i) = (i-1)/T$ and $t(i) = i/T$,

and using $D_{\mathrm{KL}}[\cdot]$ to denote the Kullback–Leibler divergence, the Negative ELBO (NELBO) equals [54]:

$$\mathbb{E}_q\left[\underbrace{-\log p_\theta(\mathbf{x}|\mathbf{z}_{t(0)})}_{\mathcal{L}_{\mathrm{recons}}}+\underbrace{\sum_{i=1}^{T}D_{\mathrm{KL}}[q(\mathbf{z}_{s(i)}|\mathbf{z}_{t(i)},\mathbf{x})\|p_\theta(\mathbf{z}_{s(i)}|\mathbf{z}_{t(i)})]}_{\mathcal{L}_{\mathrm{diffusion}}}\right]+\underbrace{D_{\mathrm{KL}}[q(\mathbf{z}_{t(T)}|\mathbf{x})\|p_\theta(\mathbf{z}_{t(T)})]}_{\mathcal{L}_{\mathrm{prior}}} \quad (2)$$

For brevity, we drop $i$ from $t(i)$ and $s(i)$ below; in general, $s$ will denote the time step before $t$.

## 2.2 Discrete Diffusion Models

Applications of diffusion modeling to discrete data can be broken into two broad categories. First are works that embed discrete structures in continuous space and then perform the Gaussian diffusion defined above on these continuous representations [9, 16, 23, 24, 30, 34, 57]. More related to our method are works that define a diffusion process directly on discrete structures. D3PM [1] introduces a framework with a Markov forward process $q(\mathbf{z}_t|\mathbf{z}_{t-1})=\mathrm{Cat}(\mathbf{z}_t;Q_t\mathbf{z}_{t-1})$ defined by the multiplication of matrices $Q_t$ over $T$ discrete time steps. This process induces marginals

$$q(\mathbf{z}_t|\mathbf{x})=\mathrm{Cat}(\mathbf{z}_t;\bar{Q}_t\mathbf{x})=\mathrm{Cat}(\mathbf{z}_t;Q_t\cdot Q_{t-1}\cdots Q_1\mathbf{x}) \quad (3)$$

that represent the discrete-state form of (1). Extending this formalism to continuous time (as in (1)) relies on continuous time Markov chain (CTMC) theory [5]. The CTMC framework in turns leads to generalizations of the score matching perspective on diffusion modeling [55] to discrete data [33, 59]. Notably, SEDD [33] connects score-based approaches with ELBO maximization, enabling performant likelihood-based training of score-based models.

# 3 Simple Masked Diffusion Models

While previous work on discrete diffusion supports general forward processes (e.g., general $Q_t$ in D3PM), absorbing state (i.e., masking) diffusion consistently achieves the best performance [1, 33]. In this work, instead of supporting general noise processes, we focus on masking and derive tight Rao-Blackwellized objectives that outperform general approaches and do not require CTMC theory. In this section, we first define the diffusion process for a categorical random variable. Later in Sec. 3.5, we extend this process to sequences containing multiple such categorical variables. We denote our overall approach as Masked Diffusion Language Models (MDLM).

**Notation.** We denote scalar discrete random variables with $K$ categories as 'one-hot' column vectors and define $\mathcal{V}\in\{\mathbf{x}\in\{0,1\}^K:\sum_{i=1}^{K}\mathbf{x}_i=1\}$ as the set of all such vectors. Define $\mathrm{Cat}(\cdot;\boldsymbol{\pi})$ as the categorical distribution over $K$ classes with probabilities given by $\boldsymbol{\pi}\in\Delta^K$, where $\Delta^K$ denotes the $K$-simplex. We also assume that the $K$-th category corresponds to a special [MASK] token and let $\mathbf{m}\in\mathcal{V}$ be the one-hot vector for this mask, i.e., $\mathbf{m}_K=1$. Additionally, let $\mathbf{1}=\{1\}^K$ and $\langle\mathbf{a},\mathbf{b}\rangle$ and $\mathbf{a}\odot\mathbf{b}$ respectively denote the dot and Hadamard products between two vectors $\mathbf{a}$ and $\mathbf{b}$.

## 3.1 Interpolating Discrete Diffusion

We restrict our attention to forward processes $q$ that interpolate between clean data $\mathbf{x}\in\mathcal{V}$ and a target distribution $\mathrm{Cat}(.;\boldsymbol{\pi})$, forming a direct extension of Gaussian diffusion in (1). Let $q$ define a sequence of increasingly noisy latent variables $\mathbf{z}_t\in\mathcal{V}$, where the time step $t$ runs from $t=0$ (least noisy) to $t=1$ (most noisy). The marginal of $\mathbf{z}_t$ conditioned on $\mathbf{x}$ at time $t$ is

$$q(\mathbf{z}_t|\mathbf{x})=\mathrm{Cat}(\mathbf{z}_t;\alpha_t\mathbf{x}+(1-\alpha_t)\boldsymbol{\pi}), \quad (4)$$

where $\alpha_t\in[0,1]$ is a strictly decreasing function in $t$, with $\alpha_0\approx 1$ and $\alpha_1\approx 0$; see Suppl. E.1 for details. This implies transition probabilities $q(\mathbf{z}_t|\mathbf{z}_s)=\mathrm{Cat}(\mathbf{z}_t;\alpha_{t|s}\mathbf{z}_s+(1-\alpha_{t|s})\boldsymbol{\pi})$, where $\alpha_{t|s}=\alpha_t/\alpha_s$. This indicates that during each diffusion step from $s\to t$, a fraction $(1-\alpha_{t|s})$ of the probability mass is transferred to the prior distribution $\boldsymbol{\pi}$. The reverse posterior is given as (see Suppl. 16 for details):

$$q(\mathbf{z}_s|\mathbf{z}_t,\mathbf{x})=\mathrm{Cat}\left(\mathbf{z}_s;\frac{[\alpha_{t|s}\mathbf{z}_t+(1-\alpha_{t|s})\mathbf{1}\boldsymbol{\pi}^\top\mathbf{z}_t]\odot[\alpha_s\mathbf{x}+(1-\alpha_s)\boldsymbol{\pi}]}{\alpha_t\mathbf{z}_t^\top\mathbf{x}+(1-\alpha_t)\mathbf{z}_t^\top\boldsymbol{\pi}}\right). \quad (5)$$

While (4) and (5) represent a special case of the more general diffusion processes proposed in D3PM [1], we show below that they yield a simplified variational lower bound objective and admit straightforward continuous time extensions.

## 3.2 Masked Diffusion

Next, we focus on masking processes and derive a simple Rao-Blackwellized objective for this choice of $q$. This objective incurs lower variance during training and improves tightness.

### 3.2.1 Forward Masking Process

In masked (i.e., absorbing state) diffusion, we set $\boldsymbol{\pi} = \mathbf{m}$. At each noising step, $t$, the input $\mathbf{x}$ transitions to a 'masked' state $\mathbf{m}$ with some probability. If an input transitions to $\mathbf{m}$ at any time $t'$, it will remain in this state for all $t > t' : q(\mathbf{z}_t \,|\, \mathbf{z}_{t'} = \mathbf{m}) = \mathrm{Cat}(\mathbf{z}_t; \mathbf{m})$. At time $T$, all inputs are masked with probability 1.

The marginal of the forward process (4) is given by $q(\mathbf{z}_t|\mathbf{x}) = \mathrm{Cat}(\mathbf{z}_t; \alpha_t \mathbf{x} + (1 - \alpha_t)\mathbf{m})$. Using properties of the masking process, the posterior $q(\mathbf{z}_s|\mathbf{z}_t,\mathbf{x})$ simplifies (5); see Suppl. A.2:

$$q(\mathbf{z}_s|\mathbf{z}_t,\mathbf{x}) = \begin{cases} \mathrm{Cat}(\mathbf{z}_s; \mathbf{z}_t) & \mathbf{z}_t \neq \mathbf{m}, \\ \mathrm{Cat}\left(\mathbf{z}_s; \frac{(1-\alpha_s)\mathbf{m} + (\alpha_s - \alpha_t)\mathbf{x}}{1 - \alpha_t}\right) & \mathbf{z}_t = \mathbf{m}. \end{cases} \tag{6}$$

### 3.2.2 Reverse Unmasking Process

The reverse process inverts the noise process defined by $q$. We consider both a finite number of steps $T$, as well as a continuous time model corresponding to $T \to \infty$. We begin with the discrete-time case for which the generative model is expressed as $p_\theta(\mathbf{x}) = \int_{\mathbf{z}} p_\theta(\mathbf{z}_1) p_\theta(\mathbf{x}|\mathbf{z}_0) \prod_{i=1}^{T} p_\theta(\mathbf{z}_s|\mathbf{z}_t) \mathrm{d}\mathbf{z}_{0:1}$.

The optimal form for $p_\theta(\mathbf{z}_s|\mathbf{z}_t)$ matches the true posterior in (6): this follows immediately from the definition of the diffusion objective in (2), which is a sum of terms of the form $\mathrm{D}_{\mathrm{KL}}(q(\mathbf{z}_s|\mathbf{z}_t,\mathbf{x}) \| p_\theta(\mathbf{z}_s|\mathbf{z}_t))$. However, (6) is conditioned on $\mathbf{x}$, which we do not know. Therefore, we introduce a model $\mathbf{x}_\theta(\mathbf{z}_t, t) : \mathcal{V} \times [0,1] \to \Delta^K$ that approximates $\mathbf{x}$ with a neural network. We can also omit explicit dependence of $\mathbf{x}_\theta$ on time $t$, which simplifies sampling, yielding a 2x inference speed-up (see Suppl. E.2).

### 3.2.3 SUBS Parameterization

The specific parameterization for $p_\theta(\mathbf{z}_s|\mathbf{z}_t)$ that we use is

$$p_\theta(\mathbf{z}_s|\mathbf{z}_t) = q(\mathbf{z}_s|\mathbf{z}_t, \mathbf{x} = \mathbf{x}_\theta(\mathbf{z}_t, t)) = \begin{cases} \mathrm{Cat}(\mathbf{z}_s; \mathbf{z}_t), & \mathbf{z}_t \neq \mathbf{m}, \\ \mathrm{Cat}\left(\mathbf{z}_s; \frac{(1-\alpha_s)\mathbf{m} + (\alpha_s - \alpha_t)\mathbf{x}_\theta(\mathbf{z}_t, t)}{1 - \alpha_t}\right). & \mathbf{z}_t = \mathbf{m}, \end{cases} \tag{7}$$

Furthermore, we induce 2 key properties of the absorbing state diffusion process into our denoising model, $\mathbf{x}_\theta(\mathbf{z}_t, t)$: an unmasked token remains unchanged during reverse diffusion, and the clean input is never masked. We implement these as substitutions to the output of $\mathbf{x}_\theta(\mathbf{z}_t, t)$, hence we call our parameterization SUBS.

**Zero Masking Probabilities**    First, notice that by definition, $\langle \mathbf{x}, \mathbf{m} \rangle = 0$. For this reason, we design the denoising network such that $\langle \mathbf{x}_\theta(\mathbf{z}_t, t), \mathbf{m} \rangle = 0$, i.e., we substitute the logit index corresponding to the [MASK] token with $-\infty$.

**Carry-Over Unmasking**    Second, if $\mathbf{z}_t$ is unmasked, then we desire $\mathbf{x}_\theta(\mathbf{z}_t, t) = \mathbf{z}_t$, i.e., unmasked latents are 'carried over'. We accomplish this by substituting the output of our network to simply copy unmasked inputs.

In Suppl. B.1, we show that "Zero Masking Probabilities" property simplifies the D3PM's NELBO (39) to (41), and "Carry-Over Unmasking" futher simplifies (41) to (43) whose continuous time equivalent is the simplified NELBO (10). Table 8 shows that each simplification leads to an improved likelihood.

## 3.3 Rao-Blackwellized Likelihood Bounds

Recall from (2) that the diffusion traning objective has the form $\mathcal{L}_{\mathrm{recons}} + \mathcal{L}_{\mathrm{diffusion}} + \mathcal{L}_{\mathrm{prior}}$. For the simplified reverse process in (7), the discrete-time diffusion loss for finite $T$ simplifies to (Suppl. B.1.3):

$$\mathcal{L}_{\mathrm{diffusion}} = \sum_{i=1}^{T} \mathbb{E}_q[\mathrm{D}_{\mathrm{KL}}(q(\mathbf{z}_{s(i)}|\mathbf{z}_{t(i)}, \mathbf{x}) \| p_\theta(\mathbf{z}_{s(i)}|\mathbf{z}_{t(i)}))] = \sum_{i=1}^{T} \mathbb{E}_q\left[\frac{\alpha_{t(i)} - \alpha_{s(i)}}{1 - \alpha_{t(i)}} \log\langle \mathbf{x}_\theta(\mathbf{z}_{t(i)}), \mathbf{x} \rangle\right]$$
$$\tag{8}$$

Note that this objective is simpler and more well-behaved than the expression one would obtain for $D_{KL}(q(\mathbf{z}_s|\mathbf{z}_t,\mathbf{x})\|p_\theta(\mathbf{z}_s|\mathbf{z}_t))$ under the parameterization induced by using $p_\theta(\mathbf{z}_s|\mathbf{z}_t) = q(\mathbf{z}_s|\mathbf{z}_t,\mathbf{x} = \mathbf{x}_\theta(\mathbf{z}_t,t))$ from (5), which is similar to what is used by D3PM [1] (see Suppl. A.2.4):

$$\left[\frac{\alpha_s - \alpha_t}{1-\alpha_t}\log\frac{\alpha_t\langle\mathbf{x}_\theta(\mathbf{z}_t,t),\mathbf{m}\rangle + (1-\alpha_t)}{(1-\alpha_t)\langle\mathbf{x}_\theta(\mathbf{z}_t,t),\mathbf{x}\rangle} + \frac{1-\alpha_s}{1-\alpha_t}\log\frac{(1-\alpha_s)(\alpha_t\langle\mathbf{x}_\theta(\mathbf{z}_t,t),\mathbf{m}\rangle + (1-\alpha_t))}{(1-\alpha_t)(\alpha_s\langle\mathbf{x}_\theta(\mathbf{z}_t,t),\mathbf{m}\rangle + (1-\alpha_s))}\right]\langle\mathbf{z}_t,\mathbf{m}\rangle \quad (9)$$

We refer to the process of obtaining (8) in lieu of (9) as a form of Rao-Blackwellization. Specifically, we analytically compute expectations such as $\langle\mathbf{x}_\theta(\mathbf{z}_t,t),\mathbf{m}\rangle = 0$ in order to simplify objective (9) to obtain (8). Without analytical simplifications, a model must learn $\theta$ such that $\langle\mathbf{x}_\theta(\mathbf{z}_t,t),\mathbf{m}\rangle = 0$ holds. Unlike in regular Rao-Blackwellization, simplifications are possible because of modeling choices for $\mathbf{x}_\theta(\mathbf{z}_t,t)$ (zero masking probabilities and carry-over unmasking). In that sense, our approach has similarities to graphical modeling, where incorporating conditional independencies into $p_\theta$ sets certain log-likelihood terms to zero. However, our approach also empirically helps reduce variance, hence we refer to it as Rao-Blackwellization, somewhat abusing the usual terminology.

### 3.4 Continuous-Time Likelihood Bounds

Previous works have shown empirically and mathematically that increasing the number of steps $T$ yields a tighter approximation to the ELBO [29]. Following a similar argument, we form an continuous extension of (8) by taking $T \to \infty$ (see Suppl. B.2), which yields the following NELBO, $\mathcal{L}_{\text{NELBO}}^\infty$:

$$\mathcal{L}_{\text{NELBO}}^\infty = \mathbb{E}_q \int_{t=0}^{t=1} \frac{\alpha_t'}{1-\alpha_t}\log\langle\mathbf{x}_\theta(\mathbf{z}_t,t),\mathbf{x}\rangle \mathrm{d}t \quad (10)$$

**Invariance to the noise schedule** The function $\alpha_t$ is invertible due to the monotonicity assumption in Sec. 3.1, and so we can perform the following change of variables in (10): $\gamma \equiv \log(1-\alpha_t)$. Thus, the diffusion loss can be equivalently expressed as $\mathcal{L}_{\text{NELBO}}^\infty = -\mathbb{E}_q\int_{\gamma=-\infty}^{\gamma=0}\log\langle\mathbf{x}_\theta(\mathbf{z}_\gamma,\gamma),\mathbf{x}\rangle\mathrm{d}\gamma$; see Suppl. E.1.1 for details. This new formulation demonstrates that the diffusion loss is invariant to the functional form of $\alpha_t$, which we verify empirically in Suppl. E.1.

### 3.5 Masked Diffusion Language Models

Next, we apply masked diffusion to language modeling over sequences $\mathbf{x}^{1:L}$ of $L$ tokens, with $\mathbf{x}^\ell$ denoting the $\ell$-th token. We make the assumption that the forward noising process is applied independently across a sequence and that, conditioned on a sequence of latents $\mathbf{z}_t^{1:L}$, the denoising process factorizes independently across tokens, i.e., $p_\theta(\mathbf{z}_s^{1:L} \mid \mathbf{z}_t^{1:L}) = \prod_{\ell=1}^L p_\theta(\mathbf{z}_s^\ell \mid \mathbf{z}_t^{1:L})$. Thus, we use a single model to compute $\mathbf{x}_\theta^\ell(\mathbf{z}_t^{1:L},t)$ for each $\ell$ from a masked sequence $\mathbf{z}_t$, optimizing:

$$\mathcal{L}_{\text{NELBO}}^\infty = \mathbb{E}_q \int_{t=0}^{t=1} \frac{\alpha_t'}{1-\alpha_t}\sum_\ell\log\langle\mathbf{x}_\theta^\ell(\mathbf{z}_t^{1:L},t),\mathbf{x}^\ell\rangle\mathrm{d}t \quad (11)$$

Interestingly, our objective has a simple form: it is the weighted average of masked language modeling (MLM) losses [15]. Thus our work establishes a connection between generative diffusion models and encoder-only BERT models. Our objective enables principled selection of a (randomized) masking rate, and also endows BERT-style models with principled generation capabilities; see Sec. 6. The full training algorithm is provided in Suppl. B.3.

**Note:** Although (11) imposes a loss on all tokens, unmasked tokens don't contribute to the loss, as they are copied over by the denoising network due to "carry-over unmasking" (Sec. 3.2.3), effectively reducing $\log\langle\mathbf{x}_\theta^\ell(\mathbf{z}_t^{1:L},t),\mathbf{x}^\ell\rangle$ to zero.

#### 3.5.1 Training Considerations for Masked Diffusion

One of the key contributions of our work is a well-engineered implementation of masked diffusion models. Our experiments demonstrate that these improvements greatly boost performance even for methods previously thought to perform poorly, e.g., Austin et al. [1]. Below we briefly summarize these implementation details. First, we find that tokenization is critical to performance. Small vocabularies, such as the 8k vocabulary in Austin et al. [1], result in longer-range dependencies that decrease the performance of both diffusion and AR models. Additionally, by focusing on masked diffusion, we

are able to provide a numerically stable implementation of the objective function. Namely, since previous formulations of discrete diffusion were constructed to accommodate a wide range of limiting distributions [1], the objective was implemented by materializing the full transition matrices $\bar{Q}_t$ and posterior probabilities. In contrast, we evaluate $D_{\mathrm{KL}}[q(\mathbf{z}_s \,|\, \mathbf{z}_t, \mathbf{x}) \,||\, p_\theta(\mathbf{z}_s \,|\, \mathbf{z}_t)]$ by examining only the masked token indices rather than comparing the full true and approximate posterior distributions.

Furthermore, we modernize the architecture for the denoising network relative to D3PM [1]. In lieu of the T5 architecture used in D3PM, we use the diffusion transformer (DiT) introduced in Peebles & Xie [42], which integrates time step conditioning into a standard encoder-only transformer [62] and uses rotary positional embeddings [58]. In addition, we implement a low-discrepancy sampler that reduces the variance of the ELBO, similar to Kingma et al. [29] and draws correlated samples $t_i$ rather than performing i.i.d. sampling.

## 4 Inference and Sampling in Masked Diffusion Language Models

### 4.1 Efficient Ancestral Sampling

To generate a sequence of length $L$, the reverse diffusion process starts with the sequence $\mathbf{z}_{t=1}^{1:L}$ where $\mathbf{z}_{t=1}^{\ell} = \mathbf{m}$, for all $\ell \in \{1, ..., L\}$. Then the subsequent latents, $\mathbf{z}_t^{1:L}$ are generated by discretizing the reverse diffusion process with some finite $T$. Given $\mathbf{z}_t^{1:L}$, we construct $\mathbf{z}_s^{1:L}$ by sampling each token $\mathbf{z}_s^{\ell}$ independently from the distribution $p_\theta(\mathbf{z}_s^{\ell} | \mathbf{z}_t^{1:L})$ given in (7).

Note that in the reverse process, unmasked tokens remain unchanged. Thus, if no new tokens in $\mathbf{z}_s^{1:L}$ become unmasked (which can occur often in early denoising stages for large $T$), then $\mathbf{z}_s^{1:L} = \mathbf{z}_t^{1:L}$. Additionally if the denoising model, $\mathbf{x}_\theta(\mathbf{z}_t^{1:L})$ is not conditioned on time, then we can simply draw a new sample from $p_\theta(\mathbf{z}_{s-1/T}^{1:L} | \mathbf{z}_s^{1:L})$ using the previously computed and cached value $\mathbf{x}_\theta(\mathbf{z}_t^{1:L})$. This means we have effectively "skipped" over the time step $s$, saving a function call to the denoising network. Note that SEDD [33] does not support this caching because the denoising network models time-dependent rates, which requires conditioning on time.

### 4.2 Semi-Autoregressive Masked Diffusion Language Models

Our method also admits an effective semi-autoregressive (SAR) decoding method that allows the model to generate sequences of arbitrary length [24, 52, 53]. Let $\tilde{\mathbf{x}}^{1:L}$ represent the output from sampling a sequence of $L$ tokens using the reverse diffusion process described above. To generate additional $L' < L$ tokens, we propose a generation algorithm in which the latter $L - L'$ tokens $\tilde{\mathbf{x}}^{L':L}$ are used as a prefix for an additional round of generation. Given the carry-over unmasking described in Sec. 3.2.3, these prefix tokens will simply be copied over at each decoding step. The remaining tokens are generated as above with $\mathbf{z}_s^{\ell} \sim p_\theta(\mathbf{z}_s^{\ell} | \mathbf{z}_t^{L:L+L'})$ for all $\ell \in \{L+1, ..., L+L'\}$, with $\mathbf{z}_{t=1}^{L-L':L}$ initialized to $\tilde{\mathbf{x}}^{L-L':L}$ as opposed to being initialized as masked tokens $\mathbf{m}$. At the end of this process, we have produced $L + L'$ tokens $\mathrm{concat}[\tilde{\mathbf{x}}^{1:L}, \tilde{\mathbf{x}}^{L+1:L+L'}]$, where $\mathrm{concat}[\cdot]$ denotes concatenation along the sequence length dimension. This process can repeat indefinitely, with the prefix shifted for every new round of generation.

## 5 Experiments

### 5.1 Masked Diffusion Language Models

**Experimental Setup** We evaluate MDLM as a generative model of language and as a representation model via fine-tuning on downstream tasks.

For language modeling likelihood evaluation, we conduct experiments on two datasets: The One Billion Words Dataset (LM1B; [8]) and OpenWebText (OWT; [18]). We use the `bert-base-uncased` tokenizer for LM1B, and report perplexities on the test split. Models have a context size of 128. For OWT, which does not have a pre-defined split, we reserve the last 100K documents as a held-out validation set and report perplexities on this set. We use the `GPT2` tokenizer [45] for OWT. Models have a context size of 1,024. We utilize the transformer architecture from Lou et al. [33], which augments the diffusion transformer [42] with rotary embeddings [58]. MDLM was trained for 1M or 10M steps (corresponding to 33B, 327B tokens, respectively) on LM1B and 1M steps on OWT (which corresponds to 262B tokens). The corresponding AR baseline was trained for half the number of steps

Table 1: Test perplexities (PPL; ↓) on LM1B. [†]Reported in He et al. [26]. Best diffusion value is bolded.

|  |  | Parameters | PPL ($\downarrow$) |
|---|---|---|---|
| *Autoregressive* | Transformer-X Base [13] | 0.46B | 23.5 |
|  | OmniNet$_T$ [61] | 100M | 21.5 |
| *Diffusion* | BERT-Mouth [64][†] | 110M | $\leq 142.89$ |
|  | D3PM (absorb) [1] | 70M | $\leq 76.90$ |
|  | Diffusion-LM [30][†] | 80M | $\leq 118.62$ |
|  | DiffusionBert [26] | 110M | $\leq 63.78$ |
|  | SEDD [33] (33B tokens) | 110M | $\leq 32.79$ |
| *Autoregressive (Retrained)* | Transformer (33B tokens) | 110M | 22.32 |
|  | Transformer (327B tokens) |  | 20.86 |
| *Diffusion (Ours)* | MDLM (33B tokens) | 110M | $\leq 27.04$ |
|  | MDLM (327B tokens) |  | $\leq \mathbf{23.00}$ |

to ensure similar number of tokens seen (details in Suppl. D.2). Full hyperparameters are given in Suppl. D.4. On OWT, we train with and without time step conditioning.

For representation learning, we pre-train models on the C4 dataset [46], then fine-tune and evaluate models on the GLUE benchmark [65]. Models have a context size of 128. We use the `bert-base-uncased` tokenizer for the representation learning experiments. We utilize the MosaicBERT architecture from Portes et al. [43], an extension of the original BERT architecture [15]. We pre-train a bidirectional MosaicBERT using an MLM objective for 37B tokens of C4, as well as a causal variant on the same data. We further fine-tune MosaicBERT model using the MDLM for 327M tokens, less than 1% of the pre-training data. We provide the full hyperparameters in Suppl. D.6.

**Likelihood Evaluation**   On LM1B, MDLM outperforms all previous diffusion methods (Table 1). Compared to the SEDD baseline reported by Lou et al. [33], trained for 33B tokens, MDLM, which we train for the same amount, achieves a 17% improvement on the perplexity bound. Finally, MDLM gets within 14% of an AR baseline and continues to improve with more training. We see the same trend for models trained on OWT, a larger dataset, shown in Table 2 – MDLM outperforms prior diffusion methods, closing the gap towards AR models. In Table 12 we find that models trained with and without time conditioning attain similar perplexities on OWT. Additionally, Figure 3 demonstrates the reduced variance we achieve from our objective, when compared to previous masked diffusion models such as SEDD [33].

**Zero-Shot Likelihood Evaluation**   We also explore models' ability to generalize by taking models trained on OWT and evaluating how well they model unseen datasets. We compare the perplexities of our MDLM with SEDD [1] and an AR Transformer language model. Our zero-shot datasets include the validation splits of Penn Tree Bank (PTB; [36]), Wikitext [38], LM1B, Lambada [41], AG News [68], and Scientific Papers (Pubmed and Arxiv subsets; [10]). Full experimental details are available in Suppl. D.4.

Table 2: Test perplexities (PPL; ↓) on OWT for models trained for 262B tokens. [†] denotes retrained models.

|  | PPL ($\downarrow$) |
|---|---|
| AR[†] | 17.54 |
| SEDD[†] | $\leq 24.10$ |
| MDLM (Ours) | $\leq \mathbf{23.21}$ |

MDLM consistently outperforms the SEDD diffusion parameterization. In some cases, e.g., for Lambada and Scientific Papers, MDLM attains better perplexity than AR. We hypothesize that these datasets are farther from OWT, and that diffusion models may be more robust to out-of-domain evaluation due to the unmasking-based objective.

**Downstream Task Evaluation**   We find that BERT fine-tuned with MDLM to be a generative model results in strong perplexities while preserving performance on downstream tasks. On the C4 validation set, the AR model attains perplexity (PPL) of 22, the pre-trained BERT attains a PPL upper bound of 78 (evaluated using the MDLM variational bound), and BERT + MDLM-FT attains a PPL upper bound of 35. In Table 4, we further find that BERT + MDLM fine-tuning has no degradation in downstream

Table 3: Zero-shot perplexities (↓) of models trained for 524B tokens on OWT. All perplexities for diffusion models are upper bounds.

|  | PTB | Wikitext | LM1B | Lambada | AG News | Pubmed | Arxiv |
|---|---|---|---|---|---|---|---|
| AR (Retrained) | **82.05** | **25.75** | **51.25** | 51.28 | **52.09** | 49.01 | 41.73 |
| SEDD (Retrained) | 100.09 | 34.28 | 68.20 | 49.86 | 62.09 | 44.53 | 38.48 |
| MDLM (Ours) | 95.26 | 32.83 | 67.01 | **47.52** | 61.15 | **41.89** | **37.37** |

Table 4: GLUE evaluation results. Evaluation measures (↑) are F1 score for QQP and MRPC, Spearman correlations for STS-B, and accuracy for the rest. For MNLI, we report match/mismatch accuracies.

|  | MNLI (m/mm) | QQP | QNLI | SST-2 | COLA | STS-B | MRPC | RTE | Avg |
|---|---|---|---|---|---|---|---|---|---|
| AR | 80.94/80.78 | 86.98 | 86.16 | 90.14 | 33.43 | 84.32 | 83.88 | 47.29 | 74.88 |
| BERT | 84.43/85.35 | 88.41 | **90.46** | **92.20** | 54.81 | **88.41** | 89.16 | 61.37 | 81.62 |
| +MDLM-FT | **84.76/85.07** | **88.49** | 90.30 | **92.20** | **57.69** | 87.48 | **90.53** | **62.09** | **82.06** |

GLUE performance compared to the BERT initialization. While the perplexity of our method is higher than the AR baseline, the downstream task performance is significantly better.

**Semi-Autoregressive Modeling**  To test the SAR decoding algorithm presented in Sec. 4.2, we compare to SSD-LM [24] a diffusion model that was designed to generate blocks of text autoregressively. We generate 200 sequences of length 2048 tokens on a single 3090 GPU and evaluate generative perplexity under a pre-trained GPT-2 [45] model. The SSD-LM sequences are generated using blocks of 25 tokens (as implemented in their pre-trained model) and the MDLM sequences are generated using $L' = 512$. In Table 5, we find that in addition to achieving better generative perplexity, MDLM enables ∼25-30x faster SAR decoding relative to SSD-LM.

Table 5: Semi-AR generative perplexity (Gen. PPL; ↓) for sequences of 2048 tokens.

|  | Gen. PPL (↓) | Sec/Seq (↓) |
|---|---|---|
| SSD-LM | 35.43 | 2473.9 |
| MDLM (Ours) | **27.18** | **89.3** |

## 5.2 Masked Diffusion DNA Models

We also explore applications to the generative modeling of biological sequences [14, 47] using a state space model (SSM) backbone [22]. Namely, we build on the recently-proposed Caduceus DNA language model [50], which uses as a backbone the data-dependent SSM Mamba block [21].

**Experimental Setup**  We pre-train the encoder-only Caduceus [50], which is an MLM, on the HG38 human reference genome [11] and perform fine-tuning using our diffusion parameterization. We use a context length of 1024 tokens and follow Schiff et al. [50] for the experimental setup, other than learning rate which was reduced to 1e-3. See Suppl. D.7 for full experimental details. We assess both generative performance using perplexity and downstream performance on Genomics Benchmarks [20] across language diffusion paradigms and AR models.

**Generative Performance**  We fine-tune the Caduceus MLM across diffusion parameterizations and compare perplexities against AR models. We report perplexity values in Table 6. MDLM outperforms all other diffusion language modeling schemes.

**Downstream Task Fine-tuning**  We perform downstream evaluation with the Genomics Benchmarks [20], a recently proposed benchmark with eight regulatory element classification tasks. As shown in Table 7, our generative fine-tuning paradigm preserves or improves upon downstream performance from MLM pre-training. Absorbing-state diffusion methods outperform Plaid across tasks except for the simplest task Human vs. Worm, where all methods have roughly the same performance. For tasks where the input is a biased subsample of the full genome, we observe that the correlation between perplexity and downstream performance is weaker; see Suppl. D.7.

Table 6: Test perplexities (PPL; ↓) of generative fine-tuning of the Caduceus MLM [50] on the HG38 reference genome. Best diffusion model values are bolded. Error bars indicate the difference between the maximum and minimum values across 5 random seeds used for fine-tuning.

|  |  | Params | PPL (↓) |
|---|---|---|---|
| *Autoregressive (Retrained)* | Mamba | 465K | $3.067 \pm .010$ |
|  | HyenaDNA | 433K | $3.153 \pm .001$ |
| *Diffusion (Retrained)* | Plaid | 507K | $\leq 3.240 \pm .005$ |
|  | SEDD | 467K | $\leq 3.216 \pm .003$ |
| *Diffusion (Ours)* | MDLM | 467K | $\leq \mathbf{3.199} \pm .010$ |

Table 7: Genomic Benchmarks. Top-1 accuracy (↑) across 5-fold cross-validation (CV) for a pre-trained AR Mamba, and a pre-trained Caduceus model fine-tuned with different diffusion parameterizations. The best values per task are bolded and the second best are italicized. Error bars indicate the difference between the maximum and minimum values across 5 random seeds used for CV.

| Model
Fine-Tuning Objective
(Parameter Count) | Mamba
AR
(465K) | Caduceus
MLM
(467K) | Caduceus
Plaid
(507k) | Caduceus
SEDD
(467k) | Caduceus
MDLM (ours)
(467k) |
|---|---|---|---|---|---|
| Mouse Enhancers | 0.763 {±0.008} | **0.810** {±0.016} | 0.745 {±0.079} | 0.784 {±0.058} | *0.795* {±0.029} |
| Coding vs. Intergenomic | 0.897 {±0.004} | **0.913** {±0.003} | *0.908* {±0.003} | **0.913** {±0.005} | **0.913** {±0.003} |
| Human vs. Worm | 0.967 {±0.002} | *0.970* {±0.002} | **0.971** {±0.001} | *0.970* {±0.003} | *0.970* {±0.003} |
| Human Enhancers Cohn | 0.734 {±0.027} | 0.737 {±0.001} | *0.743* {±0.010} | **0.746** {±0.015} | *0.743* {±0.016} |
| Human Enhancer Ensembl | 0.856 {±0.003} | **0.907** {±0.003} | 0.885 {±0.003} | *0.905* {±0.006} | 0.899 {±0.004} |
| Human Regulatory | 0.861 {±0.008} | **0.874** {±0.003} | *0.868* {±0.010} | 0.828 {±0.037} | *0.868* {±0.004} |
| Human OCR Ensembl | 0.806 {±0.005} | *0.821* {±0.000} | 0.820 {±0.004} | 0.816 {±0.008} | **0.823** {±0.008} |
| Human NonTATA Promoters | 0.926 {±0.008} | *0.935* {±0.014} | *0.935* {±0l007} | *0.935* {±0.014} | **0.940** {±0.007} |

## 5.3 Ablation Analysis

In Table 8, we can see the effect of our streamlined masked diffusion implementation. The improvements described in Sec. 3.5.1 allow us to greatly reduce perplexity of previously discounted models, such as D3PM (see the bottom row of this table, which is mathematically equivalent to the D3PM formulation). While most works assumed that D3PM achieves mediocre log-likelihoods, we show that is incorrect: our re-implementation almost matches state-of-the-art score-based methods. This introduces a new strong baseline that opens new research opportunities. Additionally, in Table 8, we ablate

Table 8: Test perplexities (PPL; ↓) for MDLM ablations on LM1B. For the discrete-time models, we use $T = 1000$. Standard deviation is measured over 5 seeds during evaluation.

|  | PPL (≤) |
|---|---|
| MDLM (47) | $\mathbf{27.04} \pm .01$ |
| w/o continuous time (43) | $27.19 \pm .07$ |
| & w/o carry-over (41) | $28.56 \pm .15$ |
| & w/o zero masking (39) | $28.51 \pm .15$ |

different components of MDLM. We observe that the perplexity for MDLM trained with a discrete $T = 1000$ marginally worsens by 0.1 compared to MDLM trained in continuous time. Additionally, removing the "carry over" operation from the SUBS parameterization increases the perplexity by 1.5 points. However, further removing the "zero masking" operation does not lead to any meaningful change in perplexity. We provide further ablations for the continuous time formulation in the Appendix, showing in Table 11 that for a pre-trained model, at inference, increasing $T$ yields better likelihoods.

## 6 Related Work

**Comparison to D3PM** Masked diffusion is a strict subset of D3PM [1]; setting $Q_{t|s} = \alpha_{t|s}\mathbf{I} + (1 - \alpha_{t|s})\mathbf{1m}^\top$ in their framework yields our forward diffusion. We improve over D3PM in three ways: (1) we adopt the SUBS parameterization; (2) this allows us to derive a simplified objective that analytically simplifies certain expectations to zero; (3) we adopt well-engineered training recipes that improve performance. Both (1) and (2) are possible because we focus on masking instead of developing a general discrete diffusion framework. Surprisingly, (3) has the largest contribution to performance.

**Comparison to CTMC** Most implementations of diffusion work best in continuous time. However, extending D3PM in this way requires computing the limit of the product of an infinite number of matrices $Q_T \cdot Q_{T-1} \cdots Q_t$ as $T \to \infty$, which requires advanced CTMC theory [5]. Our work describes simple continuous-time formulations for the most common noise processes (e.g., masking and uniform $\boldsymbol{\pi}$), thus helping make an important part of the literature more accessible. In Suppl. C, we show that our results are compatible with CTMC, using the rate forward matrix $R_t = \frac{\alpha_t'}{\alpha_t}(\mathbf{I} - \mathbf{1m}^\top)$ and the reverse rate $\tilde{R}_t(\mathbf{y}', \mathbf{y})$ for the transition $\mathbf{y} \to \mathbf{y}'$, where $\mathbf{y}, \mathbf{y}' \in \mathcal{V}$:

$$\tilde{R}_t(\mathbf{y}', \mathbf{y}) = -\frac{\alpha_t'}{1-\alpha_t}[\mathbf{y}']^\top [\mathbf{x}_\theta(\mathbf{y}, t) - \mathbf{m}]\langle \mathbf{y}, \mathbf{m} \rangle \tag{12}$$

**Comparison to Score Estimation** Score-based approaches to diffusion [55] extend to discrete states, although they typically further build upon advanced CTMC theory. In particular, SEDD [33] optimizes an ELBO[3] that is a function of the score model, obtaining state-of-the-art log-likelihoods among diffusion models. Our approach, however, is much simpler and does not require advanced theory. Furthermore, we can extract the score for MDLM (76), as demonstrated in Suppl. C.3, making it compatible with various techniques designed for score-based algorithms, such as samplers [5], score parameterization [33], efficient designs of the denoising network [59], guidance techniques, and more.

**Comparison to BERT** Our work provides a principled way of making BERT generative when trained with randomized masking rates. Previous work on generating from BERT used Gibbs sampling or ad-hoc methods [17, 32, 64]. The connection between BERT and diffusion was first made by Austin et al. [1]: their objective effectively involves unmasking. He et al. [26] additionally starts training from a pretrained BERT. However, both works use an objective that is similar to (9), which is less numerically stable than our objective (see Section 3.5.1). Austin et al. [1] mention in their appendix that their ELBO simplifies to a weighted masking (MLM) loss similar to (8), but it uses a more complex formula for the weights and is limited to the discrete time setting unlike our work. Furthermore, they do not train with that objective. Our work derives a simpler expression for the average of MLM losses, implements it, and obtains better likelihoods.

**Comparision to Latent Diffusion LMs** In contrast to this work, which defines diffusion over discrete structures, Plaid [23] and Diffusion LM [30] define a Gaussian diffusion process over word embeddings. Zhang et al. [67] and Hu et al. [28] extend this approach to flow matching over word embeddings, enabling the design of faster samplers. Discrete Flow Matching (DFM) [6] applies flow matching to discrete structures, using a cross-entropy loss as their training objective: $-\mathbb{E}_{q,t}\log p_\theta(\mathbf{x}^{1:L}|\mathbf{z}_t^{1:L})$. Similar to Chang et al. [7], DFM's objective, while effective, is not weighted to serve as a proper ELBO. In MDLM, however, we derive a tight, principled lower bound on the log-likelihood.

**Concurrent Works** Concurrent to our work, Shi et al. [51] and Ou et al. [40] derive a similar simplified objective for masked diffusion processes. While Ou et al. [40] start from a score matching perspective, we tackle this problem from a variational lens similar to Shi et al. [51]. Similar to Ou et al. [40], we formulate efficient samplers in Section 4.1 by leveraging a time-independent denoising network.

A key differentiation between our work and that of Shi et al. [51], Ou et al. [40] is the semi-autoregressive decoding method we present in Section 4.2. While [51, 40] are restricted to sample sequences of a fixed length, we propose samplers to generate arbitrary lengths of text like a traditional language model. Furthermore, we establish the connection between our simplified objective and the masked language modeling (MLM) objective. As a result, we endow BERT-style models with principled generation capabilities while maintaining representation learning capabilities. Whereas [51, 40] only evaluate on NLP datasets, we show that masked diffusion is also effective in modeling biological sequences.

## 7   Conclusion

In this work, we explore masked diffusion. With a well-engineered implementation that supports a simple variational objective, we attain state-of-the-art diffusion perplexities on language benchmarks and demonstrate how to efficiently convert BERT-style encoders into generative models. Given we are working on language modeling, we carry any of the inherent risks and opportunities that come with this line of research.

---

[3]Lou et al. [33] mention that their ELBO can be derived from Benton et al. [4], Hanson [25] but do not provide an explicit derivation. To address this, we present a rigorous derivation in Suppl. C.2 using CTMC theory.

## Acknowledgments and Disclosure of Funding

This work was partially funded by the National Science Foundation under awards DGE-1922551, CAREER awards 2046760 and 2145577, and by the National Institute of Health under award MIRA R35GM151243. Marianne Arriola is supported by a NSF Graduate Research Fellowship under award DGE-2139899 and a Hopper-Dean/Bowers CIS Deans Excellence Fellowship.

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

# Contents

# Appendices

## Appendix A   Discrete time ELBO

This section is organized as follows: First, we derive the expressions for the true posterior and the approximate posterior as outlined in Suppl. A.1. We then simplify these expressions specifically for the case of absorbing state diffusion in Suppl. A.2. Finally, we derive the expression for the ELBO for absorbing state diffusion in Suppl. A.2.3.

### A.1   Generic case

Given the state transition matrix $Q_t$, prior $\boldsymbol{\pi}$, and the latent variables $\mathbf{z}_s$ and $\mathbf{z}_t$, where $s < t$, let

$$Q_{t|s} = \alpha_{t|s}\mathbf{I} + (1-\alpha_{t|s})\mathbf{1}\boldsymbol{\pi}^\top. \tag{13}$$

### A.1.1   $q(\mathbf{z}_t|\mathbf{z}_s)$

Thus, the marginals in (3) correspond to the following forward process:

$$
\begin{aligned}
q(\mathbf{z}_t|\mathbf{z}_s) &= \mathrm{Cat}(\mathbf{z}_t; Q_{t|s}^\top \mathbf{z}_s) \\
&= \mathrm{Cat}(\mathbf{z}_t; [\alpha_{t|s}\mathbf{I} + (1-\alpha_{t|s})\mathbf{1}\boldsymbol{\pi}^\top]^\top \mathbf{z}_s) \\
&= \mathrm{Cat}(\mathbf{z}_t; \alpha_{t|s}\mathbf{z}_s + (1-\alpha_{t|s})\boldsymbol{\pi}\mathbf{1}^\top \mathbf{z}_s) && \because \mathbf{1}^\top \mathbf{z}_s = 1 \\
&= \mathrm{Cat}(\mathbf{z}_t; \alpha_{t|s}\mathbf{z}_s + (1-\alpha_{t|s})\boldsymbol{\pi}).
\end{aligned}
\tag{14}
$$

The above equation indicates that during each diffusion step from $s \to t$, a fraction $(1-\alpha_{t|s})$ of the probability mass is transferred to the prior distribution $\boldsymbol{\pi}$.

### A.1.2  $q(\mathbf{z}_s|\mathbf{z}_t,\mathbf{x})$

Austin et al. [1] show that the posterior corresponding to (14) is given as follows:

$$q(\mathbf{z}_s|\mathbf{z}_t,\mathbf{x})=\mathrm{Cat}\left(\mathbf{z}_s;\frac{Q_{t|s}\mathbf{z}_t\odot Q_s^\top\mathbf{x}}{\mathbf{z}_t^\top Q_t^\top\mathbf{x}}\right), \tag{15}$$

which we simplify to the following:

$$
\begin{aligned}
&q(\mathbf{z}_s|\mathbf{z}_t,\mathbf{x})\\
&=\mathrm{Cat}\left(\mathbf{z}_s;\frac{[\alpha_{t|s}\mathbf{I}+(1-\alpha_{t|s})\mathbf{1}\boldsymbol{\pi}^\top]\mathbf{z}_t\odot[\alpha_s\mathbf{I}+(1-\alpha_s)\mathbf{1}\boldsymbol{\pi}^\top]^\top\mathbf{x}}{\mathbf{z}_t^\top[\alpha_t\mathbf{I}+(1-\alpha_t)\mathbf{1}\boldsymbol{\pi}^\top]^\top\mathbf{x}}\right)\\
&=\mathrm{Cat}\left(\mathbf{z}_s;\frac{[\alpha_{t|s}\mathbf{z}_t+(1-\alpha_{t|s})\mathbf{1}\boldsymbol{\pi}^\top\mathbf{z}_t]\odot[\alpha_s\mathbf{x}+(1-\alpha_s)\boldsymbol{\pi}]}{\mathbf{z}_t^\top[\alpha_t\mathbf{x}+(1-\alpha_t)\boldsymbol{\pi}\mathbf{1}^\top\mathbf{x}]}\right)\\
&=\mathrm{Cat}\left(\mathbf{z}_s;\frac{[\alpha_{t|s}\mathbf{z}_t+(1-\alpha_{t|s})\mathbf{1}\boldsymbol{\pi}^\top\mathbf{z}_t]\odot[\alpha_s\mathbf{x}+(1-\alpha_s)\boldsymbol{\pi}]}{\alpha_t\mathbf{z}_t^\top\mathbf{x}+(1-\alpha_t)\mathbf{z}_t^\top\boldsymbol{\pi}}\right). \qquad \because\mathbf{1}^\top\mathbf{x}=1 \tag{16}
\end{aligned}
$$

### A.1.3  $p_\theta(\mathbf{z}_s|\mathbf{z}_t)$

Austin et al. [1] approximate the reverse process in the following manner:

$$p_\theta(\mathbf{z}_s|\mathbf{z}_t)=q(\mathbf{z}_s|\mathbf{z}_t,\mathbf{x}=\mathbf{x}_\theta(\mathbf{z}_t,t))=\mathrm{Cat}\left(\mathbf{z}_s;\frac{Q_{t|s}\mathbf{z}_t\odot Q_s^\top\mathbf{x}_\theta(\mathbf{z}_t,t)}{\mathbf{z}_t^\top Q_t^\top\mathbf{x}_\theta(\mathbf{z}_t,t)}\right). \tag{17}$$

where $\mathbf{x}_\theta(\mathbf{z}_t,t):\mathcal{V}\times[0,1]\rightarrow\Delta^K$ is an approximation for $\mathbf{x}$.

## A.2  Absorbing state

For the absorbing state diffusion process we have $\boldsymbol{\pi}=\mathbf{m}$.

### A.2.1  $q(\mathbf{z}_s|\mathbf{z}_t,\mathbf{x})$

Since, $\mathbf{z}_t\in\{\mathbf{x},\mathbf{m}\}$, takes only 2 values we consider the separate cases: $\mathbf{z}_t=\mathbf{x}$ and $\mathbf{z}_t=\mathbf{m}$.

**Case 1.**  Consider the case $\mathbf{z}_t=\mathbf{x}$ i.e. $\mathbf{z}_t$ is unmasked. From (16), we have the following:

$$
\begin{aligned}
&q(\mathbf{z}_s|\mathbf{z}_t=\mathbf{x},\mathbf{x})\\
&=\mathrm{Cat}\left(\mathbf{z}_s;\frac{[\alpha_{t|s}\mathbf{x}+(1-\alpha_{t|s})\mathbf{1}\mathbf{m}^\top\mathbf{x}]\odot[\alpha_s\mathbf{x}+(1-\alpha_s)\mathbf{m}]}{\alpha_t\mathbf{x}^\top\mathbf{x}+(1-\alpha_t)\mathbf{x}^\top\mathbf{m}}\right)\\
&=\mathrm{Cat}\left(\mathbf{z}_s;\frac{[\alpha_{t|s}\mathbf{x}]\odot[\alpha_s\mathbf{x}+(1-\alpha_s)\mathbf{m}]}{\alpha_t}\right) \qquad \because\mathbf{x}^\top\mathbf{m}=0\\
&=\mathrm{Cat}\left(\mathbf{z}_s;\frac{\alpha_t\mathbf{x}}{\alpha_t}\right) \qquad \because\mathbf{x}\odot\mathbf{m}=\mathbf{0}\ \text{and}\ \alpha_t=\alpha_{t|s}\alpha_s\\
&=\mathrm{Cat}(\mathbf{z}_s;\mathbf{x}) \qquad \because\alpha_t=\alpha_{t|s}\alpha_s \tag{18}
\end{aligned}
$$

Thus, we have the following:

$$q(\mathbf{z}_s|\mathbf{z}_t=\mathbf{x},\mathbf{x})=\mathrm{Cat}(\mathbf{z}_s;\mathbf{x}). \tag{19}$$

**Case 2.**  Consider the case $\mathbf{z}_t=\mathbf{m}$. By substituting $\mathbf{z}_t=\mathbf{m}$ and $\boldsymbol{\pi}=\mathbf{m}$ in (16), $q(\mathbf{z}_s|\mathbf{z}_t,\mathbf{x})$ simplifies to the following:

$$
\begin{aligned}
q(\mathbf{z}_s|\mathbf{z}_t=\mathbf{m},\mathbf{x})&=\mathrm{Cat}\left(\frac{(\alpha_{t|s}\mathbf{m}+(1-\alpha_{t|s})\mathbf{1})\odot(\alpha_s\mathbf{x}+(1-\alpha_s)\mathbf{m})}{(1-\alpha_t)}\right)\\
&=\mathrm{Cat}\left(\frac{(\alpha_{t|s}(1-\alpha_s)\mathbf{m}+(1-\alpha_{t|s})(1-\alpha_s)\mathbf{m}+(\alpha_s-\alpha_t)\mathbf{x})}{(1-\alpha_t)}\right)
\end{aligned}
$$

$$=\mathrm{Cat}\left(\mathbf{z}_s;\frac{(1-\alpha_s)\mathbf{m}+(\alpha_s-\alpha_t)\mathbf{x}}{1-\alpha_t}\right) \tag{20}$$

Note that the above categorical distribution is non-zero for $\mathbf{z}_s\in\{\mathbf{x},\mathbf{m}\}$ and zero for every other value. The non-zero values are specified as follows:

$$q(\mathbf{z}_s=\mathbf{x}|\mathbf{z}_t=\mathbf{m},\mathbf{x})=\frac{\alpha_s-\alpha_t}{1-\alpha_t} \tag{21}$$

$$q(\mathbf{z}_s=\mathbf{m}|\mathbf{z}_t=\mathbf{m},\mathbf{x})=\frac{1-\alpha_s}{1-\alpha_t} \tag{22}$$

Combining Cases 1 and 2, we get:

$$q(\mathbf{z}_s|\mathbf{z}_t,\mathbf{x})=\begin{cases}\mathrm{Cat}(\mathbf{z}_s;\mathbf{z}_t) & \mathbf{z}_t\neq\mathbf{m},\\ \mathrm{Cat}\left(\mathbf{z}_s;\frac{(1-\alpha_s)\mathbf{m}+(\alpha_s-\alpha_t)\mathbf{x}}{1-\alpha_t}\right) & \mathbf{z}_t=\mathbf{m}.\end{cases} \tag{23}$$

### A.2.2 $p_\theta(\mathbf{z}_s|\mathbf{z}_t)$

For the absorbing state diffusion process with $\boldsymbol{\pi}=\mathbf{m}$, we want to simplify the (17). For this reason, we consider 2 cases: first, when $\mathbf{z}_t\neq\mathbf{m}$ (**case 1**), second, when $\mathbf{z}_t\neq\mathbf{m}$ (**case 2**).

**Case 1.** Consider the case when $\mathbf{z}_t\neq\mathbf{m}$. (17) simplifies to the following:

$$p_\theta(\mathbf{z}_s|\mathbf{z}_t\neq\mathbf{m})=\mathrm{Cat}\left(\mathbf{z}_s;\frac{Q_{t|s}\mathbf{z}_t\odot Q_s^\top\mathbf{x}_\theta(\mathbf{z}_t,t)}{\mathbf{z}_t^\top Q_t^\top\mathbf{x}_\theta(\mathbf{z}_t,t)}\right)$$

$$=\mathrm{Cat}\left(\mathbf{z}_s;\frac{Q_{t|s}\mathbf{z}_t\odot Q_s^\top\mathbf{x}_\theta(\mathbf{z}_t,t)}{[Q_t\mathbf{z}_t]^\top\mathbf{x}_\theta(\mathbf{z}_t,t)}\right)$$

$$=\mathrm{Cat}\left(\mathbf{z}_s;\frac{[\alpha_{t|s}\mathbf{z}_t]\odot[\alpha_s\mathbf{I}+(1-\alpha_s)\mathbf{m}\mathbf{1}^\top]\mathbf{x}_\theta(\mathbf{z}_t,t)}{[\alpha_t\mathbf{z}_t]^\top\mathbf{x}_\theta(\mathbf{z}_t,t)}\right)$$

$$=\mathrm{Cat}\left(\mathbf{z}_s;\frac{[\alpha_{t|s}\mathbf{z}_t]\odot[\alpha_s\mathbf{x}_\theta(\mathbf{z}_t,t)+(1-\alpha_s)\mathbf{m}\langle\mathbf{1},\mathbf{x}_\theta(\mathbf{z}_t,t)\rangle]}{\alpha_t\langle\mathbf{z}_t,\mathbf{x}_\theta(\mathbf{z}_t,t)\rangle}\right)$$

since $\langle\mathbf{1},\mathbf{x}_\theta(\mathbf{z}_t,t)\rangle=1$, we have the following:

$$=\mathrm{Cat}\left(\mathbf{z}_s;\frac{[\alpha_{t|s}\mathbf{z}_t]\odot[\alpha_s\mathbf{x}_\theta(\mathbf{z}_t,t)+(1-\alpha_s)\mathbf{m}]}{\alpha_t\langle\mathbf{z}_t,\mathbf{x}_\theta(\mathbf{z}_t,t)\rangle}\right)$$

since $\mathbf{z}_t\odot\mathbf{m}=\mathbf{0}$, we have the following:

$$=\mathrm{Cat}\left(\mathbf{z}_s;\frac{\alpha_t\mathbf{z}_t\odot\mathbf{x}_\theta(\mathbf{z}_t,t)}{\alpha_t\langle\mathbf{z}_t,\mathbf{x}_\theta(\mathbf{z}_t,t)\rangle}\right)$$

$$=\mathrm{Cat}(\mathbf{z}_s;\mathbf{z}_t) \tag{24}$$

**Case 2.** Consider the case when $\mathbf{z}_t=\mathbf{m}$. (17) simplifies to the following:

$$p_\theta(\mathbf{z}_s|\mathbf{z}_t=\mathbf{m})=\mathrm{Cat}\left(\mathbf{z}_s;\frac{Q_{t|s}\mathbf{m}\odot Q_s^\top\mathbf{x}_\theta(\mathbf{z}_t,t)}{\mathbf{m}^\top Q_t^\top\mathbf{x}_\theta(\mathbf{z}_t,t)}\right)$$

$$=\mathrm{Cat}\left(\mathbf{z}_s;\frac{Q_{t|s}\mathbf{m}\odot Q_s^\top\mathbf{x}_\theta(\mathbf{z}_t,t)}{[Q_t\mathbf{m}]^\top\mathbf{x}_\theta(\mathbf{z}_t,t)}\right)$$

$$=\mathrm{Cat}\left(\mathbf{z}_s;\frac{[\alpha_{t|s}\mathbf{m}+(1-\alpha_{t|s})\mathbf{1}]\odot[\alpha_s\mathbf{I}+(1-\alpha_s)\mathbf{m}\mathbf{1}^\top]\mathbf{x}_\theta(\mathbf{z}_t,t)}{[\alpha_t\mathbf{m}+(1-\alpha_t)\mathbf{1}]^\top\mathbf{x}_\theta(\mathbf{z}_t,t)}\right)$$

$$=\mathrm{Cat}\left(\mathbf{z}_s;\frac{[\alpha_{t|s}\mathbf{m}+(1-\alpha_{t|s})\mathbf{1}]\odot[\alpha_s\mathbf{x}_\theta(\mathbf{z}_t,t)+(1-\alpha_s)\mathbf{m}\langle\mathbf{1},\mathbf{x}_\theta(\mathbf{z}_t,t)\rangle]}{\alpha_t\langle\mathbf{m},\mathbf{x}_\theta(\mathbf{z}_t,t)\rangle+(1-\alpha_t)\langle\mathbf{1},\mathbf{x}_\theta(\mathbf{z}_t,t)\rangle}\right)$$

$$=\mathrm{Cat}\left(\mathbf{z}_s;\frac{[\alpha_{t|s}\mathbf{m}+(1-\alpha_{t|s})\mathbf{1}]\odot[\alpha_s\mathbf{x}_\theta(\mathbf{z}_t,t)+(1-\alpha_s)\mathbf{m}]}{\alpha_t\langle\mathbf{x}_\theta(\mathbf{z}_t,t),\mathbf{m}\rangle+(1-\alpha_t)}\right)$$

$$=\mathrm{Cat}\left(\mathbf{z}_s;\frac{\alpha_t\mathbf{m}\odot\mathbf{x}_\theta(\mathbf{z}_t,t)+(\alpha_s-\alpha_t)\mathbf{x}_\theta(\mathbf{z}_t,t)+(1-\alpha_s)\mathbf{m}}{\alpha_t\langle\mathbf{x}_\theta(\mathbf{z}_t,t),\mathbf{m}\rangle+(1-\alpha_t)}\right) \tag{25}$$

Note that the above categorical distribution, we can obtain the values for $p_\theta(\mathbf{z}_s=\mathbf{x}|\mathbf{z}_t=\mathbf{m})$ and $p_\theta(\mathbf{z}_s=\mathbf{m}|\mathbf{z}_t=\mathbf{m})$ which are as follows:

$$p_\theta(\mathbf{z}_s=\mathbf{x}|\mathbf{z}_t=\mathbf{m})=\frac{(\alpha_s-\alpha_t)\langle\mathbf{x}_\theta(\mathbf{z}_t,t),\mathbf{x}\rangle}{\alpha_t\langle\mathbf{x}_\theta(\mathbf{z}_t,t),\mathbf{m}\rangle+(1-\alpha_t)} \tag{26}$$

$$p_\theta(\mathbf{z}_s=\mathbf{m}|\mathbf{z}_t=\mathbf{m})=\frac{\alpha_s\langle\mathbf{x}_\theta(\mathbf{z}_t,t),\mathbf{m}\rangle+(1-\alpha_s)}{\alpha_t\langle\mathbf{x}_\theta(\mathbf{z}_t,t),\mathbf{m}\rangle+(1-\alpha_t)} \tag{27}$$

As a sanity check, we can verify that (26) reduces to (21), and (27) reduces to (22) if our denoising network can reconstruct $\mathbf{x}$ perfectly, i.e., $\mathbf{x}_\theta(\mathbf{z}_t,t)=\mathbf{x}$.

Combining (24) and (25), we get the following expression for the reverse process parameterization:

$$p_\theta(\mathbf{z}_s|\mathbf{z}_t)=\begin{cases}\mathrm{Cat}(\mathbf{z}_s;\mathbf{z}_t) & \mathbf{z}_t\neq\mathbf{m},\\ \mathrm{Cat}\left(\mathbf{z}_s;\frac{\alpha_t\mathbf{m}\odot\mathbf{x}_\theta(\mathbf{z}_t,t)+(\alpha_s-\alpha_t)\mathbf{x}_\theta(\mathbf{z}_t,t)+(1-\alpha_s)\mathbf{m}}{\alpha_t\langle\mathbf{x}_\theta(\mathbf{z}_t,t),\mathbf{m}\rangle+(1-\alpha_t)}\right) & \mathbf{z}_t=\mathbf{m}.\end{cases} \tag{28}$$

### A.2.3   Diffusion Loss

For a given $T$, Let $\mathcal{L}_T=\mathbb{E}_{t\in\{\frac{1}{T},\frac{2}{T},\dots,1\}}\mathbb{E}_{q(\mathbf{z}_t|\mathbf{x})}T\mathrm{D}_{\mathrm{KL}}(q(\mathbf{z}_s|\mathbf{z}_t,\mathbf{x})\|p_\theta(\mathbf{z}_s|\mathbf{z}_t))$ denote the diffusion loss. We break down the computation of $\mathrm{D}_{\mathrm{KL}}(q(\mathbf{z}_s|\mathbf{z}_t,\mathbf{x})\|p_\theta(\mathbf{z}_s|\mathbf{z}_t))$ into 2 cases: $\mathbf{z}_t=\mathbf{x}$ (**case 1**) and $\mathbf{z}_t=\mathbf{m}$ (**case 2**).

**Case 1:**   consider the case $\mathbf{z}_t=\mathbf{x}$. Let's simplify $\mathrm{D}_{\mathrm{KL}}(q(\mathbf{z}_s|\mathbf{z}_t=\mathbf{x},\mathbf{x})\|p_\theta(\mathbf{z}_s|\mathbf{z}_t=\mathbf{x}))$.

$$\begin{aligned}&\mathrm{D}_{\mathrm{KL}}(q(\mathbf{z}_s|\mathbf{z}_t=\mathbf{x},\mathbf{x})\|p_\theta(\mathbf{z}_s|\mathbf{z}_t=\mathbf{x}))\\ &=\mathrm{D}_{\mathrm{KL}}(\mathbf{z}_t\|\mathbf{z}_t) \qquad\qquad\qquad\qquad\text{From (23) and (24)}\\ &=0\end{aligned} \tag{29}$$

**Case 2:**   Consider the case $\mathbf{z}_t=\mathbf{m}$. Let's simplify $\mathrm{D}_{\mathrm{KL}}(q(\mathbf{z}_s|\mathbf{z}_t=\mathbf{m},\mathbf{x})\|p_\theta(\mathbf{z}_s|\mathbf{z}_t=\mathbf{m}))$.

$$\begin{aligned}&\mathrm{D}_{\mathrm{KL}}(q(\mathbf{z}_s|\mathbf{z}_t=\mathbf{m},\mathbf{x})\|p_\theta(\mathbf{z}_s|\mathbf{z}_t=\mathbf{m}))\\ &=\sum_{\mathbf{z}_s}q(\mathbf{z}_s|\mathbf{z}_t=\mathbf{m},\mathbf{x})\log\frac{q(\mathbf{z}_s|\mathbf{z}_t=\mathbf{m},\mathbf{x})}{p_\theta(\mathbf{z}_s|\mathbf{z}_t=\mathbf{m})}\\ &=\sum_{\mathbf{z}_s\in\{\mathbf{x},\mathbf{m}\}}q(\mathbf{z}_s|\mathbf{z}_t=\mathbf{m},\mathbf{x})\log\frac{q(\mathbf{z}_s|\mathbf{z}_t=\mathbf{m},\mathbf{x})}{p_\theta(\mathbf{z}_s|\mathbf{z}_t=\mathbf{m})}\\ &=\underbrace{q(\mathbf{z}_s=\mathbf{x}|\mathbf{z}_t=\mathbf{m},\mathbf{x})\log\frac{q(\mathbf{z}_s=\mathbf{x}|\mathbf{z}_t=\mathbf{m},\mathbf{x})}{p_\theta(\mathbf{z}_s=\mathbf{x}|\mathbf{z}_t=\mathbf{m})}}_{\text{Simplify using (21) and (26)}}\\ &\quad+\underbrace{q(\mathbf{z}_s=\mathbf{m}|\mathbf{z}_t=\mathbf{m},\mathbf{x})\log\frac{q(\mathbf{z}_s=\mathbf{m}|\mathbf{z}_t=\mathbf{m},\mathbf{x})}{p_\theta(\mathbf{z}_s=\mathbf{m}|\mathbf{z}_t=\mathbf{m})}}_{\text{Simplify using (22) and (27)}}\\ &=\frac{\alpha_s-\alpha_t}{1-\alpha_t}\log\frac{\alpha_t\langle\mathbf{x}_\theta(\mathbf{z}_t,t),\mathbf{m}\rangle+(1-\alpha_t)}{(1-\alpha_t)\langle\mathbf{x}_\theta(\mathbf{z}_t,t),\mathbf{x}\rangle}\\ &\quad+\frac{1-\alpha_s}{1-\alpha_t}\log\frac{(1-\alpha_s)(\alpha_t\langle\mathbf{x}_\theta(\mathbf{z}_t,t),\mathbf{m}\rangle+(1-\alpha_t))}{(1-\alpha_t)(\alpha_s\langle\mathbf{x}_\theta(\mathbf{z}_t,t),\mathbf{m}\rangle+(1-\alpha_s))}\end{aligned} \tag{30}$$

Thus, $\mathrm{D}_{\mathrm{KL}}(q(\mathbf{z}_s|\mathbf{z}_t,\mathbf{x})\|p_\theta(\mathbf{z}_s|\mathbf{z}_t))$ can be written in the following manner where $\langle\mathbf{z}_t,\mathbf{x}\rangle$ evaluates to 1 if $\mathbf{z}_t=\mathbf{x}$ and $\langle\mathbf{z}_t,\mathbf{m}\rangle$ evaluates to 1 if $\mathbf{z}_t=\mathbf{m}$:

$$\mathrm{D}_{\mathrm{KL}}(q(\mathbf{z}_s|\mathbf{z}_t,\mathbf{x})\|p_\theta(\mathbf{z}_s|\mathbf{z}_t))$$

$$= \underbrace{D_{\mathrm{KL}}(q(\mathbf{z}_s|\mathbf{z}_t\!=\!\mathbf{x},\mathbf{x})\|p_\theta(\mathbf{z}_s|\mathbf{z}_t\!=\!\mathbf{x}))}_{=0,\text{ from }(29)}\langle\mathbf{z}_t,\mathbf{x}\rangle + \underbrace{D_{\mathrm{KL}}(q(\mathbf{z}_s|\mathbf{z}_t\!=\!\mathbf{m},\mathbf{x})\|p_\theta(\mathbf{z}_s|\mathbf{z}_t\!=\!\mathbf{m}))}_{\text{Given by }(30)}\langle\mathbf{z}_t,\mathbf{m}\rangle \quad (31)$$

Thus, we derive the diffusion loss, $\mathcal{L}_T$, in the following manner:

$$\mathcal{L}_T = \mathbb{E}_{t\in\left\{\frac{1}{T},\frac{2}{T},\dots,1\right\}}\mathbb{E}_{q(\mathbf{z}_t|\mathbf{x})}T D_{\mathrm{KL}}(q(\mathbf{z}_s|\mathbf{z}_t,\mathbf{x})\|p_\theta(\mathbf{z}_s|\mathbf{z}_t))$$

$$= \mathbb{E}_{t\in\left\{\frac{1}{T},\frac{2}{T},\dots,1\right\}}\mathbb{E}_{q(\mathbf{z}_t|\mathbf{x})}T\left[\frac{\alpha_s-\alpha_t}{1-\alpha_t}\log\frac{\alpha_t\langle\mathbf{x}_\theta(\mathbf{z}_t,t),\mathbf{m}\rangle+(1-\alpha_t)}{(1-\alpha_t)\langle\mathbf{x}_\theta(\mathbf{z}_t,t),\mathbf{x}\rangle}\right.$$

$$\left.+\frac{1-\alpha_s}{1-\alpha_t}\log\frac{(1-\alpha_s)(\alpha_t\langle\mathbf{x}_\theta(\mathbf{z}_t,t),\mathbf{m}\rangle+(1-\alpha_t))}{(1-\alpha_t)(\alpha_s\langle\mathbf{x}_\theta(\mathbf{z}_t,t),\mathbf{m}\rangle+(1-\alpha_s))}\right]\langle\mathbf{z}_t,\mathbf{m}\rangle \quad (32)$$

Note that $\mathcal{L}_T$ is 0 if $\mathbf{z}_t$ is an unmasked token i.e. $\mathbf{z}_t=\mathbf{x}$.

### A.2.4 NELBO

Austin et al. [1], Sohl-Dickstein et al. [54] model $\alpha_i$ as $(\alpha_i)_{i\in\{1,\dots,T\}}=1-\frac{i}{T}$ given latents $\mathbf{z}_{1,\dots,T}$. However, in this paper, we denote the latents as $\mathbf{z}_{t(0),\dots,t(T)}$; and hence, the $\alpha_{t(i)}$ are given as follows:

$$(\alpha_i)_{i\in\{1,\dots,T\}}=1-\frac{i}{T} \qquad\qquad \text{From Austin et al. [1], Sohl-Dickstein et al. [54].}$$

$$\implies (\alpha_i)_{k\in\{1,\dots,T+1\}}=1-\frac{i}{T+1} \qquad\qquad \text{For } T+1 \text{ latents}$$

$$\implies (\alpha_i)_{i\in\{0,\dots,T\}}=1-\frac{i+1}{T+1} \qquad\qquad \text{Offsetting the indices by 1.}$$

$$\implies (\alpha_{t(i)})_{i\in\{0,\dots,T\}}=1-\frac{i+1}{T+1} \qquad \text{Switching the notations from }\alpha_i\text{ to }\alpha_{t(i)}. \quad (33)$$

Consequently, from Equation 33, we derive that

$$\alpha_{t(0)}=\frac{T}{T+1}, \quad (34)$$

$$\alpha_{t(T)}=0. \quad (35)$$

Thus we have the following:

$$\mathbf{z}_{t(0)}\sim\mathrm{Cat}(.;\alpha_{t=0}\mathbf{x}+(1-\alpha_{t=0})\mathbf{m})=\mathrm{Cat}\left(.;\frac{T}{T+1}\mathbf{x}+\frac{1}{T+1}\mathbf{m}\right), \quad (36)$$

$$q(\mathbf{z}_{t(T)}|\mathbf{x})=\mathrm{Cat}(.;\alpha_{t=1}\mathbf{x}+(1-\alpha_{t=1})\mathbf{m})=\mathrm{Cat}(.;\mathbf{m}), \quad (37)$$

$$p_\theta(\mathbf{z}_{t(T)})=\mathrm{Cat}(.;\mathbf{m}) \quad (38)$$

The NELBO (2) simplifies to the following:

$$\mathbb{E}_q\left[-\log p_\theta(\mathbf{x}|\mathbf{z}_{t(0)})+\underbrace{\mathcal{L}_T}_{\text{Compute using }(32)}\right]+\underbrace{D_{\mathrm{KL}}[q(\mathbf{z}_{t(T)}|\mathbf{x})\|p_\theta(\mathbf{z}_{t(T)})]}_{=0\text{ using }(37)\text{ and }(38)}$$

$$=\mathbb{E}_{q,t}\left[-\log p_\theta(\mathbf{x}|\mathbf{z}_{t(0)})+T\left[\frac{\alpha_s-\alpha_t}{1-\alpha_t}\log\frac{\alpha_t\langle\mathbf{x}_\theta(\mathbf{z}_t,t),\mathbf{m}\rangle+(1-\alpha_t)}{(1-\alpha_t)\langle\mathbf{x}_\theta(\mathbf{z}_t,t),\mathbf{x}\rangle}\right.\right.$$

$$\left.\left.+\frac{1-\alpha_s}{1-\alpha_t}\log\frac{(1-\alpha_s)(\alpha_t\langle\mathbf{x}_\theta(\mathbf{z}_t,t),\mathbf{m}\rangle+(1-\alpha_t))}{(1-\alpha_t)(\alpha_s\langle\mathbf{x}_\theta(\mathbf{z}_t,t),\mathbf{m}\rangle+(1-\alpha_s))}\right]\langle\mathbf{z}_t,\mathbf{m}\rangle\right] \quad (39)$$

## Appendix B MDLM

In this section, we show how SUBS parameterization can simplify the functional form of the NELBO as defined in (39).

## B.1 Rao-Blackwellization

We employ the RB techniques as described in Sec. 3.2.3 to simplify the NELBO (39) to (41) using RB2, and further to (43) using RB1.

### B.1.1 Zero Masking Probabilities

Using "Zero Masking Probabilities" (RB2) from Sec. 3.2.3, we set $\langle \mathbf{x}_\theta(\mathbf{z}_t, t), \mathbf{m} \rangle = 0$ in (32) to obtain the following simplified diffusion loss:

$$\mathcal{L}_T^{\text{RB2}} = \mathbb{E}_{t \in \left\{ \frac{1}{T}, \frac{2}{T}, \dots, 1 \right\}} \mathbb{E}_{q(\mathbf{z}_t | \mathbf{x})} T \left[ \frac{\alpha_s - \alpha_t}{1 - \alpha_t} \log \frac{1}{\langle \mathbf{x}_\theta(\mathbf{z}_t, t), \mathbf{x} \rangle} \right] \langle \mathbf{z}_t, \mathbf{m} \rangle$$

$$= \mathbb{E}_{t \in \left\{ \frac{1}{T}, \frac{2}{T}, \dots, 1 \right\}} \mathbb{E}_{q(\mathbf{z}_t | \mathbf{x})} T \left[ \frac{\alpha_t - \alpha_s}{1 - \alpha_t} \log \langle \mathbf{x}_\theta(\mathbf{z}_t, t), \mathbf{x} \rangle \right] \langle \mathbf{z}_t, \mathbf{m} \rangle. \tag{40}$$

The corresponding Rao-Blackwellized NELBO is given as:

$$\mathbb{E}_q \left[ -\log p_\theta(\mathbf{x} | \mathbf{z}_{t(0)}) + \underbrace{\mathcal{L}_T^{\text{RB2}}}_{\text{Compute using (40)}} \right] + \underbrace{D_{\text{KL}}[q(\mathbf{z}_{t(T)} | \mathbf{x}) \| p_\theta(\mathbf{z}_{t(T)})]}_{= 0 \text{ using (37) and (38)}}$$

$$= \mathbb{E}_{q,t} \left[ -\log p_\theta(\mathbf{x} | \mathbf{z}_{t(0)}) + T \left[ \frac{\alpha_t - \alpha_s}{1 - \alpha_t} \log \langle \mathbf{x}_\theta(\mathbf{z}_t, t), \mathbf{x} \rangle \right] \langle \mathbf{z}_t, \mathbf{m} \rangle \right] \tag{41}$$

### B.1.2 Carry Over Unmasking

Notice that the term $\langle \mathbf{z}_t, \mathbf{m} \rangle$ in (40) is intended to reduce the diffusion loss to zero when $\mathbf{z}_t = \mathbf{x}$. Now, we will demonstrate that, by applying "Carry Over Unmasking" (RB1) from Sec. 3.2.3, $\langle \mathbf{z}_t, \mathbf{m} \rangle$ can be removed from (40).

Recall that RB1 guarantees $\mathbf{x}_\theta(\mathbf{z}_t, t) = \mathbf{x}$ when $\mathbf{z}_t = \mathbf{x}$. Thus, with the RB1 parameterization, the diffusion loss in (40) becomes zero for $\mathbf{z}_t = \mathbf{x}$, as $\log \langle \mathbf{x}_\theta(\mathbf{z}_t, t), \mathbf{m} \rangle = 0$. Consequently, $\langle \mathbf{z}_t, \mathbf{m} \rangle$ can be safely omitted from (41), yielding the following diffusion loss:

$$\mathcal{L}_T^{\text{RB2 + RB1}} = \mathbb{E}_{t \in \left\{ \frac{1}{T}, \frac{2}{T}, \dots, 1 \right\}} \mathbb{E}_{q(\mathbf{z}_t | \mathbf{x})} T \left[ \frac{\alpha_t - \alpha_s}{1 - \alpha_t} \log \langle \mathbf{x}_\theta(\mathbf{z}_t, t), \mathbf{x} \rangle \right] \tag{42}$$

### B.1.3 NELBO

Thus, we have the following NELBO:

$$\mathbb{E}_q \left[ -\log p_\theta(\mathbf{x} | \mathbf{z}_{t(0)}) + \underbrace{\mathcal{L}_T^{\text{RB2 + RB1}}}_{\text{Compute using (42)}} \right] + \underbrace{D_{\text{KL}}[q(\mathbf{z}_{t(T)} | \mathbf{x}) \| p_\theta(\mathbf{z}_{t(T)})]}_{= 0 \text{ using (37) and (38)}}$$

$$= \mathbb{E}_{q,t} \left[ -\log p_\theta(\mathbf{x} | \mathbf{z}_{t(0)}) + T \left[ \frac{\alpha_t - \alpha_s}{1 - \alpha_t} \log \langle \mathbf{x}_\theta(\mathbf{z}_t, t), \mathbf{x} \rangle \right] \right] \tag{43}$$

**Comparing (43) and (41).** Note that due to RB1, $\log p_\theta(\mathbf{x} | \mathbf{z}_{t(0)})$ in (43) reduces to 0 every time $\mathbf{z}_{t(0)} = \mathbf{x}$ as explained in (45). However, this is not the case in (41), even though it has a functionally similar expression to (43). Because of this reason (43) should lead to a better likelihood estimate and we empirically verify this in Table 8.

## B.2 Continuous Time

### B.2.1 Diffusion Loss

To derive the continuous-time diffusion loss, $\mathcal{L}_{\text{diffusion}}^\infty$, we consider the limiting case $\lim_{T \to \infty} \mathcal{L}_T^{\text{RB2 + RB1}}$ (42):

$$\mathcal{L}_{\text{diffusion}}^\infty = \lim_{T \to \infty} \mathcal{L}_T^{\text{RB2 + RB1}}$$

$$= \mathbb{E}_{t \in \left\{ \frac{1}{T}, \frac{2}{T}, \dots, 1 \right\}, q(\mathbf{z}_t | \mathbf{x})} \left[ \lim_{T \to \infty} T \frac{\alpha_t - \alpha_s}{1 - \alpha_t} \log \langle \mathbf{x}_\theta(\mathbf{z}_t, t), \mathbf{x} \rangle \right]$$

$$= \mathbb{E}_{t \sim \mathcal{U}[0,1], q(\mathbf{z}_t | \mathbf{x})} \left[ \frac{\alpha'_t}{1 - \alpha_t} \log \langle \mathbf{x}_\theta(\mathbf{z}_t, t), \mathbf{x} \rangle \right] \qquad \text{Using } \lim_{T \to \infty} T(\alpha_t - \alpha_s) = \alpha'_t$$

(44)

### B.2.2 Reconstruction Loss

For the continous time case, from (36), we have

$$\mathbf{z}_{t(0)} \sim \lim_{T \to \infty} \mathrm{Cat}\left( .; \frac{T}{T+1} \mathbf{x} + \frac{1}{T+1} \mathbf{m} \right)$$
$$\implies \mathbf{z}_{t(0)} \sim \mathrm{Cat}(.; \mathbf{x})$$
$$\implies \mathbf{z}_{t(0)} = \mathbf{x}$$

(45)

Thus, the reconstruction loss reduces to 0 in the following manner:

$$
\begin{aligned}
\mathcal{L}_{\text{recons}} &= -\log p_\theta(\mathbf{x} | \mathbf{z}_{t(0)}) \\
&= -\log p_\theta(\mathbf{x} | \mathbf{z}_{t(0)} = \mathbf{x}) && \text{From (45)} \\
&= -\log \langle \mathbf{x}_\theta(\mathbf{x}, t(0)), \mathbf{x} \rangle \\
&= -\log \langle \mathbf{x}, \mathbf{x} \rangle && \text{Due to ``carry-over unmasking'' } \mathbf{x}_\theta(\mathbf{x}, t(0)) = \mathbf{x} \\
&= 0.
\end{aligned}
$$

(46)

### B.2.3 NELBO

Thus, we have the following NELBO:

$$\mathbb{E}_q \left[ \underbrace{-\log p_\theta(\mathbf{x} | \mathbf{z}_{t(0)})}_{= 0 \text{ from (46)}} + \underbrace{\mathcal{L}^\infty_{\text{diffusion}}}_{\text{Compute using (42)}} \right] + \underbrace{D_{\text{KL}}[q(\mathbf{z}_{t(T)} | \mathbf{x}) \| p_\theta(\mathbf{z}_{t(T)}, t)]}_{= 0 \text{ using (37) and (38)}}$$

$$= \boxed{\mathbb{E}_{q,t} \left[ \frac{\alpha'_t}{1 - \alpha_t} \log \langle \mathbf{x}_\theta(\mathbf{z}_t, t), \mathbf{x} \rangle \right]}$$

(47)

### B.3 Final Algorithm

In Algorithm 1, we present the training algorithm for MDLM.

---
**Algorithm 1** Training MDLM
---
1: **repeat**
2:     $\mathbf{x}^{1:L} \sim q(\mathbf{x})$             ▷ Sample a sentence.
3:     $t \sim \mathcal{U}[0,1]$             ▷ Sample a time step.
4:     $\mathbf{z}_t^\ell \sim \mathrm{Cat}(\mathbf{z}_t^\ell; \alpha_t \mathbf{x}^\ell + (1 - \alpha_t)\mathbf{m}) \ \forall \ 1 \le \ell \le L$    ▷ Mask Each token $\mathbf{x}^\ell$ independently to obtain the latent $\mathbf{z}_t^{1:L}$.
5:     Take gradient descent step on

$$\nabla_\theta \frac{\alpha'_t}{1 - \alpha_t} \sum_\ell \log \langle \mathbf{x}_\theta^\ell(\mathbf{z}_t^{1:L}, t), \mathbf{x}^\ell \rangle$$

6: **until** converged

---

## Appendix C    Concrete Score Matching

In the previous section, we defined the discrete diffusion process as a Discrete-Time Markov Chain (DTMC) with a finite set of $T$ states, $\mathbf{z}_{\{0, \frac{1}{T}, \dots, 1\}}$, and a state transition matrix $Q_t$. To derive the continuous-time ELBO, we simply take the limit as $T \to \infty$.

In contrast, Campbell et al. [5] and Lou et al. [33] defined the discrete diffusion process as a Continuous-Time Markov Chain (CTMC), where the forward corruption process is specified by the rate change matrix $R_t \in \mathbb{R}^{|\mathcal{V}| \times |\mathcal{V}|}$, which can be thought of as the instantaneous rate at which one state transitions to another. With this formulation, the forward posterior $q(\mathbf{z}_t|\mathbf{z}_s)$ and the true reverse posterior $q(\mathbf{z}_s|\mathbf{z}_t,\mathbf{x})$ can be expressed in terms of the rate change matrix as follows:

$$q(\mathbf{z}_t=\mathbf{y}'|\mathbf{z}_s=\mathbf{y})=\delta_{\mathbf{y}',\mathbf{y}}+R_t(\mathbf{y}',\mathbf{y})\frac{1}{T}+\mathcal{O}\left(\frac{1}{T^2}\right) \tag{48}$$

$$q(\mathbf{z}_s=\mathbf{y}'|\mathbf{z}_t=\mathbf{y},\mathbf{x})=\delta_{\mathbf{y}',\mathbf{y}}+\tilde{R}_t(\mathbf{y}',\mathbf{y})\frac{1}{T}+\mathcal{O}\left(\frac{1}{T^2}\right) \tag{49}$$

where $\delta$ is the Kroenecker delta and $\mathcal{O}(\frac{1}{T^2})$ represents higher order terms of $\frac{1}{T^2}$. In Sec. C.1, we show how to express the rate matrix $R_t$ in terms of the state transition matrix $Q_t$. Lou et al. [33] propose a continuous time ELBO for this process. They mention that this expression can be derived from Benton et al. [4], though they do not provide an explicit derivation. For this reason, we present a rigorous derivation in Sec. C.2 and further demonstrate that, under the SUBS parameterization in Sec. 3.2.3, this formula reduces to our proposed continuous-time ELBO, given by (10).

For the remainder of this section, we switch to the notation $q_{t|s}(\mathbf{y}'|\mathbf{y})$ to denote $q(\mathbf{z}_t=\mathbf{y}'|\mathbf{z}_s=\mathbf{y})$, $q_{s|t}(\mathbf{y}'|\mathbf{y})$ for $q(\mathbf{z}_s=\mathbf{y}'|\mathbf{z}_t=\mathbf{y})$, and $q(\mathbf{z}_t=\mathbf{y}|\mathbf{x})$ for $q_t(\mathbf{y}|\mathbf{x})$, aligning with the notation typically used in the CTMC literature.

## C.1 Extracting the Rate Matrix

Here, we aim to express the rate change matrix $R_t$ in terms of the state transition matrix $Q_t$. To do this, we first represent the forward transition $q_{t|s}$ in terms of $Q_t$ and $R_t$ separately, allowing us to illustrate their relationship.

Using (13), we can write $q_{t|s}$ as follows:

$$\begin{aligned}
q_{t|s}(\mathbf{y}'|\mathbf{y}) &= [\mathbf{y}']^\top[\alpha_{t|s}\mathbf{I}+(1-\alpha_{t|s})\mathbf{1}\mathbf{m}^\top]^\top\mathbf{y} \\
&= [\mathbf{y}']^\top[\alpha_{t|s}\mathbf{y}+(1-\alpha_{t|s})\mathbf{m}\mathbf{1}^\top\mathbf{y}] \\
&= [\mathbf{y}']^\top[\alpha_{t|s}\mathbf{y}+(1-\alpha_{t|s})\mathbf{m}] \\
&= \alpha_{t|s}[\mathbf{y}']^\top\mathbf{y}+(1-\alpha_{t|s})[\mathbf{y}']^\top\mathbf{m}
\end{aligned} \tag{50}$$

Now let's analyze all possible combinations for the tuple $(\mathbf{y}',\mathbf{y})$:

**1. Case $(\mathbf{y}'=\mathbf{x},\mathbf{y}=\mathbf{x})$:** Using (50), we find that $q_{t|s}(\mathbf{x}|\mathbf{x})=\alpha_{t|s}$ for the DTMC. By (48), we have $q_{t|s}(\mathbf{x}|\mathbf{x})=1+R_t(\mathbf{x},\mathbf{x})\frac{1}{T}$ as $T\to\infty$, since the higher-order terms $\mathcal{O}(\frac{1}{T^2})$ vanish in the limit. Thus, we get:

$$\lim_{T\to\infty}\left[1+R_t(\mathbf{x},\mathbf{x})\frac{1}{T}\right]=\lim_{T\to\infty}\alpha_{t|s}$$

$$\implies R_t(\mathbf{x},\mathbf{x})=\lim_{T\to\infty}T(\alpha_{t|s}-1)=\frac{\alpha'_t}{\alpha_t} \tag{51}$$

**2. Case $(\mathbf{y}'=\mathbf{m},\mathbf{y}\in\mathcal{V}-\{\mathbf{m}\})$:** Similarly, using (50) and (48), we have $q_{t|s}(\mathbf{m}|\mathbf{y}\neq\mathbf{m})=1-\alpha_{t|s}$ and $q_{t|s}(\mathbf{x}|\mathbf{y}\neq\mathbf{m})=R_t(\mathbf{m},\mathbf{y}\neq\mathbf{m})\frac{1}{T}$. Thus,

$$\lim_{T\to\infty}\left[R_t(\mathbf{m},\mathbf{y}\neq\mathbf{m})\frac{1}{T}\right]=\lim_{T\to\infty}[1-\alpha_{t|s}]$$

$$\implies R_t(\mathbf{m},\mathbf{y}\neq\mathbf{m})=\lim_{T\to\infty}T(1-\alpha_{t|s})=-\frac{\alpha'_t}{\alpha_t} \tag{52}$$

**3. Case $(\mathbf{y}'=\mathbf{m},\mathbf{y}=\mathbf{m})$:** Using (50) and (48), we find $q_{t|s}(\mathbf{m}|\mathbf{m})=1$ and $q_{t|s}(\mathbf{m}|\mathbf{m})=1+R_t(\mathbf{m},\mathbf{m})\frac{1}{T}+\mathcal{O}(\frac{1}{T^2})$. Since these two expressions must be equal for any $T$, it follows that

$$R_t(\mathbf{m},\mathbf{m})=0. \tag{53}$$

Note that when $R_t(\mathbf{m},\mathbf{m})$ is constant, the term $\mathcal{O}(\frac{1}{T^2})$ reduces to zero, as it includes higher-order time derivatives of $R_t$.

**4. Case $(\mathbf{y}'=\mathbf{x},\mathbf{y}\in\mathcal{V}-\{\mathbf{x}\})$:**  In the context of absorbing state diffusion, these states are never observed. Thus,

$$R_t(\mathbf{y}'=\mathbf{x},\mathbf{y}\in\mathcal{V}-\{\mathbf{m},\mathbf{x}\})=0 \tag{54}$$

**5. Case $(\mathbf{y}'\in\mathcal{V}-\{\mathbf{m},\mathbf{x}\},\mathbf{y}\in\{\mathbf{m},\mathbf{x}\})$:**  In the context of absorbing state diffusion, these states are never observed. Thus,

$$R_t(\mathbf{y}'\in\mathcal{V}-\{\mathbf{m},\mathbf{x}\},\mathbf{y}\in\{\mathbf{m},\mathbf{x}\})=0 \tag{55}$$

Finally, we can express the forward rate matrix as:

$$\boxed{R_t=\frac{\alpha_t'}{\alpha_t}\left(\mathbf{I}-\mathbf{m}\mathbf{1}^\top\right)} \tag{56}$$

It can be seen that the columns of this matrix sum to zero, i.e.,

$$\sum_{\mathbf{y}'\in\mathcal{V}}R_t(\mathbf{y}',\mathbf{y})=0\implies R_t(\mathbf{y},\mathbf{y})=\sum_{\mathbf{y}'\neq\mathbf{y}}R_t(\mathbf{y}',\mathbf{y}), \tag{57}$$

which ensures that the probability mass is preserved in the forward diffusion process. Similarly, the reverse rate matrix $\tilde{R}_t$ can be written in terms of the forward rate matrix $R_t$ as follows [33]:

$$\tilde{R}_t(\mathbf{y}',\mathbf{y})=\begin{cases}\frac{q_t(\mathbf{y}'|\mathbf{x})}{q_t(\mathbf{y}|\mathbf{x})}R_t(\mathbf{y},\mathbf{y}') & \mathbf{y}'\neq\mathbf{y}\\ -\sum_{\tilde{\mathbf{y}}\neq\mathbf{y}}\frac{q_t(\tilde{\mathbf{y}}|\mathbf{x})}{q_t(\mathbf{y}|\mathbf{x})}R_t(\mathbf{y},\tilde{\mathbf{y}}) & \mathbf{y}'=\mathbf{y}.\end{cases} \tag{58}$$

## C.2 NELBO

Meng et al. [37] introduced the term "concrete score" for the term $q_t(\mathbf{y}'|\mathbf{x})/q_t(\mathbf{y}|\mathbf{x})$ that appears in $\tilde{R}_t$. Since this quantity is not directly accessible in the reverse diffusion process, we approximate it using a neural network, $\mathbf{s}_\theta:\mathcal{V}\to\mathcal{V}$, with parameters $\theta$. The approximate reverse posterior $p_{s|t}$ can then be expressed in terms of the approximate reverse rate matrix $\tilde{R}_t$ in the following manner:

$$p_{s|t}(\mathbf{y}'|\mathbf{y})=\delta_{y',y}+\tilde{R}_t^\theta(\mathbf{y}',\mathbf{y})\frac{1}{T}+\mathcal{O}\left(\frac{1}{T^2}\right) \tag{59}$$

$$\tilde{R}_t^\theta(\mathbf{y}',\mathbf{y})=\begin{cases}\mathbf{s}_\theta(\mathbf{y})_{\mathbf{y}'}R_t(\mathbf{y},\mathbf{y}') & \mathbf{y}'\neq\mathbf{y}\\ -\sum_{\tilde{\mathbf{y}}\neq\mathbf{y}}\mathbf{s}_\theta(\mathbf{y})_{\tilde{\mathbf{y}}}R_t(\mathbf{y},\tilde{\mathbf{y}}) & \mathbf{y}'=\mathbf{y}\end{cases} \tag{60}$$

where $\mathbf{s}_\theta(\mathbf{y})_{\mathbf{y}'}$ denotes the approximate concrete score $q_t(\mathbf{y}'|\mathbf{x})/q_t(\mathbf{y}|\mathbf{x})$. Lou et al. [33] propose the following NELBO to train such a model:

$$\mathbb{E}_{t\in[0,1],\mathbf{y}\sim q_t(.|\mathbf{x})}\left[\sum_{\mathbf{y}'\neq\mathbf{y}}R_t(\mathbf{y},\mathbf{y}')\left(\mathbf{s}_\theta(\mathbf{y})_{\mathbf{y}'}-\frac{q_t(\mathbf{y}'|\mathbf{x})}{q_t(\mathbf{y}|\mathbf{x})}\log\mathbf{s}_\theta(\mathbf{y})_{\mathbf{y}'}+K\left(\frac{q_t(\mathbf{y}'|\mathbf{x})}{q_t(\mathbf{y}|\mathbf{x})}\right)\right)\right] \tag{61}$$

where $K(a)=a\log a-a$. They mention that this expression can be derived from Benton et al. [4], though they do not provide an explicit derivation. For this reason, we present a rigorous derivation in the following section.

**Proof.**  Let's focus on the diffusion loss for this process. As mentioned in the previous section, the reconstruction and prior loss terms reduce to zero. The continuous-time diffusion loss is given by:

$$\lim_{T\to\infty}T\mathbb{E}_{t\in\{\frac{1}{T},\frac{2}{T},\dots,1\},\mathbf{y}\sim q_t(.|\mathbf{x})}[\mathrm{D_{KL}}(q_{s|t}(\mathbf{y}'|\mathbf{y},\mathbf{x})\|p_{s|t}(\mathbf{y}'|\mathbf{y}))]$$

$$=\lim_{T\to\infty}T\mathbb{E}_{t\in\{\frac{1}{T},\frac{2}{T},\dots,1\},\mathbf{y}\sim q_t(.|\mathbf{x})}\left[\sum_{\mathbf{y}'}q_{s|t}(\mathbf{y}'|\mathbf{y},\mathbf{x})\log\frac{q_{s|t}(\mathbf{y}'|\mathbf{y},\mathbf{x})}{p_{s|t}(\mathbf{y}'|\mathbf{y},\mathbf{x})}\right]$$

$$=\mathbb{E}_{t\in[0,1],\mathbf{y}\sim q_t(.|\mathbf{x})}\left[\lim_{T\to\infty}T\sum_{\mathbf{y}'}q_{s|t}(\mathbf{y}'|\mathbf{y},\mathbf{x})\log\frac{q_{s|t}(\mathbf{y}'|\mathbf{y},\mathbf{x})}{p_{s|t}(\mathbf{y}'|\mathbf{y},\mathbf{x})}\right]$$

$$= \mathbb{E}_{t\in[0,1],\mathbf{y}\sim q_t(.|\mathbf{x})}\left[\underbrace{\lim_{T\to\infty}Tq_{s|t}(\mathbf{y}|\mathbf{y},\mathbf{x})\log\frac{q_{s|t}(\mathbf{y}|\mathbf{y},\mathbf{x})}{p_{s|t}(\mathbf{y}|\mathbf{y},\mathbf{x})}}_{\textbf{Term 1}}+\underbrace{\lim_{T\to\infty}T\sum_{\mathbf{y}'\neq\mathbf{y}}q_{s|t}(\mathbf{y}'|\mathbf{y},\mathbf{x})\log\frac{q_{s|t}(\mathbf{y}'|\mathbf{y},\mathbf{x})}{p_{s|t}(\mathbf{y}'|\mathbf{y},\mathbf{x})}}_{\textbf{Term 2}}\right]$$

$$\tag{62}$$

Let's simplify these two terms separately. For the derivation, we'll rely on two key observations: In the limiting case as $T\to\infty$, it follows from (49) that $\lim_{T\to\infty}q_{s|t}(\mathbf{y}|\mathbf{y},\mathbf{x})=1$ and from (59) that $\lim_{T\to\infty}p_{s|t}(\mathbf{y}|\mathbf{y},\mathbf{x})=1$.

**Term 1:**

$$\lim_{T\to\infty}Tq_{s|t}(\mathbf{y}|\mathbf{y},\mathbf{x})\log\frac{q_{s|t}(\mathbf{y}|\mathbf{y},\mathbf{x})}{p_{s|t}(\mathbf{y}|\mathbf{y},\mathbf{x})}$$

$\because \lim_{T\to\infty}q_{s|t}(\mathbf{y}|\mathbf{y},\mathbf{x})=1$; hence,

$$=\lim_{T\to\infty}T\log\frac{q_{s|t}(\mathbf{y}|\mathbf{y},\mathbf{x})}{p_{s|t}(\mathbf{y}|\mathbf{y},\mathbf{x})}$$

The above term is in $\infty\times 0$ indeterminate form; therefore,

$$=\lim_{T\to\infty}T\left(\log q_{s|t}(\mathbf{y}|\mathbf{y},\mathbf{x})-\log p_{s|t}(\mathbf{y}|\mathbf{y},\mathbf{x})\right)$$

Substituting $q_{s|t}$ and $p_{s|t}$ from (49) and (59), we get:

$$=\lim_{T\to\infty}T\left[\log\left(1+\tilde{R}_t(\mathbf{y},\mathbf{y})\frac{1}{T}+\mathcal{O}\left(\frac{1}{T^2}\right)\right)-\log\left(1+\tilde{R}_t^\theta(\mathbf{y},\mathbf{y})\frac{1}{T}+\mathcal{O}\left(\frac{1}{T^2}\right)\right)\right]$$

Applying the Taylor series expansion for $\log(1+x)$, we get:

$$=\lim_{T\to\infty}T\left[\tilde{R}_t(\mathbf{y},\mathbf{y})\frac{1}{T}+\mathcal{O}\left(\frac{1}{T^2}\right)-\tilde{R}_t^\theta(\mathbf{y},\mathbf{y})\frac{1}{T}-\mathcal{O}\left(\frac{1}{T^2}\right)\right]$$

$$=\tilde{R}_t(\mathbf{y},\mathbf{y})+\lim_{T\to\infty}T\mathcal{O}\left(\frac{1}{T^2}\right)-\tilde{R}_t^\theta(\mathbf{y},\mathbf{y})-\lim_{T\to\infty}T\mathcal{O}\left(\frac{1}{T^2}\right)$$

$\because \lim_{T\to\infty}T\mathcal{O}\left(\frac{1}{T^2}\right)=0$, we get:

$$=\tilde{R}_t(\mathbf{y},\mathbf{y})-\tilde{R}_t^\theta(\mathbf{y},\mathbf{y})$$

Using (58) and (60), we get:

$$=-\sum_{\mathbf{y}'\neq\mathbf{y}}R_t(\mathbf{y},\mathbf{y}')\left(\frac{q_t(\mathbf{y}'|\mathbf{x})}{q_t(\mathbf{y}|\mathbf{x})}-\mathbf{s}_\theta(\mathbf{y})_{\mathbf{y}'}\right)\tag{63}$$

**Term 2:**

$$\lim_{T\to\infty}T\sum_{\mathbf{y}'\neq\mathbf{y}}q_{s|t}(\mathbf{y}'|\mathbf{y},\mathbf{x})\log\frac{q_{s|t}(\mathbf{y}'|\mathbf{y},\mathbf{x})}{p_{s|t}(\mathbf{y}'|\mathbf{y},\mathbf{x})}$$

Substituting $q_{s|t}$ and $p_{s|t}$ from (49) and (59), we get:

$$=\lim_{T\to\infty}T\sum_{\mathbf{y}'\neq\mathbf{y}}\left[\left[\frac{q_s(\mathbf{y}'|\mathbf{x})}{q_t(\mathbf{y}|\mathbf{x})}R_t(\mathbf{y},\mathbf{y}')\frac{1}{T}+\mathcal{O}\left(\frac{1}{T^2}\right)\right]\log\frac{\frac{q_s(\mathbf{y}'|\mathbf{x})}{q_t(\mathbf{y}|\mathbf{x})}R_t(\mathbf{y},\mathbf{y}')\frac{1}{T}+\mathcal{O}\left(\frac{1}{T^2}\right)}{\mathbf{s}_\theta(\mathbf{y})_{\mathbf{y}'}R_t(\mathbf{y},\mathbf{y}')\frac{1}{T}+\mathcal{O}\left(\frac{1}{T^2}\right)}\right]$$

$$=\lim_{T\to\infty}\sum_{\mathbf{y}'\neq\mathbf{y}}\left[\left[\frac{q_s(\mathbf{y}'|\mathbf{x})}{q_t(\mathbf{y}|\mathbf{x})}R_t(\mathbf{y},\mathbf{y}')+T\mathcal{O}\left(\frac{1}{T^2}\right)\right]\log\frac{\frac{q_s(\mathbf{y}'|\mathbf{x})}{q_t(\mathbf{y}|\mathbf{x})}R_t(\mathbf{y},\mathbf{y}')+T\mathcal{O}\left(\frac{1}{T^2}\right)}{\mathbf{s}_\theta(\mathbf{y})_{\mathbf{y}'}R_t(\mathbf{y},\mathbf{y}')+T\mathcal{O}\left(\frac{1}{T^2}\right)}\right]$$

$\because \lim_{T\to\infty}T\mathcal{O}\left(\frac{1}{T^2}\right)=0$, we get:

$$=\sum_{\mathbf{y}'\neq\mathbf{y}}\frac{q_s(\mathbf{y}'|\mathbf{x})}{q_t(\mathbf{y}|\mathbf{x})}R_t(\mathbf{y},\mathbf{y}')\log\frac{\frac{q_s(\mathbf{y}'|\mathbf{x})}{q_t(\mathbf{y}|\mathbf{x})}\cancel{R_t(\mathbf{y},\mathbf{y}')}}{\mathbf{s}_\theta(\mathbf{y})_{\mathbf{y}'}\cancel{R_t(\mathbf{y},\mathbf{y}')}}$$

$$= \sum_{\mathbf{y}' \neq \mathbf{y}} \frac{q_t(\mathbf{y}'|\mathbf{x})}{q_t(\mathbf{y}|\mathbf{x})} R_t(\mathbf{y},\mathbf{y}') \left( \log \frac{q_t(\mathbf{y}'|\mathbf{x})}{q_t(\mathbf{y}|\mathbf{x})} - \log \mathbf{s}_\theta(\mathbf{y})_{\mathbf{y}'} \right) \tag{64}$$

Finally, plugging (63) and (64) into (62) yields us the NELBO as proposed in Lou et al. [33]:

$$\mathbb{E}_{t \in [0,1], \mathbf{y} \sim q_t(.|\mathbf{x})} \left[ - \sum_{\mathbf{y}' \neq \mathbf{y}} R_t(\mathbf{y},\mathbf{y}') \left( \frac{q_t(\mathbf{y}'|\mathbf{x})}{q_t(\mathbf{y}|\mathbf{x})} - \mathbf{s}_\theta(\mathbf{y})_{\mathbf{y}'} \right) \right.$$

$$\left. + \sum_{\mathbf{y}' \neq \mathbf{y}} \frac{q_t(\mathbf{y}'|\mathbf{x})}{q_t(\mathbf{y}|\mathbf{x})} R_t(\mathbf{y},\mathbf{y}') \left( \log \frac{q_t(\mathbf{y}'|\mathbf{x})}{q_t(\mathbf{y}|\mathbf{x})} - \log \mathbf{s}_\theta(\mathbf{y})_{\mathbf{y}'} \right) \right]$$

$$= \mathbb{E}_{t \in [0,1], \mathbf{y} \sim q_t(.|\mathbf{x})} \left[ \sum_{\mathbf{y}' \neq \mathbf{y}} R_t(\mathbf{y},\mathbf{y}') \left( - \frac{q_t(\mathbf{y}'|\mathbf{x})}{q_t(\mathbf{y}|\mathbf{x})} + \mathbf{s}_\theta(\mathbf{y})_{\mathbf{y}'} \right. \right.$$

$$\left. \left. + \frac{q_t(\mathbf{y}'|\mathbf{x})}{q_t(\mathbf{y}|\mathbf{x})} \log \frac{q_t(\mathbf{y}'|\mathbf{x})}{q_t(\mathbf{y}|\mathbf{x})} - \frac{q_t(\mathbf{y}'|\mathbf{x})}{q_t(\mathbf{y}|\mathbf{x})} \log \mathbf{s}_\theta(\mathbf{y})_{\mathbf{y}'} \right) \right]$$

$$= \boxed{\mathbb{E}_{t \in [0,1], \mathbf{y} \sim q_t(.|\mathbf{x})} \left[ \sum_{\mathbf{y}' \neq \mathbf{y}} R_t(\mathbf{y},\mathbf{y}') \left( \mathbf{s}_\theta(\mathbf{y})_{\mathbf{y}'} - \frac{q_t(\mathbf{y}'|\mathbf{x})}{q_t(\mathbf{y}|\mathbf{x})} \log \mathbf{s}_\theta(\mathbf{y})_{\mathbf{y}'} + K \left( \frac{q_t(\mathbf{y}'|\mathbf{x})}{q_t(\mathbf{y}|\mathbf{x})} \right) \right) \right]} \tag{65}$$

where $K(a) = a \log a - a$. This concludes the proof.

## C.3   Concrete Score for MDLM

Given a latent variable $\mathbf{z}_t$ and the output of the denoising model, $\mathbf{x}_\theta(\mathbf{z}_t,t)$ parameterized using SUBS, we aim to recover the concrete score $\mathbf{s}_\theta(\mathbf{z}_t) \in (\mathbb{R}^+ + \{0\})^{|\mathcal{V}|}$. Note that $\mathbf{s}_\theta(\mathbf{z}_t)_{\mathbf{y}}$ is the ratio $\frac{p_t(\mathbf{y})}{p_t(\mathbf{z}_t)}$ in the reverse process. Since $\mathbf{x}_\theta$ approximates $\mathbf{x}$, we use $p_t(\mathbf{y}) = q_t(\mathbf{y}|\mathbf{x}_\theta(\mathbf{z}_t,t))$; therefore,

$$\mathbf{s}_\theta(\mathbf{z}_t)_{\mathbf{y}} = \frac{p_t(\mathbf{y})}{p_t(\mathbf{z}_t)} = \frac{q_t(\mathbf{y}|\mathbf{x}_\theta(\mathbf{z}_t,t))}{q_t(\mathbf{z}_t|\mathbf{x}_\theta(\mathbf{z}_t,t))}. \tag{66}$$

To obtain the score, we first compute $q_t(\mathbf{y}|\mathbf{x}_\theta(\mathbf{z}_t,t))$ for all possible $\mathbf{y}$ and $\mathbf{z}_t$. Using (4), we derive the following expressions under the SUBS parameterization:

$$q_t(\mathbf{y} \neq \mathbf{m}|\mathbf{x}_\theta(\mathbf{z}_t = \mathbf{m},t)) = \alpha_t \langle \mathbf{x}_\theta(\mathbf{z}_t,t),\mathbf{y} \rangle \tag{67}$$
$$q_t(\mathbf{y} \notin \{\mathbf{m},\mathbf{z}_t\}|\mathbf{x}_\theta(\mathbf{z}_t \neq \mathbf{m},t)) = 0 \tag{68}$$
$$q_t(\mathbf{y} = \mathbf{z}_t|\mathbf{x}_\theta(\mathbf{z}_t \neq \mathbf{m},t)) = \alpha_t \tag{69}$$
$$q_t(\mathbf{y} = \mathbf{m}|\mathbf{x}_\theta(\mathbf{z}_t \in \mathcal{V},t)) = 1 - \alpha_t \tag{70}$$

Plugging these into (66), we get:

$$\mathbf{s}_\theta(\mathbf{z}_t = \mathbf{m})_{\mathbf{y} \neq \mathbf{m}} = \frac{\alpha_t}{1 - \alpha_t} \langle \mathbf{x}_\theta(\mathbf{z}_t,t),\mathbf{y} \rangle \qquad \text{Using (70) and (67)} \tag{71}$$

$$\mathbf{s}_\theta(\mathbf{z}_t = \mathbf{m})_{\mathbf{y} = \mathbf{m}} = 1 \qquad \text{Using (70)} \tag{72}$$

$$\mathbf{s}_\theta(\mathbf{z}_t \neq \mathbf{m})_{\mathbf{y} = \mathbf{m}} = \frac{1 - \alpha_t}{\alpha_t} \qquad \text{Using (69) and (70)} \tag{73}$$

$$\mathbf{s}_\theta(\mathbf{z}_t \neq \mathbf{m})_{\mathbf{y} = \mathbf{z}_t} = 1 \qquad \text{Using (69)} \tag{74}$$

$$\mathbf{s}_\theta(\mathbf{z}_t \neq \mathbf{m})_{\mathbf{y} \notin \{\mathbf{m},\mathbf{z}_t\}} = 0 \qquad \text{Using (68) and (69)} \tag{75}$$

These can be consolidated into the following expression:

$$\boxed{\mathbf{s}_\theta(\mathbf{z}_t)_{\mathbf{y}} = \mathbf{y}^\top \left[ \delta_{\mathbf{z}_t,\mathbf{m}} \frac{\alpha_t}{1 - \alpha_t} \mathbf{x}_\theta(\mathbf{z}_t,t) + (1 - \delta_{\mathbf{z}_t,\mathbf{m}}) \frac{1 - \alpha_t}{\alpha_t} \mathbf{m} + \mathbf{z}_t \right]} \tag{76}$$

## C.4 Reverse Rate Matrix for MDLM

We can formulate the reverse rate matrix for MDLM using (76) and (56). Recall that the reverse rate matrix $\tilde{R}_t(\mathbf{y}',\mathbf{y})$ is given by:

$$\tilde{R}_t(\mathbf{y}',\mathbf{y}) = \begin{cases} \mathbf{s}_\theta(\mathbf{y})_{\mathbf{y}'} R_t(\mathbf{y},\mathbf{y}') & \mathbf{y}' \neq \mathbf{y} \\ -\sum_{\tilde{\mathbf{y}} \neq \mathbf{y}} \mathbf{s}_\theta(\mathbf{y})_{\mathbf{y}'} R_t(\mathbf{y},\tilde{\mathbf{y}}) & \mathbf{y}' = \mathbf{y}. \end{cases}$$

Let's examine the cases where $\mathbf{y} = \mathbf{m}$ and $\mathbf{y} \neq \mathbf{m}$.

**Case $\mathbf{y} = \mathbf{m}$:**   For $\mathbf{y}' \neq \mathbf{m}$, the reverse rate $\tilde{R}_t(\mathbf{y}',\mathbf{y}=\mathbf{m})$ is given by:

$$\tilde{R}_t(\mathbf{y}' \neq \mathbf{m},\mathbf{y}=\mathbf{m})$$
$$= \mathbf{s}_\theta(\mathbf{y}=\mathbf{m})_{\mathbf{y}' \neq \mathbf{m}} R_t(\mathbf{y}=\mathbf{m},\mathbf{y}')$$

Using (71) and (56), we get:

$$= \frac{\alpha_t}{1-\alpha_t} \langle \mathbf{x}_\theta(\mathbf{z}_t,t),\mathbf{y}' \rangle \left[ -\frac{\alpha_t'}{\alpha_t} \right]$$
$$= -\frac{\alpha_t'}{1-\alpha_t} \langle \mathbf{x}_\theta(\mathbf{z}_t,t),\mathbf{y}' \rangle \tag{77}$$

For $\mathbf{y}' = \mathbf{m}$, the reverse rate $\tilde{R}_t(\mathbf{y}',\mathbf{y}=\mathbf{m})$ is given by:

$$\tilde{R}_t(\mathbf{y}'=\mathbf{m},\mathbf{y}=\mathbf{m})$$
$$= -\sum_{\tilde{\mathbf{y}} \neq \mathbf{m}} \tilde{R}_t(\tilde{\mathbf{y}},\mathbf{y}=\mathbf{m})$$

Using (77), we get:

$$= \sum_{\tilde{\mathbf{y}} \neq \mathbf{m}} \frac{\alpha_t'}{1-\alpha_t} \langle \mathbf{x}_\theta(\mathbf{z}_t,t),\tilde{\mathbf{y}} \rangle$$
$$= \frac{\alpha_t'}{1-\alpha_t} \sum_{\tilde{\mathbf{y}} \neq \mathbf{m}} \langle \mathbf{x}_\theta(\mathbf{z}_t,t),\tilde{\mathbf{y}} \rangle$$

"zero-masking probability" in Sec. 3.2.3 $\implies \sum_{\tilde{\mathbf{y}} \neq \mathbf{m}} \langle \mathbf{x}_\theta(\mathbf{z}_t,t),\tilde{\mathbf{y}} \rangle = 1$; hence,

$$= \frac{\alpha_t'}{1-\alpha_t}. \tag{78}$$

**Case $\mathbf{y} \neq \mathbf{m}$:**   For $\mathbf{y}' \neq \mathbf{y}$ we have:

$$\tilde{R}_t(\mathbf{y}' \notin \{\mathbf{y},\mathbf{m}\},\mathbf{y} \neq \mathbf{m})$$
$$= \mathbf{s}_\theta(\mathbf{y} \neq \mathbf{m})_{\mathbf{y}' \notin \{\mathbf{y},\mathbf{m}\}} \underbrace{R_t(\mathbf{y} \neq \mathbf{m},\mathbf{y}' \notin \{\mathbf{y},\mathbf{m}\})}_{=0 \text{ from } (56)}$$
$$= 0 \tag{79}$$

For $\mathbf{y}' = \mathbf{m}$, we have:

$$\tilde{R}_t(\mathbf{y}'=\mathbf{m},\mathbf{y} \neq \mathbf{m})$$
$$= \mathbf{s}_\theta(\mathbf{y} \neq \mathbf{m})_{\mathbf{y}'=\mathbf{m}} \underbrace{R_t(\mathbf{y} \neq \mathbf{m},\mathbf{y}'=\mathbf{m})}_{=0 \text{ from } (56)}$$
$$= 0 \tag{80}$$

Thus, for $\mathbf{y}' = \mathbf{y}$, we have:

$$\tilde{R}_t(\mathbf{y}'=\mathbf{y},\mathbf{y} \neq \mathbf{m})$$
$$= -\sum_{\tilde{\mathbf{y}} \neq \mathbf{y}} \tilde{R}_t(\tilde{\mathbf{y}},\mathbf{y} \neq \mathbf{m})$$

$$= -\underbrace{\tilde{R}_t(\tilde{\mathbf{y}}=\mathbf{m},\mathbf{y}\neq\mathbf{m})}_{=0 \text{ from } (80)} - \underbrace{\sum_{\tilde{\mathbf{y}}\notin\{\mathbf{y},\mathbf{m}\}}\tilde{R}_t(\tilde{\mathbf{y}},\mathbf{y}\neq\mathbf{m})}_{=0 \text{ from } (79)}$$

$$= 0 \tag{81}$$

Summarizing (77), (78), (79), (80), (81), we have:

$$\tilde{R}_t(\mathbf{y}',\mathbf{y}) = \begin{cases} -\langle\mathbf{x}_\theta(\mathbf{y},t),\mathbf{y}'\rangle\frac{\alpha_t'}{1-\alpha_t} & \mathbf{y}'\neq\mathbf{m},\mathbf{y}=\mathbf{m} \\ \frac{\alpha_t'}{1-\alpha_t} & \mathbf{y}'=\mathbf{m},\mathbf{y}=\mathbf{m} \\ 0 & \text{otherwise.} \end{cases}$$

$$= \boxed{-\frac{\alpha_t'}{1-\alpha_t}[\mathbf{y}']^\top[\mathbf{x}_\theta(\mathbf{y},t)-\mathbf{m}]\langle\mathbf{y},\mathbf{m}\rangle} \tag{82}$$

## C.5 Deriving MDLM's NELBO via CTMC

Now, we aim to show that substituting the expression for the rate matrix $R_t$ in terms of state transition matrix $Q_t$ from (56) into (65) and switching from score-parameterization to the SUBS parameterization (Sec. 3.2.3) yields the simplified NELBO for MDLM as given by (10). We present the proof below. Recall that the term $\langle\mathbf{a},\mathbf{b}\rangle$ denotes the dot product of two vectors $\mathbf{a}$ and $\mathbf{b}$. When $\mathbf{a}$ and $\mathbf{b}$ represent two *one-hot* vectors, this quantity evaluates to $1$ if $\mathbf{a}=\mathbf{b}$ and $0$ otherwise.

**Proof.** Recall that for absorbing state diffusion, $\mathbf{y}$ takes only two possible values, i.e., $\mathbf{y}\in\{\mathbf{x},\mathbf{m}\}$. Thus, we expand (65) as follows:

$$\mathbb{E}_{t\in[0,1],\mathbf{y}\sim q_t(.|\mathbf{x})}\left[\langle\mathbf{y},\mathbf{x}\rangle\left[\sum_{\mathbf{y}'\neq\mathbf{x}}R_t(\mathbf{x},\mathbf{y}')\left(\mathbf{s}_\theta(\mathbf{x})_{\mathbf{y}'}-\frac{q_t(\mathbf{y}'|\mathbf{x})}{q_t(\mathbf{x}|\mathbf{x})}\log\mathbf{s}_\theta(\mathbf{x})_{\mathbf{y}'}+K\left(\frac{q_t(\mathbf{y}'|\mathbf{x})}{q_t(\mathbf{x}|\mathbf{x})}\right)\right)\right]\right.$$

$$\left.+\langle\mathbf{y},\mathbf{m}\rangle\left[\sum_{\mathbf{y}'\neq\mathbf{m}}R_t(\mathbf{m},\mathbf{y}')\left(\mathbf{s}_\theta(\mathbf{m})_{\mathbf{y}'}-\frac{q_t(\mathbf{y}'|\mathbf{x})}{q_t(\mathbf{m}|\mathbf{x})}\log\mathbf{s}_\theta(\mathbf{m})_{\mathbf{y}'}+K\left(\frac{q_t(\mathbf{y}'|\mathbf{x})}{q_t(\mathbf{m}|\mathbf{x})}\right)\right)\right]\right]$$

$\because R_t(\mathbf{x},\mathbf{y}'\neq\mathbf{x})=0$ from (54), we get:

$$=\mathbb{E}_{t\in[0,1],\mathbf{y}\sim q_t(.|\mathbf{x})}\langle\mathbf{y},\mathbf{m}\rangle\left[\sum_{\mathbf{y}'\neq\mathbf{m}}R_t(\mathbf{m},\mathbf{y}')\left(\mathbf{s}_\theta(\mathbf{m})_{\mathbf{y}'}-\frac{q_t(\mathbf{y}'|\mathbf{x})}{q_t(\mathbf{m}|\mathbf{x})}\log\mathbf{s}_\theta(\mathbf{m})_{\mathbf{y}'}+K\left(\frac{q_t(\mathbf{y}'|\mathbf{x})}{q_t(\mathbf{m}|\mathbf{x})}\right)\right)\right]$$

Substituting $R_t(\mathbf{m},\mathbf{y}'\neq\mathbf{m})$ from (52), we get:

$$=\mathbb{E}_{t\in[0,1],\mathbf{y}\sim q_t(.|\mathbf{x})}\langle\mathbf{y},\mathbf{m}\rangle\left[\sum_{\mathbf{y}'\neq\mathbf{m}}-\frac{\alpha_t'}{\alpha_t}\left(\mathbf{s}_\theta(\mathbf{m})_{\mathbf{y}'}-\frac{q_t(\mathbf{y}'|\mathbf{x})}{q_t(\mathbf{m}|\mathbf{x})}\log\mathbf{s}_\theta(\mathbf{m})_{\mathbf{y}'}+K\left(\frac{q_t(\mathbf{y}'|\mathbf{x})}{q_t(\mathbf{m}|\mathbf{x})}\right)\right)\right]$$

$$=\mathbb{E}_{t\in[0,1],\mathbf{y}\sim q_t(.|\mathbf{x})}\langle\mathbf{y},\mathbf{m}\rangle\left[-\frac{\alpha_t'}{\alpha_t}\left(\underbrace{\sum_{\mathbf{y}'\neq\mathbf{m}}\mathbf{s}_\theta(\mathbf{m})_{\mathbf{y}'}}_{\textbf{Term 1}}-\underbrace{\sum_{\mathbf{y}'\neq\mathbf{m}}\frac{q_t(\mathbf{y}'|\mathbf{x})}{q_t(\mathbf{m}|\mathbf{x})}\log\mathbf{s}_\theta(\mathbf{m})_{\mathbf{y}'}}_{\textbf{Term 2}}+\underbrace{\sum_{\mathbf{y}'\neq\mathbf{m}}K\left(\frac{q_t(\mathbf{y}'|\mathbf{x})}{q_t(\mathbf{m}|\mathbf{x})}\right)}_{\textbf{Term 3}}\right)\right]$$

$$\tag{83}$$

**Term 1:**

$$\sum_{\mathbf{y}\neq\mathbf{m}}\mathbf{s}_\theta(\mathbf{m})_{\mathbf{y}}$$

Using (67), we get,

$$=\sum_{\mathbf{y}\neq\mathbf{m}}\frac{\alpha_t}{1-\alpha_t}\langle\mathbf{x}_\theta(\mathbf{m},t),\mathbf{y}\rangle$$

$$= \frac{\alpha_t}{1-\alpha_t} \sum_{\mathbf{y} \neq \mathbf{m}} \langle \mathbf{x}_\theta(\mathbf{m},t), \mathbf{y} \rangle$$

"zero-masking probability" in Sec. 3.2.3 $\implies \sum_{\mathbf{y} \neq \mathbf{m}} \langle \mathbf{x}_\theta(\mathbf{m},t), \mathbf{y} \rangle = 1$; hence,

$$= \frac{\alpha_t}{1-\alpha_t} \tag{84}$$

**Term 2:**

$$\sum_{\mathbf{y}' \neq \mathbf{m}} \frac{q_t(\mathbf{y}'|\mathbf{x})}{q_t(\mathbf{m}|\mathbf{x})} \log \mathbf{s}_\theta(\mathbf{m})_{\mathbf{y}'}$$

$$= \frac{q_t(\mathbf{x}|\mathbf{x})}{q_t(\mathbf{m}|\mathbf{x})} \log \mathbf{s}_\theta(\mathbf{m})_{\mathbf{x}} + \sum_{\mathbf{y}' \notin \{\mathbf{m},\mathbf{x}\}} \frac{q_t(\mathbf{y}'|\mathbf{x})}{q_t(\mathbf{m}|\mathbf{x})} \log \mathbf{s}_\theta(\mathbf{m})_{\mathbf{y}'}$$

$\because q_t(\mathbf{y}'|\mathbf{x}) = 0$ for $\mathbf{y}' \notin \{\mathbf{x},\mathbf{m}\}$ from (4)) we get:

$$= \frac{q_t(\mathbf{x}|\mathbf{x})}{q_t(\mathbf{m}|\mathbf{x})} \log \mathbf{s}_\theta(\mathbf{m})_{\mathbf{x}}$$

Using (4), we get:

$$= \frac{\alpha_t}{1-\alpha_t} \log \mathbf{s}_\theta(\mathbf{m})_{\mathbf{x}}$$

Using (67), we get:

$$= \frac{\alpha_t}{1-\alpha_t} \log \left[ \frac{\alpha_t}{1-\alpha_t} \langle \mathbf{x}_\theta(\mathbf{m},t), \mathbf{x} \rangle \right]$$

$$= \frac{\alpha_t}{1-\alpha_t} \log \frac{\alpha_t}{1-\alpha_t} + \frac{\alpha_t}{1-\alpha_t} \log \langle \mathbf{x}_\theta(\mathbf{m},t), \mathbf{x} \rangle \tag{85}$$

**Term 3:**

$$\sum_{\mathbf{y}' \neq \mathbf{m}} K\left( \frac{q_t(\mathbf{y}'|\mathbf{x})}{q_t(\mathbf{m}|\mathbf{x})} \right)$$

$$= \sum_{\mathbf{y}' \neq \mathbf{m}} \left[ \frac{q_t(\mathbf{y}'|\mathbf{x})}{q_t(\mathbf{m}|\mathbf{x})} \log \frac{q_t(\mathbf{y}'|\mathbf{x})}{q_t(\mathbf{m}|\mathbf{x})} - \frac{q_t(\mathbf{y}'|\mathbf{x})}{q_t(\mathbf{m}|\mathbf{x})} \right]$$

$$= \frac{q_t(\mathbf{x}|\mathbf{x})}{q_t(\mathbf{m}|\mathbf{x})} \log \frac{q_t(\mathbf{x}|\mathbf{x})}{q_t(\mathbf{m}|\mathbf{x})} - \frac{q_t(\mathbf{x}|\mathbf{x})}{q_t(\mathbf{m}|\mathbf{x})} + \sum_{\mathbf{y}' \notin \{\mathbf{x},\mathbf{m}\}} \left[ \frac{q_t(\mathbf{y}'|\mathbf{x})}{q_t(\mathbf{m}|\mathbf{x})} \log \frac{q_t(\mathbf{y}'|\mathbf{x})}{q_t(\mathbf{m}|\mathbf{x})} - \frac{q_t(\mathbf{y}'|\mathbf{x})}{q_t(\mathbf{m}|\mathbf{x})} \right]$$

$\because q_t(\mathbf{y}'|\mathbf{x}) = 0$ for $\mathbf{y}' \notin \{\mathbf{x},\mathbf{m}\}$, we get,

$$= \frac{q_t(\mathbf{x}|\mathbf{x})}{q_t(\mathbf{m}|\mathbf{x})} \log \frac{q_t(\mathbf{x}|\mathbf{x})}{q_t(\mathbf{m}|\mathbf{x})} - \frac{q_t(\mathbf{x}|\mathbf{x})}{q_t(\mathbf{m}|\mathbf{x})}$$

Substituting the values using (4), we get:

$$= \frac{\alpha_t}{1-\alpha_t} \log \frac{\alpha_t}{1-\alpha_t} - \frac{\alpha_t}{1-\alpha_t} \tag{86}$$

Substituing (84), (85), and (86) in (83) we get,

$$\mathbb{E}_{t \in [0,1], \mathbf{y} \sim q_t(.|\mathbf{x})} \langle \mathbf{y}, \mathbf{m} \rangle \left[ -\frac{\alpha_t'}{\alpha_t} \left( \frac{\alpha_t}{1-\alpha_t} - \frac{\alpha_t}{1-\alpha_t} \log \frac{\alpha_t}{1-\alpha_t} - \frac{\alpha_t}{1-\alpha_t} \log \langle \mathbf{x}_\theta(\mathbf{m},t), \mathbf{x} \rangle \right. \right.$$

$$\left. \left. + \frac{\alpha_t}{1-\alpha_t} \log \frac{\alpha_t}{1-\alpha_t} - \frac{\alpha_t}{1-\alpha_t} \right) \right]$$

$$= \mathbb{E}_{t \in [0,1], \mathbf{y} \sim q_t(.|\mathbf{x})} \langle \mathbf{y}, \mathbf{m} \rangle \left[ -\frac{\alpha_t'}{\alpha_t} \left( -\frac{\alpha_t}{1-\alpha_t} \log \langle \mathbf{x}_\theta(\mathbf{m},t), \mathbf{x} \rangle \right) \right]$$

$$= \mathbb{E}_{t \in [0,1], \mathbf{y} \sim q_t(.|\mathbf{x})} \langle \mathbf{y}, \mathbf{m} \rangle \left[ \frac{\alpha_t'}{1-\alpha_t} \log \langle \mathbf{x}_\theta(\mathbf{m},t), \mathbf{x} \rangle \right]$$

Under the SUBS parameterization, $\log\langle\mathbf{x}_\theta(\mathbf{z}_t,t),\mathbf{x}\rangle=0$ when $\mathbf{z}_t=\mathbf{x}$; hence $\langle\mathbf{y},\mathbf{m}\rangle$ can be dropped:

$$= \boxed{\mathbb{E}_{t\in[0,1],\mathbf{y}\sim q_t(.|\mathbf{x})}\left[\frac{\alpha'_t}{1-\alpha_t}\log\langle\mathbf{x}_\theta(\mathbf{z}_t,t),\mathbf{x}\rangle\right]} \tag{87}$$

This concludes the proof.

## Appendix D   Experimental details

### D.1   Likelihood Evaluation

We use a single monte-carlo estimate for $t$ to evaluate the likelihood. The low discrepancy sampler (D.3) plays a key role in reducing the variance of the estimate as seen in Table 8.

### D.2   Avg. Number of Tokens seen

Given `training_steps`, `batch_size`, `context_length`, the number of tokens seen by the AR model is given as:

$$\texttt{training\_steps}\times\texttt{batch\_size}\times\texttt{context\_length}. \tag{88}$$

However, this expression doesn't hold for a diffusion model, since at each training step, a fraction of the input tokens are masked before being fed to the model. Let $p_m$ be the probability of a token being masked at a timestep $t$. For the log-linear schedule in our experiments, $p_m=t$. Thus, the expected number of tokens seen by the diffusion model is:

$$
\begin{aligned}
&\mathbb{E}_{t\sim\mathcal{U}[0,1]}[\texttt{training\_steps}\times\texttt{batch\_size}\times\texttt{context\_length}\times p_m]\\
&=\texttt{training\_steps}\times\texttt{batch\_size}\times\texttt{context\_length}\times\mathbb{E}_{t\sim\mathcal{U}[0,1]}[p_m]\\
&=\texttt{training\_steps}\times\texttt{batch\_size}\times\texttt{context\_length}\times\mathbb{E}_{t\sim\mathcal{U}[0,1]}[t] &&\because p_m=t\\
&=\texttt{training\_steps}\times\texttt{batch\_size}\times\texttt{context\_length}\times 0.5. &&\because\mathbb{E}_{t\sim\mathcal{U}[0,1]}[t]=0.5
\end{aligned}
\tag{89}
$$

**LM1B.**  Following [1, 33, 26], we train MDLM for 1M training steps with a `batch_size` = 512, and a context length of 128. Like [33] we use a log-linear schedule and hence the number of tokens seen by our model is $\approx$ 33B (89). Similarly, MDLM trained for 10M steps, saw 327B tokens in expectation. The corresponding AR baseline was trained for 0.5M and 5M steps to ensure a similar number of tokens was seen.

**OWT.**  We train SEDD and MDLM for 1M training steps with a `batch_size` = 512, `context_length` = 1024, and log-linear schedule. Hence, these models saw 262B tokens during training. Similarly, the AR model saw the same number of tokens when trained for 0.5M steps with the same `batch_size` and `context_length`.

### D.3   Low discrepancy sampler

To reduce variance during training we use a low-discrepancy sampler, similar to that proposed in Kingma et al. [29]. Specifically, when processing a minibatch of $N$ samples, instead of independently sampling $N$ from a uniform distribution, we partition the unit interval and sample the time step for each sequence $i\in\{1,...,N\}$ from a different portion of the interval $t_i\sim U[\frac{i-1}{N},\frac{i}{N}]$. This ensures that our sampled timesteps are more evenly spaced across the interval $[0,1]$, reducing the variance of the ELBO.

### D.4   Language Modeling

For our forward noise process, we use a log-linear noise schedule similar to Lou et al. [33].

We detokenize the One Billion Words dataset following Lou et al. [33], whose code can be found here[4]. We tokenize the One Billion Words dataset with the `bert-base-uncased` tokenizer, following He et al. [26]. We pad and truncate sequences to a length of 128.

---

[4]https://github.com/louaaron/Score-Entropy-Discrete-Diffusion/blob/main/data.py

We tokenize OpenWebText with the `GPT2` tokenizer. We do not pad or truncate sequences – we concatenate and wrap them to a length of 1,024. When wrapping, we add the `eos` token in-between concatenated. We additionally set the first and last token of every batch to be `eos`. Since OpenWebText does not have a validation split, we leave the last 100k docs as validation.

We parameterize our autoregressive baselines, SEDD, and MDLM with the transformer architecture from Lou et al. [33]. We use 12 layers, a hidden dimension of 768, 12 attention heads, and a timestep embedding of 128 when applicable. Word embeddings are not tied between the input and output.

We use the AdamW optimizer with a batch size of 512, constant learning rate warmup from 0 to a learning rate of 3e-4 for 2,500 steps. We use a constant learning rate for 1M, 5M, or 10M steps on One Billion Words, and 1M steps for OpenWebText. We use a dropout rate of 0.1.

### D.5 Zeroshot Likelihood

We evaluate zeroshot likelihoods by taking the models trained on OpenWebText and evaluating likelihoods on the validation splits of 7 datasets: Penn Tree Bank (PTB; Marcus et al. [36]), Wikitext [38], One Billion Word Language Model Benchmark (LM1B; Chelba et al. [8]), Lambada [41], AG News [68], and Scientific Papers (Pubmed and Arxiv subsets; Cohan et al. [10]). We detokenize the datasets following Lou et al. [33]. For the AG News and Scientific Papers (Pubmed and Arxiv), we apply both the Wikitext and One Billion Words detokenizers. Since the zeroshot datasets have different conventions for sequence segmentation, we wrap sequences to 1024 and do not add `eos` tokens in between sequences.

### D.6 Representation Learning

Following Devlin et al. [15], we evaluate on all GLUE tasks [65], but exclude WNLI.

We pre-train a MosaicBERT model on C4 [46] for 70k steps, corresponding to 36B tokens. We pad and truncate the data to 128 tokens using the `bert-base-uncased` tokenizer.

MosaicBERT [43] has a similar architecture to `bert-base-uncased` and has 137M parameters, 12 layers, 12 attention heads, a hidden dimension of 768, an intermediate size of 3072, and ALiBi attention bias [44].

For pre-training, we use the following hyperparameters: A global batch size of 4096 with gradient accumulation, a learning rate of 5e-4, linear decay to 0.02x of the learning rate with a warmup of 0.06x of the full training duration, and the decoupled AdamW optimizer with 1e-5 weight decay and betas 0.9 and 0.98.

For diffusion fine-tuning we use AdamW with a warmup of 2,500 steps from a learning rate of 0 to 5e-5, betas 0.95 and 0.999, and batch size 512. We train for 5k steps total, corresponding to 32M tokens.

For GLUE evaluation, we use the HuggingFace script found here[5]. We use the default parameters for all datasets, except for a batch size of 16, which we found helped with smaller datasets. This includes the default of 3 epochs for all datasets and learning rate of 2e-5.

### D.7 Diffusion DNA Models

**Dataset** We pre-train the Caduceus MLM [50] on the HG38 human reference genome [11]. Following Schiff et al. [50], we use character- / base pair-level tokenization. The dataset is based on the splits used in Avsec et al. [3]: the training split comprises of 35 billion tokens covering the human genome. This consists of 34,021 segments extended to a maximum length of 1,048,576 (220 segments). We maintain a constant $2^{20}$ tokens per batch. For the Genomics Benchmark tasks, we use 5-fold cross-validation where we split the training set into 90/10 train/validation splits.

**Architecture** The Caduceus MLM uses as a backbone a bi-directional variant of the data-dependent SSM Mamba block proposed in Gu et al. [22]. This architecture is ideal as it contains inductive biases that preserve reverse complement (RC) equviariance, respecting the inherent symmetry of double-stranded DNA molecules [35, 50, 69].

---

[5]https://github.com/huggingface/transformers/tree/main/examples/pytorch/text-classification

**Training details**  All models are pre-trained on 10B tokens (10K steps) and fine-tuned on a generative objective for an additional 50B tokens (50K steps). We use a global batch size of 1024 for a context length of 1024 tokens. Downstream task fine-tuning is performed for 16K steps ( 1B tokens).

For performing Caduceus MLM pre-training, we follow Schiff et al. [50] for the model size configuration, and hyperparameter selection. For pre-training, we use a fixed 15% mask rate as done in Devlin et al. [15]. Of the 'masked' tokens, 80% are replaced with [MASK] , 10% are replaced with a random token from the vocabulary, and 10% are left unchanged.

For fine-tuning all Mamba-based models (including Caduceus) on diffusion objectives, we lower the learning rate from 8e-3 to 1e-3. For fine-tuning HyenaDNA [39], we lower the learning rate from 6e-4 to 5e-5. Similar to Gu et al. [22], Schiff et al. [50], we found that Mamba-based models were robust to higher learning rates. We exclude timestep embeddings for all Diffusion DNA experiments, as we show it has minimal impact on generative performance (see Table 12, Suppl. E.5).

We perform downstream task fine-tuning on the final hidden state embedding from pre-training. We perform mean pooling across the sequence length, which may vary from 200 to approximately 2,000 bps. We report the mean and $\pm$ on max/min classification accuracy over 5-fold cross-validation (CV) using different random seeds, with early stopping on validation accuracy. For each task, we do a hyperparameter sweep over batch size and learning rate and report the values of the 5-fold CV for the best configuration.

**Genomic Benchmark Task Distributions**  We use a subset of the Genomic Benchmark tasks with an emphasis on tasks from Human data. The positive samples for each dataset were generated by selecting samples that were annotated, either computationally or experimentally, in previous work (e.g enhancers, promoters, open chromatin regions (OCR)) [20]. These annotations each correspond to subsets of the genome of varying sizes that may exhibit different distributions of DNA than those observed globally over the reference genome. Due to this, the observed dataset may have a different distribution than the data used for pre-training and calculating perplexity. This might in turn lead to a case where perplexity and downstream performance may not necessarily correlate.

## Appendix E  Additional Experiments

### E.1  Noise schedule parameterization

As described in Sec. 3.4, the ELBO is invariant to the functional form of $\alpha_t$. To demonstrate this, we evaluate MDLM, initially trained using a log-linear schedule on OWT, by replacing the noise schedule with various other noise schedules as mentioned below. Following prior works [1, 33, 54], we parameterize $\alpha_t = e^{-\sigma(t)}$, where $\sigma(t):[0,1] \to \mathbb{R}^+$. Various functional forms of $\sigma(t)$ are listed below:

**Log Linear [1, 33, 54].** The log linear schedule is given as:

$$\sigma(t) = -\log(1-t) \tag{90}$$

**Cosine Squared schedule [24].** The Cosine Squared schedule is given as:

$$\sigma(t) = -\log \cos^2\left(\frac{\pi}{2}(1-t)\right) \tag{91}$$

**Cosine schedule.** The Cosine schedule is given as:

$$\sigma(t) = -\log \cos\left(\frac{\pi}{2}(1-t)\right) \tag{92}$$

**Linear.** The Linear schedule is given as:

$$\sigma(t) = \sigma_{\max} t \tag{93}$$

where $\sigma_{\max}$ is a very large number. In our experiments we set it to $10^8$.

### E.1.1  ELBO Invariance

The function $\alpha_t$ is invertible due to the monotonicity assumption in Sec. 3.1, and so we can perform the following change of variables in (10): $\gamma \equiv \log(1-\alpha_t)$. Let $f : [0,1] \to \mathbb{R}^-$ be a function such

that $\gamma = f(t)$. Note that $\alpha_t$ goes through a monotonic transformation to obtain $\gamma$; hence, $\gamma$ is also monotonic in $t$ since $\alpha_t$ is monotonic in $t$. This implies that the function $f$ is invertible. Let $t = f^{-1}(\gamma)$. Then, we can we have the following diffusion loss:

$$
\begin{aligned}
\mathcal{L}^{\infty}_{\text{NELBO}} &= \mathbb{E}_q \int_{t=0}^{t=1} \frac{\alpha_t'}{1-\alpha_t} \log\langle \mathbf{x}_\theta(\mathbf{z}_t,t), \mathbf{x}\rangle \mathrm{d}t \\
&= -\mathbb{E}_q \int_{t=0}^{t=1} \log\langle \mathbf{x}_\theta(\mathbf{z}_t,t), \mathbf{x}\rangle \frac{\mathrm{d}}{\mathrm{d}t}[\log(1-\alpha_t)]\mathrm{d}t \\
&= -\mathbb{E}_q \int_{t=0}^{t=1} \log\langle \mathbf{x}_\theta(\mathbf{z}_t,t), \mathbf{x}\rangle \frac{\mathrm{d}}{\mathrm{d}t}[f(t)]\mathrm{d}t && \text{Substituting } f(t) = \log(1-\alpha_t) \\
&= -\mathbb{E}_q \int_{\gamma=-\infty}^{\gamma=0} \log\langle \mathbf{x}_\theta(\mathbf{z}_{f^{-1}(\gamma)}, f^{-1}(\gamma)), \mathbf{x}\rangle \mathrm{d}\gamma && \text{Change of variables } \gamma \equiv f(t) \\
&= -\mathbb{E}_q \int_{\gamma=-\infty}^{\gamma=0} \log\langle \mathbf{x}_\theta(\tilde{\mathbf{z}}_\gamma, f^{-1}(\gamma)), \mathbf{x}\rangle \mathrm{d}\gamma && \tilde{\mathbf{z}}_\gamma \equiv \mathbf{z}_{f^{-1}(\gamma)} \\
&= -\mathbb{E}_q \int_{\gamma=-\infty}^{\gamma=0} \log\langle \tilde{\mathbf{x}}_\theta(\tilde{\mathbf{z}}_\gamma, \gamma), \mathbf{x}\rangle \mathrm{d}\gamma && \tilde{\mathbf{x}}_\theta(\tilde{\mathbf{z}}_\gamma, \gamma) \equiv \mathbf{x}_\theta(\tilde{\mathbf{z}}_\gamma, f^{-1}(\gamma)) \quad (94)
\end{aligned}
$$

This new formulation demonstrates that the diffusion loss is invariant to the functional form of $\alpha_t$. In Table 9, we demonstrate empirically that noise schedules with different functional forms evaluate to the same Likelihood which is consistent with our theory. However, different schedules lead to different per data point variance. Notably, the log-linear schedule exhibits the lowest variance among all the noise schedules considered.

Table 9: Likelihood in bits per dimension (BPD) for different noise schedules on OWT dataset, is reported along with the mean and variance associated with each noise schedule per data point. We empirically observe that noise schedules with different functional forms yield the same likelihood, consistent with our theory in Sec. 3.4; however, different schedules result in different variances.

| $\sigma(t)$ | Mean | Variance per datapoint |
|---|---|---|
| Log Linear (90) | 3.30 | 1.81 |
| Cosine (92) | 3.30 | 3.30 |
| Cosine Squared (91) | 3.30 | 3.30 |
| Linear (93) | 3.30 | 7.57 |

## E.2 Faster sampling with caching

In Figure 10, we compare the wall clock times of variaous methods: AR, SEDD, MDLM with caching, and MDLM without caching for generating 64 samples on a single GPU. When sampling in batches, a change of 1 token would necessitate a call to the denoising model. Therefore, smaller batch sizes have a lower likelihood of a token being unmasked. This might lead one to prefer generating samples in smaller batches, as opposed to using a larger batch size that fully saturates the GPU. Table 10 shows that generating samples with a batch size of 1 and using caching is twice as fast as generating samples without caching while fully utilizing the GPU. In Fig. 2, we observe that MDLM without caching yields samples that consistently get better generative perplexity than SEDD. For $T = \{5k, 10k\}$, both SEDD and MDLM get better generative perplexity than the AR model.

Table 10: Wall clock time reported in minutes to generate 64 samples on a single A5000 GPU.

| | $T=5k(\downarrow)$ | $T=10k(\downarrow)$ |
|---|---|---|
| SEDD | 85.3 | 155.2 |
| MDLM | 70.3 | 127.9 |
| + caching | **40.1** | **60.4** |

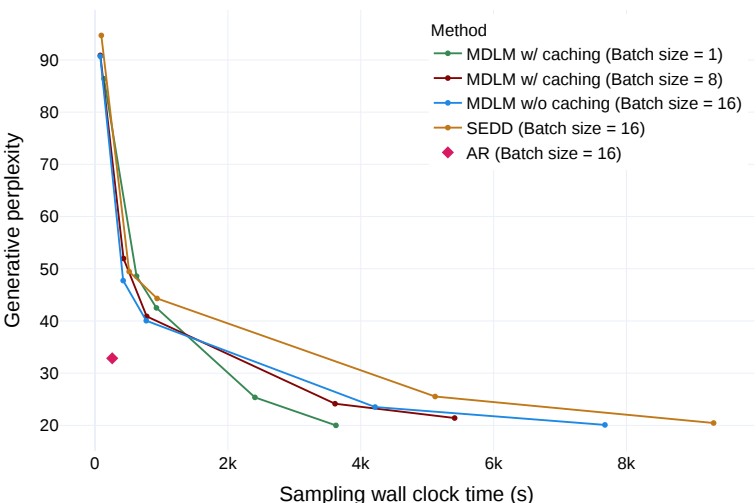

Figure 2: Generative perplexities across wall clock time for generating 64 samples on OWT using a single 32GB A5000 GPU are compared by varying $T \in \{100, 500, 1000, 5000, 10000\}$ in the reverse diffusion process. The samples are generated in mini-batches with a batch size of 16 for AR, SEDD, and MDLM without caching, as it is the largest batch size that fits on this GPU. For MDLM with caching, we vary the batch size.

### E.3 LM1B ablations

We assess the importance of our continuous-time framework by performing ablation on diffusion steps $T$. In Table 11, we compare NLL and PPL under continuous and discrete T in MDLM. We find that NLL consistently decreases as $T \to \infty$.

Table 11: Discrete vs continuous time evaluation for MDLM w/o time-conditioning on OWT. MDLM was trained with $T = \infty$. We report test perplexity for a discrete $T$.

| T | PPL($\leq$) |
|---|---|
| $\infty$ | **23.05** |
| 10 | 42.18 |
| 20 | 30.70 |
| 50 | 25.77 |
| 100 | 24.35 |
| 200 | 23.66 |
| 500 | 23.26 |
| 1000 | 23.15 |

### E.4 Train NLL curves on OWT

In Figure 3, we show that MDLM achieves lower variance loss during training compared to a previous diffusion language model, SEDD. Training is performed over 1M steps on OWT (which corresponds to 524B tokens).

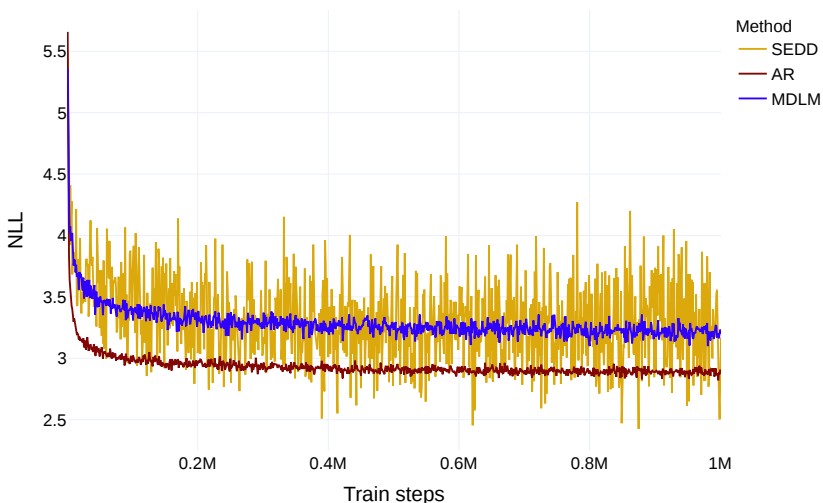

Train Negative Log-Likelihood (NLL) on OpenWebText

Figure 3: Train negative log-likelihood (NLL) curves across 1M gradient steps (524B tokens) on OpenWebText [18]. NLL is logged every 1K steps without value smoothing.

### E.5 Time-conditioning ablation on OWT

In Table 12, we assess the importance of time conditioning in MDLM on OWT. We observe that time-conditioning has minimal impact on perplexity. Training is performed over 1M steps on OWT (which corresponds to 524B tokens).

Table 12: Ablation on time-conditioning in MDLM on OWT.

| Method | PPL |
| --- | --- |
| MDLM w/ time-conditioning | 23.21 |
| MDLM w/o time-conditioning | **23.05** |

### E.6 Unconditional Samples

Here, we present some unconditional samples generated by MDLM trained on OWT with a context length of $L = 1024$ for $T = \{1000, 10000\}$.

### E.6.1 T = 1000

**Example 1**  `<|endoftext|> a 17-10 victory and a trip to the playoffs.`

`The last wildcard seed:  New York Jets, Houston and the last potable playoff spot.  The last-second home wins:  New Orleans and Carolina, 21-21.`

`The Saints finish sixth with the highest regular season (42) NFC wins.  They lost 14 of their 13 games in the conference playoffs.`

`The Cardinals were in Group A in Round 1 with Game 2, Round 3, Round 4 and Quarter Game 5, but they made their last trip to the playoffs off North Carolina on the road as even North Carolina.`

`True to their reputation, the Cards swept the Saints in the first round, but knocked it out at home.  No Panthers went to the playoffs more than the Saints.`

Don Jean no longer is the South Carolina Panther.

The Cardinals thought that provided that he had a chance to be an NFL player.

"I did," said defensive end Lorenzo Williams with a laugh as he exited his car at the airport.  "Also, I won that game."

KC win brings Carson back home.

Griffin made promise on Sunday to never exactly give up the dunk.  Although he failed to score 40 points in the playoffs, he has had better luck in them this year.

With turnovers and fumbles returning, he has to play out because the team doesn't trust him.  He's long years of injuries, turnovers and calls because he knows he can play that way for everybody.

Griffin is no stranger to Saints fans.

"Players want him to know them," someone from South Carolina said after coming out against the Panthers – in their best home home Week 7 win Sunday – in an 11-9 rout.  South Carolina did win its final three and passed the Saints, 24-1.

Although the Cardinals are in the South, they am a step behind.

They still have little time left to take down the West Coast wild card.  There is no chance they get another victory.

The West was out by Beshear in their first round games last season, losing by 63 to the 49ers.

The outcome will be tough.

"Now we're so close, let's figure out the time to win," Brees said.  "We still have a few games left; I'm glad about that."

South Carolina takes the revenge.

When asked about his second time since Super 4, Brees shot back that he understood.

"You can doubt the answer but I think that was a no-brainer.  In time, you try to prove an answer wrong," Brees said.  "I think his ability will be as cool as Julio Jones' ability, but having that time [out] to my season was difficult overall.  I did what I was expecting to do.  Hopefully they'll tell me to try again."

After their late win, the NFL calls the Saints reschedule'must try.'

"Because Saints," those who am there still say it, "focus on defense and, offense is defense."

ESPN said Saints's star receiver Dashon Jeffishard, turning heads on long passes and connecting with open defenders, already had their 20-yard overall score from the field.  When Jeffishard finished with three passing he set up.  He obviously had no difference; it was his first snap-off.

With his changes in his starting lineup, Brees was just hoping they had little to prove against Carolina over the weekend.

"I felt like we didn't have quite enough focus on and there was so puny coverage that I wanted more of our guys at the same position so we'd up our game," he said.

Brees said South Carolina was well at linebacker.

"If that's part of it, if you're going to try and stay with what you're going [out with].  What'd you want?  Smart play," Brees said.  "What would you say?  That you're always ready to play.  So you're going out there strong and ready to go to play football."

That said, New Orleans was damned shy when it came to Carolina.

"My guys admit to feeling it a little bit [Sunday but] I say to them that they always knew, 'I don't think that was necessarily how I would beat you, that will give them their confidence," Brees said.

"It was really hard because I've obviously learned a lot of detail about how to deal with everyone and as hard as I have to be, I also feel part of the stuff that they've been through on the team, like they're still going to go through things they know are somewhat right, but they feel a lot of pressure so it's got to be important to get it right now to get it in the future."

Could all ask for more roses?

Let's just take a slip, South Carolina, and face the NFC<|endoftext|>

**Example 2**   <|endoftext|> Memorial Hospital.

Valia and Hill had been working with the Coast Guard in response to public questions, and when they were reached couldn't comment on the new information, Chapman said.

People referred to Valia during the years from Hill's family in Ants, and she cut in contact with their family and friends in 2016.

"Each day they stepped on the bus, when they left they saw me on TV," she said.

After separating from their family recently, Valia, 32, also moved into a Richmond house last October.

Read or Share this story:  http://usat.ly/1NNC4zY<|endoftext|>CIVIL C. "Marky" Hogan has been charged with homicides with a few days remaining after the April 2 purdade high school shooting where an undercover medical examiner and two other state and Illinois police officers was using heroin to go see a therapist.

DICEZ TV's Zach Putler reported Tuesday that Hogan was charged with felony drug possession by the Chicago Police Department at the pre-liminary hearing on Monday.  Putler interviewed on Monday.  Author-ities could offer a limit until Cook County takes Tuesday afternoon or they have to assign plea agreements.

Dogan said in a news conference he made during a conference call Wednesday in Chicago that he believes the people who used him as a legal tool in the killing and fired employees hired for suffering also participated.

He said the couple's request to an attorney Monday will let the charges finally play out.  Their lawyer did not respond Wednesday.

Dogan would not give away to possibility that he speculated in a statement that he would escape and return unless shot.

Chicago police initially said the other drug charges failed to raise enough evidence to establish why the killers were charged last year, raising the possibility that the drug mix contributed to a reason for their arrest.  But a new statement was made Tuesday by a man

who worked as the shooter in his unit on campus, and suggested the charges might be related to his work at university supervision.

His attorney and the university's president last week signaled that the incident of the bat gun was not a police investigation at Wednesday's conference.

Michael Durin, Illinois State University spokesman said he did not meet with university officials at the conference, and that university officials don't have any updates yet either.

"The fact that the Defendants were charged is a major factor in why it would get this much attention," the university spokesperson said. "Given that all the matters are not being resolved for months and months, any new specific information and other concerns they may be tasked with investigating now are understandable."

After city police began looking for evidence in connection to Hogan's April shooting, Durin said he had not noticed it until the Chicago Police Department found a person who was producing a bind gun on Illinois State campus.  That same department found that 14 officers shot and were injured during a standoff, but it led to the launch of a combination of unrelated and related investigations leading to homicides charges in May 2015.

A memo from special school investigators suggests it had identified the drug fentanyl, and says the department had described the individual-oriented and inconsistent use of the gun, as well as the substance administered by CODC.

Dogan's allegations claim that a 2009 police gravesite package showed water running over campus and shows that the dental show photos of supposed victims were reinterpreted.

David Mann, a member of the Police Department, said he spoke exclusively to News 1 on condition of anonymity because university officials can't review documents immediately, and university officials had to change information that had been a consideration.

"We didn't change our information until he personally told the drug overdose problem," he said.  "He said that drug dealing wasn't really a focal point."

He however, in a May 15 statement, also was "proleased" with university and state officialsin the U.S. Attorney's Office.  "We do not have any way to estimate the crime syndicate, given that the finding in the case does not preclude a separate review of standard CPD policy directed at any school employee, and whether that employee worked or went out of work," Mann said.

The lawsuit alleges that donations from a full visit with Hogan's workers and thousands of dollars, spent on gift cards, cigars, stock tickets to trips across the country, as well as other financial accounts and income from social members of former employees, were missing from buyers bought in stock.

Both employees and co-participants remain employed at the university.

On the Illinois StateUniversity Facebook page<|endoftext|>Image Steve Healey, the Cook Electronics Co.  Ltd., general manager of Aug.  8-Feb.  16 at a recent internal session meeting (see link) held in The Apple Building in London.  This July 1939 photo shows Peggy Deaver in 1986.  (Photo courtesy the Cook Elec-

```
tronics Co.  Ltd., displayed here under G. Healey in prison
dress.)<|endoftext|>Steve GIRO's H-P<|endoftext|>
```

## E.6.2   T =10000

**Example   1**   `<|endoftext|>` has been presenting a number of temporary
measures to help help resolve a crisis.

```
The last crisis was like this, but we have to resolve it at the spec-
ified level; we have to stay low.  The people of Greece will be in-
terested in the effectiveness of the measures.  They will not only
manage these measures, but also they will help in order to cope with
the problems of the fiscal stability.
```

```
However, we do not want to dispose assets for the treasury.  This
also, so we will work on developing the national economy, and also
paying on the national debt.
```

```
This affects the national incomes
```

```
So, as of 2007-2011, we use the government's temporary measures as
a measure, helping resolve the crisis.  In addition, we are able
to pay for the borrowing costs.  Additionally, we pay $440 billion
to settle debt issues, which can never be settled by default in a
country.
```

```
These temporary measures will be aimed on several fronts, because
the government will have three different partners in the system, in
my right.
```

```
Firstly, we will be allowed to borrow a lot more upon the addition
of the emergency measures and these temporary measures will help
provide for the repayment of our debts before we are forced into a
crisis as a result of our borrowing bills.  Secondly, in the case
of this, we will have resort to temporary measures in the revenue
budget for Greece.  The budget costs the government another $5.2
billion a year.
```

```
So what you propose - if it happens again, does this mean that,
since 2010, will you resolve the deficits which will occur on the
basis of what we already have?
```

```
The fiscal situation
```

```
We would be able to settle our debts by the end of June which is the
end.  That said, we are taking our part as one of the most important
countries in Europe, not only to make a proper transfer of the money
but also to rely on it in the economy.  However, first of all, we
cannot achieve this on a day-to-day basis.
```

```
It is still true that we have decided to be able to deal with the
economic situation of the country, but there might be another change
in the fiscal situation, and therefore, we will try to negotiate on
the situation at the end of June and over the summer.
```

```
The changes in the fiscal situation would be up to the parliament
of management, bureaucrats, judges and a legitimate parliament of
Greece.
```

```
Is the government planning to talk about thethe 'temporary measures'
of Greece?
```

```
We will continue the process to operate through the temporary mea-
sures.  This is not a temporary measure at this point, because after
a crisis, not thereyet at crisis level, you can still have enough
investment purchases until the end of the month.
```

Again the government decided to create a temporary measure and now it depends upon a particular event, such as that there's another liquidity crunch. It is better that the government and the authorities decided to create a temporary measure effective in June at fair sum monthly bond rates.

The temporary measures will also enhance the government's economic status, especially when following the measures at the end of the month.

Temporary measures is a real tool for growth, not just for the economy.

Knowing that there are several measures in place to increase our supply, for example, the level offor profit on public sector enterprises is certain, under all of these temporary measures the increases in output, after that, will increase the external demand and the internal demand.

We will be able to create the demand, and also strengthen the government's credibility through fiscal organization. What is important here, here is that we will apply these measures to our reserves, and at the same time, we apply these measures to the debt level, which will also be the aim of debt-free Greece.

So, first of all, everything is certain ofwhat continues to be collected by the government. Given the situation and after the release of the last data on October 16, you also recognize that this will not be any kind of non-payment.

In the case of the payment against the equipment, we will be able to manage with the measures.

What does government expect in its plans to create a fiscal consolidation for the public sector and the new budget.

Regarding this is the temporary measure, we will be able to cope with the troubled finances. However, I do not think it is any measure which threatens the fiscal stability of the economy. However, that is not a temporary measure, a permanent measure.

On the other hand, there will be our ongoing work on construction in the ministry. If this falls, we will continue work on job creation, the expansion of the economy.

Also, also the government mentioned the new government reforms, which increased labor hours for the employees, which will further the economic growth, and the second aspect of budget as well and this is government welfare, which will improve the quality of life. We will<|endoftext|>

**Example 2**  <|endoftext|> him. He said: "What are you doing?"

I hesitated before answering. "Boy, this is so exciting. You need a better girl. Is she?"

And I said, "You don't have a brain. You have no brain anymore."

After a minute, he had walked back and said on his own, working through that, he thought he had got himself going in a new direction.

He could've been a better boy in the first three years.

"You'll only have once before it starts."

MVP

The story is always, "That's what the other guy has to do." He was the guy who had to do anything. He had to reason with school officials. My cousin mentioned to me that some of my friends almost doubled over at one meeting.

I'd picked up a lot of the money I owed him from high people in me; he liked my grades. Drop-outs didn't consider me high enough to let me go hang out. He hung up when I challenged him after practice to show a new talk. He started making, and, quite, never

I first saw him C. morning, in the sixth grade class. He wouldn't hang up with him on point at team meetings. He started talking about things about me: "I'm an M, I'll get an A. Tonight." Having had that conversation over lunch, my heart touched mine with pride. He came, my boy. Now he looks like he's going back to school. I don't know if he's going to sue. Let's just have a two-bedroom apartment, a $500-dollar condo for renting, and a pool. And then he was back.

That was a part of my life as I think about it. It was the school year.

I never saw a guy come up at the locker room and show a new talk. That day one day, I told the high school, "We'll show up one day right here, we can have a little fun," and after this, I remember a small handful of the boys made friends, and they never, ever showed up for a new talk.

I call them "M's kids. I always remember him, I remember his ass up his ass, getting ready for a freshman orientation out there. He'll show everything.

He'll show if I'm freshman, I'll act I was going to play junior. In a few years I'll try it, then he'll make sure he's going to judge me. He will come over to me one day.

Then one day, the senior class was sitting on the bench, pressing his ball on the floor of the locker room, the referee was just standing the knelt it down.

And when he heard about that, another boy, three of his friends, and one of his cousins were on the other side of the room. The boys' class was filling in with his new brother and his new cousin and his new M.M's player.

The senior class watched me walk me through the chair to the bench. Everyone passed by the boy. Just on his toes on a foot, too.

He [and a girl] passed over his head and, as I looked at them, he carried me into the locker room. And the biggest part of the story, was a mistake.

With his elbows out, he pulled me down on my shoe, my other, sort of a- don't know what they were; palebelly somethings, like bleeding very much, or on little toes. He was up on the stairs and everybody watches, with men and high school kids, who saw him in the locker room. And he caught a breath. Then his old man approached me and, disappeared into the middle of the room. He took off his vest, fast enough as to herd him into the locker room. I walked into the room and read him little cards with my own eyes to make notes, to pull him under my shoes.

I told them: "Listen, because I say this today, when you talk to 'em today, "Just make sure you talk is more than you'll show. He's

just listening to me, and he's telling me I'm going to be there for him."

I'm always the one who wants to do something important about you than I show up.  I walk around and ask, I want a message from you, "Keep it going.  It<|endoftext|>

