# OpenReview forum: "Simple and Effective Masked Diffusion Language Models"
_NeurIPS.cc/2024/Conference — NeurIPS 2024 poster_

### Official Review · Reviewer_wXn1 · 2024-07-12

**Soundness:** 3
**Presentation:** 3
**Contribution:** 3
**Rating:** 6
**Confidence:** 4

**Summary:**

The paper introduces a masked diffusion language modeling (MDLM) framework that enhances the performance of diffusion models in generating high-quality text, closing the gap with autoregressive methods. By applying an effective training strategy and a simplified objective function, MDLM achieves state-of-the-art results among diffusion models on language benchmarks. The approach is shown to be simple yet effective, with potential applications beyond language, including biological sequence modeling.

**Strengths:**

- The method is well motivated and well designed based on the previous works SEDD
- The story is well told for better understanding.

**Weaknesses:**

- In Figure 1, unmasking loss is misleading. What's the difference between the red and yellow colors? Which tokens should we enforce the losses on?
- In Line 38, "Simple" seems overstated. I couldn't quickly understand the method, even though I have some background in diffusion models and sequence modeling.
- In Line 168, there are too many experimental details about the network structure and tokenizer changes, making it unclear how to prove the method works. This, in turn, suggests that the proposed method is not so simple.
- The sampling steps for diffusion-based methods should be highlighted in Table 1 and other related tables.
- In Line 178, the semi-autoregressive part is not clear to me. I lose track of how it works. The need for semi-autoregressive models is not well-motivated. It's a common replacement trick used in diffusion models. The author should explain it.
- In Table 5, memory should also be included. Secondly, Mamba is more favorable for longer token numbers. The authors should also compare with Mamba under various token numbers: 1k, 5k, 10k, and 20k.
- The paper needs to provide the mathematical proof for the continuity equation, showing that \( u_t \) in the equation generates the probability path \( p_t \). It should also highlight the connection to the continuous case.
- In Line59, several works about using flow-based method is missing,[1][2]
- In LIne65, citation missing, Chapell's discrete flow matching paper
- Better discuss with this work: https://arxiv.org/abs/2406.03736

[1]. Language Rectified Flow: Advancing Diffusion Language Generation with Probabilistic Flows

[2]. Flow Matching for Conditional Text Generation in a Few Sampling Steps

Overall, the authors claim the method is simple, but it’s not as simple as they claimed. Additionally, there are many issues with the presentation and experimental parts. Therefore, I am inclined to weakly reject this paper.

**Questions:**

As above

**Limitations:**

Yes

---

> ### Author Response · Authors · 2024-08-07
> **Response to wXn1 (1/3)**
>
> We want to thank the reviewer for their constructive feedback. We address specific comments and questions below.
>
> ---
>
> ### ****Concern 1:**** What is “simple”? It was hard to understand the method.
>
> We use the term “simple” because **our algorithm is very similar to BERT** except for a small change: a random masking rate. Please see the general response for how one of our main contributions, the SUBS parameterization and continuous time perspective, allow us to simplify the training objective.
>
> We will clarify the method by providing the following pseudocode in the next version of the paper. We use **$x$** to denote a one-hot word embedding, and **$m$** is a one-hot vector encoding the mask token.
>
> **Algorithm: MDLM Training:** Inputs: dataset $\mathcal{D}$, monotonic masking schedule $\alpha_t : [0,1] \to [0,1]$, BERT-like model $x_\theta$.
>
> 1. **repeat**
>     1. $\mathbf{x}^{1:L} \sim \mathcal{D}$       *// Sample a sequence of length L from dataset (can be a batch)*
>     2.  $\mathbf{z}^\ell_t \sim \text{cat}(\mathbf{z}^\ell_t; \alpha_t \mathbf{x}^\ell + (1 - \alpha_t)\mathbf{m})$   $\forall 1 \leq \ell \leq L$ for random $t \sim \mathcal{U}[0, 1]$  *// Mask each token $\mathbf{x}^\ell$ independently with masking rate $\alpha_t$ to obtain the latent $\mathbf{z}^{1:L}_t$.*
>     3. Update weights $\theta$ of denoising (BERT) model $\mathbf{x}_\theta$ by gradient descent step on
>
>         $\nabla_\theta \frac{\alpha'\_t}{1 - \alpha\_t} \sum_{\ell} \log \langle \mathbf{x}\_\theta^\ell(\mathbf{z}^{1:L}_t), \mathbf{x}^\ell \rangle$  // Note: this is simply a “weighted” BERT-style loss
>
>
> **until** converged
>
>
> The only differences relative to BERT are that at step (1.2), $\alpha_t$ is a constant, and in step (1.3.) there is no weighting term in front of the cross-entropy loss. Otherwise the training algorithms are equivalent! Indeed, one could even take an off-the-shelf pre-trained BERT and with minimal modifications to the training loop, render it a generative model (Table 4) with an principled ELBO objective and sampling algorithm.
>
> Additionally, unlike previous works, such as CTMC [1] and SEDD [2], which rely on the machinery of continuous-time Markov chains, our work admits a straightforward generation algorithm based on ancestral sampling that we present below:
>
> **Algorithm: MDLM Sampling / Inference**
>
> 1. $\mathbf{z}^{1:L}_1 =$ [MASK, MASK, …, MASK]    *// Sampling process starts with all MASK tokens.*
> 2. **for** t = {$1, \frac{T-1}{T}, \dots, \frac{1}{T}$} **do**
>     1. $s \xleftarrow{} t - \frac{1}{T}$
>     2. $\mathbf{z}^\ell_{s} \sim \text{Cat}(\mathbf{z}^\ell_s; \frac{(1 - \alpha_s)\mathbf{m} + (\alpha_s - \alpha_t)\mathbf{x}_\theta(\mathbf{z}_t)}{1 - \alpha_t} )$ if  $\mathbf{z}_t^\ell = \text{[MASK]}$ else $\mathbf{z}_s^\ell = \mathbf{z}_t^\ell$   $\forall 1 \leq \ell \leq L$.
>     3. $\mathbf{z}^{1:L}_t \xleftarrow{} \mathbf{z}^{1:L}_s$
>
>     end **for**
>
> 3. **return** $\mathbf{z}_0$
>
> To further underscore the simplicity of our method, we highlight the minimal form of our variational lower bound, Equation 10, which ends up being a weighted average of BERT-style losses:
> $\mathcal{L} = \mathbb{E}\_q\int_{t=0}^{t=1}\frac{\alpha\_t'}{1-\alpha\_t} \log \langle\mathbf{x}_\theta(\mathbf{z}_t), \mathbf{x}_0\rangle dt$
>
> In summary, our work presents a principled approach to turn widely-used bi-directional language models into generative ones.

---

> ### Author Response · Authors · 2024-08-07
> **Response to wXn1 (2/3)**
>
> ### **Concern 2:** MDLM involves a number of hyper-parameters. It is unclear these are same for baselines, and whether the comparison is fair.
>
> We stress that our experiments **use exactly the same hyper-parameters** (tokenizer, network structure, training set, optimizer settings, etc.) **across every model** that we have implemented (AR, SEDD, which is the current state-of-the-art in discrete diffusion, MDLM, as well as models in other tables such as D3PM).
>
> |  | SEDD | MDLM | AR |
> | --- | --- | --- | --- |
> | Tokenizer | `bert-base-uncased` for LM1B / `gpt-2` for OWT   | `bert-base-uncased` for LM1B / `gpt-2` for OWT | `bert-base-uncased` for LM1B / `gpt-2` for OWT |
> | Architecture | DiT | DiT | DiT |
> |  Context size | 128 for LM1B / 1024 for OWT | 128 for LM1B / 1024 for OWT | 128 for LM1B / 1024 for OWT |
> | Train steps | 1M steps | 1M steps | 1M steps |
> | Perplexity ($\downarrow$) on LM1B (1M train steps) | 32.79 | 27.04 | 22.32 |
> | Perplexity ($\downarrow$) on OWT | 24.10 | 23.21 | 17.54 |
>
> To further clarify: in Section `3.5.1`, we specifically highlighted architectural details used in MDLM that contribute to a performance relative to the results in the original D3PM paper. However, our main tables compare against key baselines (SEDD, AR, and D3PM, which is MDLM after all ablations) **with the same architecture.**
>
> ---
>
> ### ****Concern 3:**** It is unclear how semi-autoregressive sampling works.
>
> Naively, diffusion models only generate fixed-length sequences. Semi-autoregressive sampling allows diffusion to produce arbitrary-length sequences by generating (via diffusion) a block of text that is conditioned on previously generated blocks of text (blocks previously obtained via diffusion).
>
> ****Algorithm: MDLM Semi-AR sampling****
>
> Generates a Sequence of length $B \times L$ where $L$ is the context length of the diffusion model.
>
> 1. $S = \phi$    *// We start with a null sequence.*
> 2. $\mathbf{x}^{1:L} =$$[MASK]^{1:L}$   *// Sampling process starts with all MASK tokens.*
> 3. **for** $i$ = {${1, \dots, B - 1}$} **do**
>     1. Unmask all masked tokens in $\mathbf{x}^{1:L}$  using the **MDLM Sampling algorithms** as described above.
>     2. $S \xleftarrow{} S \cup \mathbf{x}^{1:L/2}$  // Save the first $L/2$ tokens
>     3. $\mathbf{x}^{1:L/2} \xleftarrow{} \mathbf{x}^{L/2:L}$ // Condition the generation of future tokens on the trailing $L/2$ tokens
>     4. $\mathbf{x}^{L/2:L} \xleftarrow{} [MASK]^{1:L/2}$ // Mask out the last $L/2$ tokens to be filled out by MDLM in the next iteration.
>
>     end **for**
>
> 4. return $S$
>
> For a figure illustrating how our proposed semiAR algorithm works, please see the schematic in the attached supplementary materials [here](https://openreview.net/forum?id=L4uaAR4ArM&noteId=WereEXicWc)
>
> ----
>
> ### **Additional questions and concerns**
>
> > **Include sampling steps for Table 1**
>
> There are different ways to interpret this question. First, sampling steps could refer to the value of T used at training time. We report our best PPL values with the continuous-time formulation ($T \to \infty$) for MDLM and SEDD, and report results with finite $T$ among our ablations. A second interpretation is that “sampling steps” could refer to the number of generate text from the model when we evaluate generative PPL. This number is reported on the x-axis of our generative PPL figure and varies between 32 and the length of the sequence.
>
> A third interpretation refers to the Monte Carlo sampling to evaluate PPL. When evaluating the perplexity bound of MDLM or SEDD, we use a single Monte Carlo sample per datapoint to approximate the integral of the continuous-time bound. Our low-discrepancy antithetic sampler allows us to estimate the perplexity bound with low variance, as shown in the table below. On OpenWebText, we find that we are able to accurately estimate the perplexity bound with just a single sample. In contrast, prior work required 1,000 samples to obtain an accurate estimate.
>
> Below we include a table that shows the limited effect of including more time steps in the Monte-Carlo estimate of MDLM on the OpenWebText dataset:
> | Num MC samples            | 1     | 2     | 5     | 10    | 100   |
> |---------------------------|-------|-------|-------|-------|-------|
> | Perplexity ($\downarrow$) | 23.21 | 23.22 | 23.22 | 23.22 | 23.21 |
>
> > **Better explaining Figure 1**
>
> The loss in Figure 1 is computed over red boxes only. Since the yellow boxes are unmasked, by the definition of our SUBS parameterization, the loss there is zero by construction. We are adding a legend to explain this.
>
> > **Mathematical proof for continuity equation.**
>
> There is currently no mention of $u_t$ in our work. Can you please provide further clarification as to what you are requesting in this regard?

---

> ### Author Response · Authors · 2024-08-07
> **Response to wXn1 (3/3)**
>
> > **Comparing efficiency w/ Mamba under different token numbers**
>
> We compare the wall clock times of 1 forward pass of the bidirectional transformer used in MDLM vs. Mamba under varying context lengths. While Mamba achieves favorable runtimes at context lengths greater than 8k tokens, there is not a significant speed improvement at the context lengths studied in this paper (where we seek to match GPT-2 using diffusion). Although long context generation is not a focus of this work, we believe it is an interesting area for future extensions.
>
> | # of tokens (wikitext103) | 2k | 4k | 8k | 16k | 32k |
> | --- | --- | --- | --- | --- | --- |
> | Mamba (runtime in ms) | 11 | 12 | 22 | 42 | 83 |
> | Transformer (runtime in ms) | 13 | 17 | 36 | 88 | 250 |
>
>
> We report the above runtimes on a single A5000 GPU, excluding the wall clock time of word embeddings.

---

> > ### Comment · Reviewer_wXn1 · 2024-08-12
> > **Thanks for the reply**
> >
> > Thanks for the authors' reply and the reviewer's feedback.
> >
> > My concern about the simplicity of the method is addressed by the pseudo alg. However, without explicit discussion about related works in the main paper, I don't feel my concerns are fully resolved.

---

> ### Author Response · Authors · 2024-08-13
> **Discussion on related works**
>
> Thank you for your feedback and response to our rebuttal. Below we provide discussions on the requested related works [1, 2, 3, 4]:
>
> Language Rectified Flow [1] and Flow Matching for Conditional Text Generation in a Few Sampling Steps [2] perform flow matching over word embeddings. These works are more similar to Plaid [6] and Diffusion-LM [7], where the diffusion process is defined over a continuous space. In contrast, MDLM applies diffusion processes to discrete structures. We will include this discussion in the updated version of the manuscript.
>
> Furthermore, we would like to highlight that in Section 6 of the paper, where we have provided a comprehensive literature review covering areas such as Continuous Time Markov Chains [8, 11], score estimation [9, 10], and techniques that use BERT for sample generation [12, 13, 14, 15].
>
> ## Comparisons to Discrete Flow Matching [3]:
> Discrete Flow Matching (DFM) proposes flow matching for discrete structures. They use the following cross entropy loss as their training objective:
> $\mathcal{L}\_\text{DFM} = - \int_{t=0}^{t=1}\log p_\theta(\mathbf{x}^{1:L}|\mathbf{z}^{1:L}_t) dt$
>
> Similar to [5], DFM’s objective, while effective, is not weighted appropriately to make it a proper ELBO. However, in MDLM, we derive a tight and principled lower bound to the log-likelihood.
>
> ## Comparisons to Your Absorbing Discrete Diffusion Secretly Models the Conditional Distributions of Clean Data. [4]:
> [4] reaches a similar training objective to that derived in our work, although they start their analysis from the score matching perspective presented in SEDD [10]. Interestingly, they find an important equivalence between denoising concrete score matching and the variational lower bound that we derive. A similar connection was also made in Shi et al. [16]. In contrast, we tackle this problem starting via a variational lens: where we derive the exact continuous time ELBO by endowing D3PM with the novel SUBS parameterization and taking the number of diffusion steps T to infinity. Also of note is that [4] leverages a time-independent de-noising network to formulate an efficient sampler that limits function evaluations, as in our work.
>
> A key point of differentiation between our works is that we tackle an important limitation of diffusion models, namely that they are limited to generating a fixed sized context. Our proposed semi-AR algorithm mitigates this shortcoming, and  is therefore a **novel contribution to this line of work that separates MDLM from [4].**
>
> Finally, **turning to a comparison of the empirical evaluation: here, we believe that our work is significantly better.**
> - Most importantly, we explicitly compare our discrete diffusion model on the widely used objective of **perplexity** and demonstrate that non-autoregressive models are approaching the performance of autoregressive ones. This is a significant milestone, as perplexity has been the guiding watermark for the field of language modeling. In contrast, [4] only use the less informative metrics of “zero-shot” perplexity and “generative perplexity”.
> - Furthermore, we analyze domains outside of NLP and demonstrate that the approach can be used on biological sequences, unlike [4] and [16], which only focus on NLP datasets.
> - Finally, our Table 4, demonstrates how one can **take an off-the-shelf BERT model and render it generative without losing representation learning capabilities.** This is an important result in our work that is not present in either [4] or [16].
>
> We end by noting that [4,16] are highly concurrent and were posted on arXiv after the Neurips submission deadline. According to Neurips guidelines, authors are **not expected to compare their work to papers that appeared on arXiv less than two months before the submission deadline**, let alone those published after it.

---

> > ### Comment · Reviewer_wXn1 · 2024-08-13
> > **accept**
> >
> > I appreciate the author's reply about the related works. Even though some papers are concurrent works, I think it can be discussed. I am not asking for a comparison.
> >
> > I am excited to see the deep relationship between them. My concerns have been fully resolved, and I will increase my score to reflect this.

---

> > > ### Author Response · Authors · 2024-08-13
> > > **Thank you**
> > >
> > > Dear Reviewer,
> > >
> > > We really appreciate your feedback and continued engagement. We'll incorporate these discussions in the next version of the manuscript.
> > >
> > > Thanks,
> > > authors

---

> > ### Comment · Reviewer_wXn1 · 2024-08-13
> >
> > can you update the reference? I cannot find them

---

> > > ### Author Response · Authors · 2024-08-13
> > > **Updated references**
> > >
> > > We have added the references in a separate comment here:  https://openreview.net/forum?id=L4uaAR4ArM&noteId=fwSYxq1dTI

---

> > ### Author Response · Authors · 2024-08-13
> > **References**
> >
> > ## References:
> >
> > [1]. Zhang, S., Wu, L., Gong, C., & Liu, X. (2024). Language rectified flow: Advancing diffusion language generation with probabilistic flows. arXiv preprint arXiv:2403.16995.
> >
> > [2]. Hu, V., Wu, D., Asano, Y., Mettes, P., Fernando, B., Ommer, B., & Snoek, C. (2024, March). Flow Matching for Conditional Text Generation in a Few Sampling Steps. In Proceedings of the 18th Conference of the European Chapter of the Association for Computational Linguistics (Volume 2: Short Papers) (pp. 380-392).
> >
> > [3] Campbell, A., Yim, J., Barzilay, R., Rainforth, T., & Jaakkola, T. (2024). Generative flows on discrete state-spaces: Enabling multimodal flows with applications to protein co-design. arXiv preprint arXiv:2402.04997.
> >
> > [4] Ou, J., Nie, S., Xue, K., Zhu, F., Sun, J., Li, Z., & Li, C. (2024). Your Absorbing Discrete Diffusion Secretly Models the Conditional Distributions of Clean Data. arXiv preprint arXiv:2406.03736.
> >
> > [5] Chang, Huiwen, et al. "Maskgit: Masked generative image transformer." Proceedings of the IEEE/CVF Conference on Computer Vision and Pattern Recognition. 2022.
> >
> > [6] Gulrajani, Ishaan, and Tatsunori B. Hashimoto. "Likelihood-based diffusion language models." Advances in Neural Information Processing Systems 36 (2024).
> >
> > [7] Li, Xiang, et al. "Diffusion-lm improves controllable text generation." Advances in Neural Information Processing Systems 35 (2022): 4328-4343.
> >
> > [8] Andrew Campbell, Joe Benton, Valentin De Bortoli, Thomas Rainforth, George Deligiannidis, and Arnaud Doucet. A continuous time framework for discrete denoising models. Advances in Neural Information Processing Systems, 35:28266–28279, 2022.
> >
> > [9] Yang Song and Stefano Ermon. Generative modeling by estimating gradients of the data distribu- tion. Advances in neural information processing systems, 32, 2019.
> >
> > [10] Aaron Lou, Chenlin Meng, and Stefano Ermon. Discrete diffusion language modeling by estimating the ratios of the data distribution. arXiv preprint arXiv:2310.16834, 2023.
> >
> > [11] Haoran Sun, Lijun Yu, Bo Dai, Dale Schuurmans, and Hanjun Dai. Score-based continuous-time discrete diffusion models. arXiv preprint arXiv:2211.16750, 2022.
> >
> > [12] Zhengfu He, Tianxiang Sun, Kuanning Wang, Xuanjing Huang, and Xipeng Qiu. Diffusion- bert: Improving generative masked language models with diffusion models. arXiv preprint arXiv:2211.15029, 2022.
> >
> > [13] Marjan Ghazvininejad, Omer Levy, Yinhan Liu, and Luke Zettlemoyer. Mask-predict: Parallel decoding of conditional masked language models. In Kentaro Inui, Jing Jiang, Vincent Ng, and Xi- aojun Wan (eds.), Proceedings of the 2019 Conference on Empirical Methods in Natural Language Processing and the 9th International Joint Conference on Natural Language Processing (EMNLP- IJCNLP), pp. 6112–6121, Hong Kong, China, November 2019. Association for Computational Linguistics. doi: 10.18653/v1/D19-1633. URL https://aclanthology.org/D19-1633.

---

### Official Review · Reviewer_DasC · 2024-07-13

**Soundness:** 3
**Presentation:** 3
**Contribution:** 2
**Rating:** 5
**Confidence:** 3

**Summary:**

This paper presents a method for language modeling using simple masked discrete diffusion models. The authors show that a simplified objective combined with optimized training achieves performance improvements over previous diffusion language models. The paper reports state-of-the-art results for diffusion models on standard language modeling benchmarks, approaching the performance of autoregressive models.

**Strengths:**

1. The simplified diffusion objective is effective in stabilizing training and is well supported by detailed derivations.
2. Section 3 provides clear steps to build a connection between MLM and Diffusion LM
3. The empirical performance is superior to other diffusion language models.

**Weaknesses:**

1. I believe the paper should be structured more towards introducing the simplified objective rather than framing the architecture as MDLM since the main idea of bridging absorbing state discrete diffusion with MLM has been introduced in DiffusionBert and D3MM.
2. The comparative results are not fair comparisons. More specifically the use of a more advanced backbone (DiT) and a low-variance sampler weakens the claim that the performance and stability are sourced from the proposed method.

**Questions:**

1. Other than the simplified objective and new backbone and sampler, what’s the difference between your method and absorbing state D3MM or more specifically DiffusionBert?
2. Is MLM pretraining required for your method?

**Limitations:**

1. Masked language models are special discrete Diffusion models, this is demonstrated in the D3PM paper by rewriting an x_0-parameterized absorbing state model.  This limits the novelty of the paper to the simplified objective only.
2. The idea in this paper has been hinted at in D3MM, limiting the theoretical contribution of this paper.

---

> ### Author Response · Authors · 2024-08-07
> **Response to DASc (1/2)**
>
> We want to thank the reviewer for their constructive feedback. We address concerns and questions below.
>
> ---
>
> ### ****Concern 1:**** Novelty of MDLM relative to the D3PM framework and other algorithms.
>
> In addition to attaining ****state-of-the-art diffusion language model results****, MDLM presents important novel elements relative to prior work.
>
> **Relative to D3PM:** [1]
>
> MDLM is a special case of the D3PM framework that focuses only on masked noise. This allows introducing multiple algorithmic innovations that greatly simplify D3PM and improve performance.
> These design decisions also yield a learning objective that is a ****weighted average of BERT-style loss terms**.** Although D3PM mentions in their appendix connections to BERT, we make this connection much more clear and derive a simpler and more performant algorithm. Our NELBO is given as:
> $\mathcal{L}\_{NELBO} = \int_{t=0}^{t=1}\frac{\alpha'\_t}{1 - \alpha\_t} \log p_\theta(\mathbf{x}^{1:L}|\mathbf{z}^{1:L}_t) dt$
> which achieves a test perplexity of **27.04** after being trained for 1M steps on LM1B while our well engineered implementation of D3PM achieves a worse perplexity of 28.56.
>
>
> - **Algorithmic innovations**
>     - **Simplified noise process:** In D3PM, the noise process is defined via matrix multiplications. In MDLM, we can define simple interpolating noise $\mathbf{z}_t = \alpha_t \mathbf{x} + (1 - \alpha_t)\mathbf{m}$ which allows easily taking $T \to \infty$, which in turn tightens the ELBO.
>     - **Specialized denoising process:** We propose the SUBS parameterization, which includes two key elements: carry-forward unmasking and zero masking probabilities.
>     - **Simplified and tightened ELBO:** Our choice of noising and denoising processes allows us to greatly simplify the D3PM objective (several pages of math in the appendix) and produce a *simplified* and *tighter* ELBO (compare our Equation 8 to Equation 9, which is what D3PM uses).
>
> - **Experimental innovations** We complement the above innovations with a modern training recipe (architecture, optimization) that demonstrates the effectiveness of masked diffusion models. While previous attempts (namely D3PM) seemed to indicate that there exists a large gap between AR and discrete diffusion models, we provide clear evidence to the contrary on the OWT dataset using to train GPT-2 sized models.
>
>
> **Relative to SEDD:** [2]
>
> Our key novelty elements are as follows.
> - We achieve better language modeling results relative to this work, which was previously ****state-of-the-art for diffusion models.****
> - We present a more intuitive and simple objective which boils down to a weighted average of BERT-style losses, circumventing the need to use score-matching techniques and the formalisms of continuous time Markov chains.
> - We support a more efficient inference algorithm that greatly reduces required function evaluations.
>
> **Relative to DiffusionBERT [3]:**
>
> - DiffusionBERT can be described as an instance D3PM with a pre-trained model initialization and a custom noising schedule. Thus, all of the ELBO improvements and optimized training of our work relative to D3PM are also novel relative to DiffusionBERT.
>
>
> ---
>
> ****References:****
>
> [1]  Austin, Jacob, et al. "Structured denoising diffusion models in discrete state-spaces." Advances in Neural Information Processing Systems 34 (2021): 17981-17993.
>
> [2] Lou, Aaron, Chenlin Meng, and Stefano Ermon. "Discrete diffusion language modeling by estimating the ratios of the data distribution." arXiv preprint arXiv:2310.16834 (2023).
>
> [3] He, Zhengfu, et al. "Diffusionbert: Improving generative masked language models with diffusion models." arXiv preprint arXiv:2211.15029 (2022).

---

> ### Author Response · Authors · 2024-08-07
> **Response to DASc (2/2)**
>
> ### **Concern 2:** Is it better to lead the paper with MDLM or with the simplified objective as a key contribution?
>
> To clarify, we use the term MDLM to refer to a language modeling algorithm that is optimized for masked language discrete diffusion and that has the following components:
>
> - **Parameterization:** Our denoising network predicts clean data and uses the SUBS parameterization (carry over masking and zero mask probabilities).
>
> - **Learning objective:** We take $T \rightarrow \infty$ to train with a tight ELBO given in Equation 11.
>
> - **Faster inference:** By removing time-conditioning, we significantly reduce function evaluations during ancestral sampling.
>
> - **Support for arbitrary sequence length generation:** Using a semiAR algorithm, we alleviate a key shortcoming of diffusion models: fixed length generation.
>
> Note that these are novel features over SEDD or D3PM. Thus, while the objective is an important part of our work, it is only one novel part of the MDLM algorithm.
>
> We also acknowledge that we are not the first ones to consider masking diffusion. However, we are the first to optimize our algorithm for masking diffusion, and this motivates its name (MDLM).
>
> ---
>
> ### **Concern 3:** The use of a more advanced backbone (DiT) weakens the claim that the performance comes from the proposed method.
>
> As the table below demonstrates, our ****experimental setup is effectively identical**** between our work, AR, and SEDD (the previous state-of-the-art diffusion-based model), including the backbone. Thus, our ability to more effectively reduce the gap between diffusion and AR models is directly related to our proposed methodology.
>
> Experimental details:
>
> |  | SEDD | MDLM | AR |
> | --- | --- | --- | --- |
> | Tokenizer | `bert-base-uncased` for LM1B / `gpt-2` for OWT   | `bert-base-uncased` for LM1B / `gpt-2` for OWT | `bert-base-uncased` for LM1B / `gpt-2` for OWT |
> | Architecture | DiT | DiT | DiT |
> |  Context size | 128 for LM1B / 1024 for OWT | 128 for LM1B / 1024 for OWT | 128 for LM1B / 1024 for OWT |
> | Train steps | 1M steps | 1M steps | 1M steps |
> | Perplexity ($\downarrow$) on LM1B (1M train steps) | 32.79 | 27.04 | 22.32 |
> | Perplexity ($\downarrow$) on OWT | 24.10 | 23.21 | 17.54 |
>
> To further clarify: in Section 3.5.1, we specifically highlighted architectural details used in MDLM that contribute to a performance relative to the results in the original D3PM paper. However, our main tables compare against key baselines (SEDD, AR, and D3PM, which is MDLM after all ablations) **with the same architecture.**
>
> ---
>
> ### Additional question
>
> > **Is MLM pre-training required for your method?**
>
> No, pre-training with standard MLM loss is ****not**** a requirement of our method. Indeed, the main results of our work, Tables 1 and 2, do not rely on any pre-training. To clarify, in Table 4, we included results of fine-tuning a pre-trained BERT using MDLM to highlight how one could take an off-the-shelf pre-trained representation learning model and turn it into a generative one, without sacrificing the representational learning capabilities of the pre-trained model.

---

> > ### Comment · Reviewer_DasC · 2024-08-10
> >
> > Thanks for the additional information and clarification!
> >
> > Regarding answers to concern 1: My opinion remains the same, and the methods especially zero-masking, carry-over unmasking, and semi-autoregressive sampling are shown to be the source of performance improvement, However, using such methods makes this work a mix of AR and Diffusion, which provides a limited contribution to the bridging of performance gaps between AR and Diffusion.
> >
> > Regarding answers to concern 2: The refined objective's sole contribution remains unclear, and needs to be studied separately from the other design changes. I believe that the refined objective can stabilize the training, while the other design changes are the major sources of improvement over SEDD. However, these additional design changes introduce an AR nature to your method from my intuition. If such designs do not show superior performance than AR, these design changes seem like "borrowing power" from the strong AR baselines, as mentioned above.
> >
> > Regarding answers to concern 3: Thanks for the clarification.

---

> ### Author Response · Authors · 2024-08-11
> **Response to Reviewer DasC**
>
> Thank you for your feedback and response to our rebuttal. Unfortunately, we believe there are several factual errors in your understanding of our method, which we would like to clarify:
>
>
>
> 1. There are **no other sources of contribution to the performance of the method besides our new objective** in the head-to-head comparison against SEDD that we report.
> 2. Our method is not a mix of auto-regression and diffusion, and **no major perplexity result relies on auto-regressive components**.
>
>
> ## **Concern 1:** The refined objective's sole contribution remains unclear. Other design changes are the major sources of improvement over SEDD.
>
> We’d like to re-emphasize that SEDD and MDLM have an identical experimental setup on language modeling benchmarks, as detailed below. **The only difference between these two methods lies in the objective**. Thus, the below table quantifies the **sole contribution** of MDLM’s objective over SEDD.
>
>
> |  | SEDD (Our reproduction) | MDLM |
> | --- | --- | --- |
> | **_Common experimental details_** |
> | Tokenizer | `bert-base-uncased` for LM1B / `gpt-2` for OWT   | `bert-base-uncased` for LM1B / `gpt-2` for OWT |
> | Architecture | DiT | DiT |
> |  Context size | 128 for LM1B / 1024 for OWT | 128 for LM1B / 1024 for OWT |
> | Train steps | 1M steps | 1M steps |
> | **_Objectives (the only difference between SEDD and MDLM)_** |
> | | Diffusion weighted denoising score entropy (Eq 9 in SEDD) | Simplified ELBO that becomes weighted sum of BERT style losses (Eq 11 in our work) |
> | **_Performance_** |
> | Perplexity ($\downarrow$) on LM1B | 28.90 | **27.04** |
> | Perplexity ($\downarrow$) on OWT | 24.10 | **23.21** |
>
>
> ## **Concern 2:** This work is a mix of AR and Diffusion. MDLM’s improved performance is because it “borrows” power from AR.
>
> This is a major misunderstanding. Please note that **no** significant result (Tables 1, 2, 3, 4, 6) uses anything related to auto-regression. All of our perplexity results are **benchmarked against SEDD in a fully non-autoregressive setting following their setup (data, context window, etc.)**. In fact, one of our key contributions is that these results are obtained using a simple BERT-style non-autoregressive loss.
>
> The proposed semi-autoregressive sampler is only used in a minor experiment (Table 5) to allow MDLM to generate sequences of much longer lengths than those on which it was originally trained. However, this sampler **does not** contribute to MDLM’s improved perplexity scores when compared to baselines such as SEDD or D3PM. In fact, the semi-AR experiments do not even compare against SEDD, only against SSD-LM.

---

### Official Review · Reviewer_BSmY · 2024-07-13

**Soundness:** 3
**Presentation:** 2
**Contribution:** 2
**Rating:** 5
**Confidence:** 3

**Summary:**

While previous works considered diffusion language models less competitive than autoregressive models in text generation tasks, the authors propose a simple framework named masked diffusion language modeling (MDLM), where they claim to have better performance than previous thoughts. The authors derive a simplified Rao-Blackwellized objective and find that the objective equals to a masked language modeling with a varying mask ratio. The proposed methods are evaluated on both classification tasks and generation tasks. The performance excels in classification tasks and keeps consistent with autoregressive methods in generation tasks, which achieves a new state-of-the-art among diffusion models.

**Strengths:**

* The structure of the paper is complete.
* The connection between diffusion model and masked modeling sounds good.
* The experiments are comprehensive, covering both classification and generation tasks.

**Weaknesses:**

The main weakness of this paper is about the presentation. This includes the following two points:
* In the third section, the authors first derive an objective with a diffusion process. Then, the authors try to claim what difference has been made compared to the previous D3PM. It seems that the different parameterization made in Equation (7) leads to a simpler formulation in Equation (8) compared to Equation (9), if I understand correctly. However, I am quite confused with the illustration in Section 3.2 and Section 3.3. There appears some technical terms without furthur explanations or citations, such as Rao-Blackwellization and graphical modeling. The abbreviation SUBS is also unclear. Some omittions and simplifications are not provided with some explanations to clarify their rationality.
* There also lacks an overview on the algorithm. Due to the highly complicated derivation process and objective function, I suggest the authors making an algorithm bar to illustrate how the training and the sampling are conducted, like Algorithm 1 and Algorithm 2 in DDPM [1].

There also lacks some discussions with some previous works, including:
* There has been a nearly same method in the CV field called MAGE [2]. MAGE also conducts masked modeling on image patch tokens with a varying mask ratio and generates by the ancestral sampling.
* There lacks a citation for semi-autoregressive modeling [3] since I believe this concept is not first raised in this paper.

[1] Denoising Diffusion Probabilistic Models, https://arxiv.org/pdf/2006.11239

[2] MAGE: MAsked Generative Encoder to Unify Representation Learning and Image Synthesis, CVPR 2023

[3] Semi-Autoregressive Neural Machine Translation, EMNLP 2018

**Questions:**

* I wonder what previous tries have been made in the language diffusion model and what makes them fail to outperform autoregressive model in generation tasks, as diffusion models in the CV field largely outperform autoregressive models.
* I wonder the computational cost of the method and its comparison with the autoregressive model's since it is known that the diffusion model usually costs more time to converge.

**Limitations:**

The authors have claimed their limitation in the paper.

---

> ### Author Response · Authors · 2024-08-07
> **Response to BSmY (1/3)**
>
> We thank the reviewer for their constructive feedback. We address the concerns and questions below.
>
> ---
>
> ### ****Concern 1:**** Adding algorithms for training and inference.
>
> Below we provide pseudocode for MDLM training and inference. We also include these in our revised manuscript. We call out the simplicity of the training algorithm and its similarity to BERT-style training of masked language models:
>
> ****Algorithm: MDLM Training:**** Inputs: dataset $\mathcal{D}$, monotonic masking schedule $\alpha_t : [0,1] \to [0,1]$, BERT-like model $x_\theta$, $\mathbf{m}$ denotes the mask vector, $\mathbf{x}^{1:L}$ denotes a sentence with $L$ tokens, $\mathbf{z}^{1:L}_t$ denotes the latent vector with $L$ tokens.
>
> 1. **repeat**
>     1. $\mathbf{x}^{1:L} \sim \mathcal{D}$       *// Sample a sequence of length L from dataset (can be a batch)*
>     2.  $\mathbf{z}^\ell_t \sim \text{Cat}(\mathbf{z}^\ell_t; \alpha_t \mathbf{x}^\ell + (1 - \alpha_t)\mathbf{m})$   $\forall 1 \leq \ell \leq L$ for random $t \sim \mathcal{U}[0, 1]$  *// Mask each token $\mathbf{x}^\ell$ independently with masking rate $\alpha_t$ to obtain the latent $\mathbf{z}^{1:L}_t$.*
>     3. Update weights $\theta$ of denoising (BERT) model $\mathbf{x}_\theta$ by gradient descent step on
>
>        $\nabla_\theta \frac{\alpha'\_t}{1 - \alpha\_t} \sum_{\ell} \log \langle \mathbf{x}_\theta^\ell(\mathbf{z}^{1:L}_t), \mathbf{x}^\ell \rangle$  // Note: this is simply a “weighted” BERT-style loss
>
>
>     **until** converged
>
>
> Note that the only differences to BERT are that in standard BERT
>
> 1. In step 1.b., $\alpha_t$ is a constant (set to 0.85 implying a fixed masking rate of 15%).
> 2. In step 1.c. there is no weighting term in front of the cross-entropy loss.
>
> Otherwise the training algorithms are equivalent!
>
> ****Algorithm: MDLM Sampling / Inference****
>
> 1. $\mathbf{z}^{1:L}_1 =$ [MASK, MASK, …, MASK]    *// Sampling process starts with all MASK tokens.*
> 2. **for** t = {$1, \frac{T-1}{T}, \dots, \frac{1}{T}$} **do**
>     1. $s \xleftarrow{} t - \frac{1}{T}$
>     2. $\mathbf{z}^\ell_{s} \sim \text{Cat}(\mathbf{z}^\ell_s; \frac{(1 - \alpha_s)\mathbf{m} + (\alpha_s - \alpha_t)\mathbf{x}_\theta(\mathbf{z}_t)}{1 - \alpha_t} )$ if  $\mathbf{z}_t^\ell = \text{[MASK]}$ else $\mathbf{z}_s^\ell = \mathbf{z}_t^\ell$   $\forall 1 \leq \ell \leq L$.
>     3. $\mathbf{z}^{1:L}_t \xleftarrow{} \mathbf{z}^{1:L}_s$
>
>     end **for**
>
> 3. **return** $\mathbf{z}_0$

---

> ### Author Response · Authors · 2024-08-07
> **Response to BSMY (2/3)**
>
> ### ****Concern 2:**** The derivation of the simplified ELBO objective needs more explanation.
>
> The reviewer is asking for clarifications regarding how the unsimplified loss (9), given by
>
> $\mathbb{E}_q \Big[\frac{\alpha_s - \alpha_t}{1 - \alpha_t} \log \frac{\alpha_t \langle \mathbf{x}\_\theta(\mathbf{z}_t, t), \mathbf{m} \rangle + (1 - \alpha\_t)}{(1 - \alpha\_t) \langle \mathbf{x}\_\theta(\mathbf{z}_t, t), \mathbf{x} \rangle} + \frac{1 - \alpha\_s}{1 - \alpha\_t} \log \frac{(1 - \alpha_s)(\alpha_t \langle \mathbf{x}\_\theta(\mathbf{z}_t, t), \mathbf{m} \rangle + (1 - \alpha_t))}{(1 - \alpha_t)(\alpha_s \langle \mathbf{x}\_\theta(\mathbf{z}_t, t), \mathbf{m} \rangle + (1 - \alpha_s))} \Big] \langle \mathbf{z}_t, \mathbf{m}\rangle$
>
> is transformed into the simplified ELBO (8)
>
> $\mathbb{E}_{q,t}\frac{\alpha\_t'}{1-\alpha\_t}\log\langle\mathbf{x}\_\theta(\mathbf{z}_t), \mathbf{x}_0\rangle$
>
>
> and the role that Sections 3.2 (SUBS parameterization) and 3.3 (Rao-Blackwellization) play in this process.
>
> In brief, we obtain (9) by taking our interpolating masking forward process (first paragraph in Section 3.2.1) and inserting it into the general D3PM noise process (3) to obtain analytical formulas (4) and (5) for the marginals and posterior of $q$. Inserting (4) and (5) into the standard diffusion loss (2) yields (9).
>
> Simplifying (9) into (8) mainly involves algebraic manipulations; see Appendix Sec. G. This algebra crucially requires two identities:
>
> 1. $\langle\mathbf{x}_\theta(\mathbf{z}_t, t), \mathbf{m}\rangle = 0$ , i.e. the masked token has zero predicted probability
> 2. if $\mathbf{z}\_t$ is unmasked, then we desire $\mathbf{x}_\theta(\mathbf{z}_t, t) = \mathbf{z}_t$
>
> In order to ensure that these two properties hold, we require that the denoising process has the form given in (7) in Section 3.2.2. We call this novel parameterization SUBS.
>
> Section 3.3 (and its corresponding appendix) focuses on transforming (9) to (8) using these properties. We refer to the process of obtaining (8) in lieu of (9) as a form of Rao-Blackwellization.
>
> The above is a summary of how we derive the simplified ELBO (8) from (9). Next, we define the technical terms identified by the reviewer, and explain how they correspond to instances of general statistical techniques used in the simplification of (9) to (8).
>
> **SUBS parameterization:** This refers to the parameterization in (7). It implements two substitutions that we enforce on the output of $x_\theta$:
>
> 1. We design the denoising network such that $\langle\mathbf{x}_\theta(\mathbf{z}_t, t), \mathbf{m}\rangle$ = 0, i.e., we substitute the logit index corresponding to the [MASK] token with −∞. This ensures property #1 above.
>
> 2. If $\mathbf{z}\_t$ is unmasked, then we desire  $\mathbf{x}_\theta(\mathbf{z}_t, t) = \mathbf{z}_t$ , i.e., unmasked latents are ‘carried over’. We accomplish this by substituting the output of our network to simply copy unmasked inputs. This ensures property #2 above.
>
> **Rao-Blackwellization** is a statistical method that improves an estimator’s efficiency by computing expectations analytically, thereby reducing variance. In our case, we analytically compute expectations such as $\langle\mathbf{x}\_\theta(\mathbf{z}\_t, t), \mathbf{m}\rangle = 0$  in order to simplify objective (9) to obtain (8). Without these analytical simplifications, a model must learn $\theta$ such that  $\langle\mathbf{x}\_\theta(\mathbf{z}\_t, t), \mathbf{m}\rangle = 0$ holds, which slows down training. Unlike in regular Rao-Blackwellization, simplifications are possible because of modeling choices for $\mathbf{x}_\theta(\mathbf{z}\_t, t)$ (zero masking probabilities and carry-over unmasking). However, our approach also empirically helps reduce variance, hence we refer to it as Rao-Blackwellization, somewhat abusing the usual terminology.
>
> **Graphical modeling** is a branch of machine learning that studies how to design parameter-efficient model by explicitly incorporating conditional independencies between random variables into the design of a model. Examples of algorithms in this field include Bayes networks and Markov random fields. In that sense, our approach has similarities to graphical modeling, where incorporating conditional independencies into $p_\theta$ (via our modeling choices for $\mathbf{x}\_\theta(\mathbf{z}\_t,t))$ sets certain log-likelihood terms to zero.

---

> ### Author Response · Authors · 2024-08-07
> **Response to BSmY (3/3)**
>
> ### ****Concern 3:**** MDLM vs MAGE
>
> While both MDLM and MAGE use BERT-style losses to train generative models, the main differences are as follows:
>
> - ****Objective:**** In MDLM, we derive and train with a tight and principled lower bound to the log-likelihood. MAGE’s objective, while effective, is more heuristic with a combination of BERT-style cross entropy on masked tokens (not weighted appropriately to make it an ELBO) and a contrastive loss term.
>
> - ****Sampler:**** In MDLM, we use a principled ancestral sampling method, whereas in MAGE once again the sampling is done in a more heuristic manner that yields impressive results in the image generation domain.
>
> Additionally, we thank the reviewer for the additional reference Semi-Autoregressive Neural Machine Translation. In our updated manuscript we add this citation and previous works on semi-autoregressive modeling.
>
> ---
>
> ### Additional questions
>
> ****### **Question 1:****** What previous tries have been made in the language diffusion model and what makes them fail to outperform autoregressive model in generation tasks?
>
> Broadly, past attempts at diffusion modeling for discrete data can be broken into two categories, (1) works that first embed text in continuous space and then run standard Gaussian diffusion on the embeddings before coming back to the discrete domain and (2) works that directly define a corruption process on the discrete data. The first line of work seems to suffer from training instability and the results lack in quality relative to the dominant AR approach (see for example DiffusionLM [1]). The second approach has recently shown promise.
>
> Both our work and the previous state-of-the-art diffusion language model, SEDD [2], propose novel parameterizations of the denoising network that induce a better loss:
>
> - concrete score matching with positivity constraints, in the case of SEDD
> - and a weighted sum of cross-entropy terms, with reconstruction and prior regularization loss terms analytically evaluating to zero, in our case.
>
> These better losses are key ingredients that contribute to the success of recent efforts that close the gap to AR language modeling.
>
> ****### **Question 2:******  What is the computational cost of the method in comparison to  autoregressive models?
>
> We find that AR models are able to optimize the training loss better than diffusion models [see fig. 1 in the attached supplementary material](https://openreview.net/forum?id=L4uaAR4ArM&noteId=WereEXicWc) However, diffusion models are able to reach the same loss values with additional training. Crucially, the gap between the two curves is much smaller than previously thought. We are excited by the new opportunities that non-AR generation present, such as methods for controlled generation, and the promise of more efficient sampling.
>
> ---
>
> ****References:****
>
> [1] Li, Xiang, et al. "Diffusion-lm improves controllable text generation." Advances in Neural Information Processing Systems 35 (2022): 4328-4343.
>
> [2] Lou, Aaron, Chenlin Meng, and Stefano Ermon. "Discrete diffusion language modeling by estimating the ratios of the data distribution." arXiv preprint arXiv:2310.16834 (2023).

---

### Official Review · Reviewer_p1DW · 2024-07-14

**Soundness:** 3
**Presentation:** 3
**Contribution:** 3
**Rating:** 5
**Confidence:** 5

**Summary:**

This paper introduces a new approach to masked diffusion language models (MDLMs) that improves performance over previous discrete diffusion methods. The authors present a simplified, Rao-Blackwellized objective for training MDLMs, which is derived from a substitution-based parameterization of the reverse diffusion process. This objective takes the form of a weighted average of masked language modeling losses, establishing a connection between generative diffusion models and encoder-only BERT models. The paper also describes efficient sampling techniques, including a semi-autoregressive decoding method. The authors demonstrate state-of-the-art performance among diffusion models on language modeling benchmarks, approaching the perplexity of autoregressive models. Additionally, they show that their approach can be applied to biological sequence modeling, achieving competitive results on DNA modeling tasks. The paper emphasizes the importance of well-engineered implementation details and provides comprehensive ablation studies to validate their design choices. Overall, this work presents a simple yet effective framework for discrete diffusion models that bridges the gap between diffusion-based and traditional language modeling approaches.

**Strengths:**

- The paper introduces a novel, simplified objective for masked diffusion language models, which is a  combination of existing ideas from diffusion models and masked language modeling. The substitution-based (SUBS) parameterization of the reverse diffusion process is an original contribution that enables the derivation of a more effective training objective. The connection established between generative diffusion models and encoder-only BERT models is an interesting perspective in the field.
- The empirical results are convincing, demonstrating good performance among diffusion models on multiple benchmarks. The extension to biological sequence modeling shows the versatility and robustness of the method.
- The paper is well-structured and clearly written, with a logical flow from theoretical foundations to practical implementations and results.
- The work  narrows the performance gap between diffusion-based and autoregressive language models, potentially leading to more adoption of diffusion models in NLP.

**Weaknesses:**

- The paper focuses heavily on empirical results but lacks a deeper theoretical analysis of why the proposed MDLM approach outperforms previous methods. A more rigorous theoretical foundation could provide insights into the method's success and potential limitations.
- While the paper compares MDLM to other diffusion-based approaches, it lacks a comprehensive comparison to state-of-the-art non-diffusion language models. This makes it difficult to fully assess the method's competitiveness in the broader context of language modeling.
- The experiments focus on relatively small models. It's unclear how well the approach scales to larger models that are more common in current NLP research. Addressing potential scalability issues or limitations would be valuable.
- The paper primarily focuses on perplexity as an evaluation metric. Including human evaluations or other metrics that assess the quality and coherence of generated text would provide a more comprehensive view of the model's capabilities.
- It's unclear if MDLM can achieve practical speed-up comparing to AR models with same FLOPs budget due to the lack of comparative e valuation on this.
- Incomplete reference and comparision to previous related works, see Questions section.

**Questions:**

- Why didn't the authors include a comparison to Plaid in their perplexity evaluation experiments (Tables 1 & 2)?
- How do the authors explain their observation that time-step conditioning can be optional in MDLM? Particularly, why is there no conditioning in OWT? Does this suggest that with a larger amount of data tokens, the need for timestep conditioning in MDLM decreases?
- How many steps are used when evaluating the perplexity of MDLM? How does this compare to other text diffusion models?
- Can MDLM achieve any efficiency gains in inference and sampling compared to autoregressive (AR) models within the same compute budget? For instance, if we need $K$ steps to generate $B$ tokens at a time in semi-MDLM, how can we make a fair and scientific comparison to AR models generating $L$ tokens in this setting?
- One disadvantage of BERT-style models is that they are less token-efficient in training compared to decoder-only AR models. Can the authors provide some comparative analysis and plots showing how MDLM compares to AR baselines across a range of training token amount budgets?
- What are the authors' thoughts on MDLM versus continuous-diffusion models for text, such as Plaid and CDCD, in terms of training, model performance, and inference efficiency? Is MDLM superior to these continuous-space text diffusion models in terms of resulting model quality and reducing the gap to AR models?
- Is it possible to speed up the generation process of MDLM, similar to the advanced ODE-solvers people are using in continuous-space diffusion models?
- How does MDLM compare to MaskGiT in terms of model formulations? They share lots of same design space but I didnt see any reference or comparison to this related line of works.

---

> ### Author Response · Authors · 2024-08-07
> **Response to p1DW (1/3)**
>
> We want to thank the reviewer for their constructive feedback. We address the reviewers comments and questions below.
>
> ---
>
> ### ****Concern 1:**** Why does MDLM outperform previous methods? There is a need for a deeper theoretical analysis.
>
> Our work outperforms previous discrete diffusion methods because our evidence lower bound is theoretically tighter. There are two specific methodological elements that achieve this:
>
> 1. **Continuous Time ELBO:** Deriving a continuous time bound by taking $T \to \infty$.
> 2. **ELBO Simplification:** Setting certain terms in the ELBO analytically to 0, which is possible because of our choice of SUBS parameterization (7)
>
> Our ablation table shows that these elements can explain the gap relative to the previous state-of-the-art (SEDD), which uses the exact same backbone, optimizer, etc., and differs only in the training objective and parametrization.
>
> | Method | PPL (↓) |
> | --- | --- |
> | MDLM | 27.04 |
> | - w/o continuous time ELBO (1) | 27.19 |
> |   - w/o ELBO simplification & SUBS (2) | 28.56 |
> | SEDD (our implementation) | 28.90 |
>
> Next, we seek to understand why these elements improve perplexity. Recall that the diffusion ELBO can be written as follows.
>
> $\mathcal{L}_{ELBO} =$
>
> $\mathcal{L}_{recon}$
>
> $+ \mathcal{L}_{prior}$
>
> $+\mathcal{L}_{diffusion}$
>
> Our design elements improve each term.
>
> - $\mathcal{L}_{prior}$: The prior regularization term simplifies analytically to zero by design, since the noise schedule $\alpha(t)$ is set such that $\alpha(t=1) = 0$ (as in prior works [4, 5]).
> - $\mathcal{L}_{recon}$: The reconstruction loss term simplifies analytically to zero due to taking $T \rightarrow \infty$ and our “copy over unmasking” SUBS parameterization (i.e. we copy over unmasked tokens from $\mathbf{z}_t$ to $\mathbf{z}_s$).
>
> - $\mathcal{L}_{diffusion}$: This diffusion loss term simplifies to a weighted cross entropy (BERT-style) loss due to the SUBS parameterization and we make the ELBO as tight as possible by taking $T \rightarrow \infty$ (see also VDM [1] for a similar analysis of how the lower-bound becomes tighter as $T \rightarrow \infty$).
>
> In addition, our work shows that masked diffusion models like MDLM and D3PM work better than previously thought when paired with a training recipe. Specifically, they are competitive with score-based methods such as SEDD. We closed the gap using both the modeling contributions above, as well as a well-engineered, numerically stable implementation and effective training recipe, including a low-discrepancy sampler that reduces the variance of both perplexity bound evaluations and our gradient estimators for training.
>
> In our revised manuscript, we include a more detailed appendix that includes the above explanations and shows how our analysis and parameterization lead to the final simplified ELBO.
>
> ---
>
> ### ****Concern 2:**** Missing comparison to non-diffusion language models
>
> In addition to the extensive set of baselines that we provide in Tables 1 and 2, we have added a new experiment on the `text8` dataset, which allows us to compare against many non-diffusion results reported in the literature. As requested by the reviewer, this table contains non-diffusion baselines, including Flow-based models (IAF/SCF) [2] and a Bayesian Flow Network (BFN) [3].
>
> | Method | BPC($\downarrow$) | Train steps |
> | --- | --- | --- |
> | IAF/SCF | 1.88 | 1M |
> | AR Argmax Flow | 1.39 | 1M |
> | BFN | 1.41 | 1M |
> | D3PM | 1.45 | 1M |
> | SEDD | 1.39 | 1M |
> | MDLM | 1.44* | 0.4M* |
> | AR | 1.23 | 1M |
>
> * We provide BPC for an MDLM model that was only trained for **40%** of the total training steps, compared to all baselines. We hope to report the metrics for a fully trained model during the discussion period. Our partially trained MDLM is within 4% of the top-performing non-AR method, SEDD. Note also that our result is based on a single training run with no hyper-parameter tuning.
>
> The details of this `text8` experiment are as follows: following D3PM [4], we train a 77M param model on sequence chunks of 256 characters with batch size 512; baseline values were taken from SEDD [5].
>
> However, please note that **our existing experiments already compare to the strongest possible baselines**, including the best existing diffusion-based model (SEDD), and an AR baseline.
>
> ---
>
> ### ****Concern 3:**** Including additional non-perplexity-based metrics
>
> While perplexity is the dominant metric used in language model evaluation and has been extensively shown to correlate with downstream performance, we agree with the reviewer’s comment that other metrics that directly estimate downstream task accuracy are useful. We therefore evaluated our model on the Lambada [7] benchmark and present results below:
> |  | Lambada Accuracy ($\uparrow$) |
> | --- | --- |
> | AR | 49.62% |
> | MDLM | **53.10%** |
>
> We report accuracy of predicting the final word given a context of at least 50 tokens. MDLM improves over AR, aligning with our zero-shot results in Table 3.

---

> > ### Author Response · Authors · 2024-08-12
> > **Updated results on text8**
> >
> > On the text8 dataset, we managed to train the MDLM for only 800K steps due to resource constraints, unlike the baselines, which were trained for 1M steps. Despite this, we outperformed all non-autoregressive baselines, such as the flow-based models (IAF/SCF) [1] and a Bayesian Flow Network (BFN) [2], while matching the previous state-of-the-art, SEDD, in just **80%** of the training steps.
> >
> > | Method | BPC($\downarrow$) | Train steps |
> > | --- | --- | --- |
> > | IAF/SCF | 1.88 | 1M |
> > | AR Argmax Flow | 1.39 | 1M |
> > | BFN | 1.41 | 1M |
> > | D3PM | 1.45 | 1M |
> > | SEDD | 1.39 | 1M |
> > | MDLM | **1.39*** | **0.8M*** |
> > | AR | 1.23 | 1M |
> >
> > *We provide numbers for a partially trained MDLM model at 800K steps. All baseline models were trained for 1M steps. Due to resource constraints, we were unable to train our model to 1M steps during the rebuttal period.
> >
> > ---
> >
> > ****References:****
> >
> > [1] Ziegler, Zachary, and Alexander Rush. "Latent normalizing flows for discrete sequences." International Conference on Machine Learning. PMLR, 2019.
> >
> > [2] Graves, Alex, et al. "Bayesian flow networks." arXiv preprint arXiv:2308.07037 (2023).

---

> ### Author Response · Authors · 2024-08-07
> **Response to P1DW (2/3)**
>
> ### **Concern 4:**  It's unclear how well the approach scales to larger models
>
> We agree with the reviewer that finding scaling laws for discrete diffusion is a very interesting line of research, and one that we hope to conduct in follow-up work. Unfortunately, we cannot run such experiments in the limited rebuttal window and with a limited (academic) compute budget.
>
> ---
>
> ### **Concern 5:** It's unclear if MDLM can achieve practical speed-up comparing to AR models with same time budget.
>
> We highlight that MDLM can generate tokens in parallel and supports varying the number of generation steps, whereas AR models are limited by a fixed budget due to sequential generation. **MDLM achieves up to 1.8x speedup compared to AR** by varying the generation steps for sampling 64 batches of 256 tokens. **MDLM reaches 8% better generative perplexity under roughly the sampling speed as AR** (within 4%).
>
> More generally, our number of function evaluations (FE) to generate, say, a block of 128 tokens will be at most 128 steps. At the same time, because inference is typically memory bound, the wall clock time to run one (FE) is about the same for 1 block as it is for 128 blocks (and this is true up to a block size of, say, ~200 on an H100, after which we become compute bound again). Therefore, while an optimized implementation of MDLM is outside the scope of this paper, our diffusion approach holds the promise of significant speed improvements.
>
> ---
>
> ### Additional questions
>
> **### **Question 1:**** How do the authors explain their observation that time-step conditioning can be optional in MDLM?
>
> In absorbing state discrete diffusion the time step / noise level can be inferred from the number of masked tokens and hence explicit conditioning on time step embedding is not necessary. DiffusionBERT [8] also empirically found that for the absorbing state diffusion, time-conditioning the denoising model is ****not**** critical.
>
> **### **Question 2:**** How many steps are used when evaluating the perplexity of MDLM? How does this compare to other text diffusion models?
>
> There are different ways to interpret this question. First, sampling steps could refer to the value of $T$ used at training time. We report our best PPL values with the continuous-time formulation ($T \to \infty$) for MDLM and SEDD, and report results with finite $T$ among our ablations. A second interpretation is that “sampling steps” could refer to the number of steps to generate text from the model when we evaluate generative PPL. This number is reported on the x-axis of our generative PPL figure and varies between 32 and the length of the sequence.
>
> A third interpretation refers to the Monte Carlo sampling to evaluate PPL. When evaluating the perplexity bound of MDLM or SEDD, we use a single Monte Carlo sample per datapoint to approximate the integral of the continuous-time bound. Our low-discrepancy antithetic sampler allows us to estimate the perplexity bound with low variance, as shown in the table below. On OpenWebText, we find that we are able to accurately estimate the perplexity bound with just a single sample. In contrast, prior work required 1,000 samples to obtain an accurate estimate.
>
> Below we include a table that shows the limited effect of including more time steps in the Monte-Carlo estimate of MDLM on the OpenWebText dataset:
> | Num MC Samples | 1 | 2 | 5 | 10 | 100 |
> | --- | --- | --- | --- | --- | --- |
> | Perplexity ($\downarrow$) | 23.21 | 23.22 | 23.22 | 23.22 | 23.21 |
>
> **### **Question 3:**** Can the authors provide some comparative analysis and plots showing how MDLM compares to AR baselines across a range of training token amount budgets?
>
> We find that AR models are able to optimize the training loss better than diffusion models [see fig. 1 in the attached supplementary material](https://openreview.net/forum?id=L4uaAR4ArM&noteId=WereEXicWc) However, diffusion models are able to reach the same loss values with additional training. Crucially, the gap between the two curves is much smaller than previously thought. We are excited by the new opportunities that non-AR generation presents, such as methods for controlled generation, and the promise of more efficient sampling.

---

> ### Author Response · Authors · 2024-08-07
> **Response to P1DW (3/3)**
>
> **### **Question 4:**** What are the authors' thoughts on MDLM versus continuous-diffusion models for text, such as Plaid and CDCD, in terms of training, model performance, and inference efficiency?
>
> Several works using continuous diffusion for discrete data (e.g., Plaid [6], Diffusion LM [9], CDCD [10]) have not been able to approach AR perplexity. In our work, we demonstrate that discrete diffusion can successfully close this performance gap to AR models. Additionally, in our previous experience working with continuous diffusion models, such as DiffusionLM [9], we have found them to be less stable to train and require *ad hoc* techniques such as nearest embedding clipping. More recent attempts such as Plaid [6] and CDCD [10] seem promising but do not yield good PPL values.
>
> One of the potential benefits of discrete diffusion relative to AR sampling is more efficient sampling if the number of diffusion steps $T$ can be made less than the sequence length $L$, especially in batch size 1 regimes (e.g. on-device inference). We believe that discrete diffusion methods, such as ours, are more well suited to realize this gain relative to continuous diffusion for discrete data.
>
> **### **Question 5:**** Is it possible to speed up the generation process of MDLM, similar to the advanced ODE-solvers people are using in continuous-space diffusion models?
>
> This is indeed  a very promising research question, which we hope to explore in future work.
>
> **### **Question 6:**** How does MDLM compare to MaskGiT?
>
> While both MDLM and MaskGiT [11] use BERT-style losses to train generative models, the main differences are as follows:
>
> - **Objective:** In MDLM, we derive a tight and principled lower bound to the log-likelihood. MaskGiT’s objective, while effective, is not weighted appropriately to make it a proper ELBO.
>
> - **Sampler:** In MDLM, we use a principled ancestral sampling method, whereas in MaskGiT once again the sampling is done in a more heuristic manner that yields impressive results in the image generation domain. Additionally, MDLM can unmask any number of tokens at a given time step unlike MaskGiT where only a fixed number of tokens are unmasked at each iteration.
>
> ---
>
> #### **References:**
>
> [1] Kingma, Diederik, et al. "Variational diffusion models." Advances in neural information processing systems 34 (2021): 21696-21707.
>
> [2] Ziegler, Zachary, and Alexander Rush. "Latent normalizing flows for discrete sequences." International Conference on Machine Learning. PMLR, 2019.
>
> [3] Graves, Alex, et al. "Bayesian flow networks." arXiv preprint arXiv:2308.07037 (2023).
>
> [4]  Austin, Jacob, et al. "Structured denoising diffusion models in discrete state-spaces." Advances in Neural Information Processing Systems 34 (2021): 17981-17993.
>
> [5] Lou, Aaron, Chenlin Meng, and Stefano Ermon. "Discrete diffusion language modeling by estimating the ratios of the data distribution." arXiv preprint arXiv:2310.16834 (2023).
>
> [6] Gulrajani, Ishaan, and Tatsunori B. Hashimoto. "Likelihood-based diffusion language models." Advances in Neural Information Processing Systems 36 (2024).
>
> [7] Paperno, Denis, et al. "The LAMBADA dataset: Word prediction requiring a broad discourse context." arXiv preprint arXiv:1606.06031 (2016).
>
> [8] He, Zhengfu, et al. "Diffusionbert: Improving generative masked language models with diffusion models." arXiv preprint arXiv:2211.15029 (2022).
>
> [9] Li, Xiang, et al. "Diffusion-lm improves controllable text generation." Advances in Neural Information Processing Systems 35 (2022): 4328-4343.
>
> [10] Dieleman, Sander, et al. "Continuous diffusion for categorical data." arXiv preprint arXiv:2211.15089 (2022).
>
> [11] Chang, Huiwen, et al. "Maskgit: Masked generative image transformer." Proceedings of the IEEE/CVF Conference on Computer Vision and Pattern Recognition. 2022.

---

### Author Rebuttal · Authors · 2024-08-07

# General Response to Reviewers

Dear reviewers, we thank you all for the useful comments and feedback. In addition to the individual responses we provide directly to each of your comments, we wanted to highlight additional results and clarifications that are common to several of your reviews.

### **Improved theoretical understanding:**

The simplified objective we present is a product of a more careful derivation of the ELBO for absorbing state diffusion and of our proposed parameterization, that we call SUBS, which stands for **SUBS**titution. In our revised manuscript, we will include a more detailed appendix that shows how the two aspects of SUBS: (1) Zero Masking Probabilities and (2) Carry-Over Unmasking contribute to the simplified objective.

Below, we briefly recap the derivation of our simplified ELBO and how the SUBS parameterization and continuous-time analysis tighten this lower bound.

Recall the diffusion ELBO:
$\mathcal{L}_{ELBO} = \mathcal{L}\_{recon} + \mathcal{L}\_{diffusion} + \mathcal{L}\_{prior}$

- $\mathcal{L}_{prior}$: We set this prior regularization term analytically to zero by designing the noise schedule $\alpha(t)$ such that $\alpha(t=1) = 0$.
- $\mathcal{L}_{recon}$: We set this reconstruction loss term analytically to zero by taking $T \rightarrow \infty$ and by our “copy over unmasking” parameterization (i.e. we copy over unmasked tokens from $\mathbf{z}_t$ to $\mathbf{z}_s$ with $s < t$.
- $\mathcal{L}_{diffusion}$: This diffusion loss term simplifies to a weighted cross entropy (BERT-style) loss due to the SUBS parameterization and we make the ELBO as tight as possible by taking $T \rightarrow \infty$ (see also VDM [1] for a similar analysis of how the lower-bound becomes tighter as $T \rightarrow \infty$).

### **Algorithms:**

Below we include algorithms for training and inference of MDLM, which we plan to include in our revised manuscript:

**Algorithm: MDLM Training:** Inputs: dataset $\mathcal{D}$, monotonic masking schedule $\alpha_t : [0,1] \to [0,1]$, BERT-like model $x_\theta$, $\mathbf{m}$ denotes the mask vector, $\mathbf{x}^{1:L}$ denotes a sentence with $L$ tokens, $\mathbf{z}^{1:L}_t$ denotes the latent vector with $L$ tokens.

1. **repeat**
    1. $\mathbf{x}^{1:L} \sim \mathcal{D}$       *// Sample a sequence of length L from dataset (can be a batch)*
    2.  $\mathbf{z}^\ell_t \sim \text{cat}(\mathbf{z}^\ell_t; \alpha_t \mathbf{x}^\ell + (1 - \alpha_t)\mathbf{m})$   $\forall 1 \leq \ell \leq L$ for random $t \sim \mathcal{U}[0, 1]$  *// Mask each token $\mathbf{x}^\ell$ independently with masking rate $\alpha_t$ to obtain the latent $\mathbf{z}^{1:L}_t$.*
    3. Update weights $\theta$ of denoising (BERT) model $\mathbf{x}_\theta$ gradient descent step on

&nbsp; &nbsp;&nbsp;&nbsp;&nbsp;&nbsp;&nbsp;&nbsp;$\nabla_\theta \frac{\alpha'_t}{1 - \alpha_t} \sum\_{\ell}\log \langle \mathbf{x}\_\theta^\ell(\mathbf{z}^{1:L}_t),\mathbf{x}^\ell \rangle$ // Note: this is simply a “weighted” BERT-style loss

**until** converged


Note that the only differences to BERT are that in standard BERT

1. In step `1.2.`, $\alpha_t$ is a constant (set to 0.85 implying a fixed masking rate of 15%).
2. In step `1.3.` there is no weighting term in front of the cross-entropy loss.

Otherwise the training algorithms are equivalent!

****Algorithm: MDLM Sampling / Inference****

1. $\mathbf{z}^{1:L}_1 =$ [MASK, MASK, …, MASK]    *// Sampling process starts with all MASK tokens.*
2. **for** t = {$1, \frac{T-1}{T}, \dots, \frac{1}{T}$} **do**
    1. $s \xleftarrow{} t - \frac{1}{T}$
    2. $\mathbf{z}^\ell_{s} \sim \text{Cat}(\mathbf{z}^\ell_s; \frac{(1 - \alpha_s)\mathbf{m} + (\alpha_s - \alpha_t)\mathbf{x}_\theta(\mathbf{z}_t)}{1 - \alpha_t} )$   $\forall 1 \leq \ell \leq L$.
    3. $\mathbf{z}_s^\ell = \mathbf{z}_t^\ell$  if $\mathbf{z}_t^\ell \neq \text{[MASK]}$ $\forall 1 \leq \ell \leq L$  *// To prevent unmasked token being remasked in the reverse process; see Eqn(7) in the paper.*
    4. $\mathbf{z}^{1:L}_s = \mathbf{z}^{1:L}_t$

    end **for**


### **Additional Experiments:**

**Text8:**

To better compare to other non-diffusion baselines (see reviewer p1DW’s concerns), in addition to the extensive set of baselines that we provide in Tables 1 and 2, we have added a new experiment on the `text8` dataset, which we provide below:

| Method         | BPC($\downarrow$) |
|----------------|-------------------|
| IAF/SCF        | 1.88    |
| AR Argmax Flow | 1.39 |
| BFN            | 1.41 |
| D3PM           | 1.45 |
| SEDD           | 1.39  |
| MDLM           | 1.44* |
| AR             | **1.23** |

*We provide the numbers are for a partially trained MDLM model on 400K steps and hope to report the numbers for a fully trained model during the discussion period. All the baselines models were trained for 1M steps.
(Details of this `text8` experiment are as follows: following D3PM [4], we train a 77M param model on sequence chunks of 256 characters with batch size 512; baseline values were taken from SEDD [5]).

**Comparing MDLM vs AR sample efficiency**

In addition to the sampling efficiency analyses in Figure 2, we compare MDLM and AR in the batch size 1 setting where MDLM achieves optimal sampling efficiency by caching the outputs of the denoising model (Suppl. D.2). Whereas AR models are limited to a fixed number of generation steps, MDLM may flexibly trade sample efficiency for quality by varying the number of diffusion steps $T$. In Figure 3 (Supplementary material submitted in the general response)  MDLM achieves **up to 1.8x faster** **sampling** and **53% better generative perplexity** compared to AR when generating 256 tokens.

---

**References:**

[1] Kingma et al. "Variational diffusion models.".
[4]  Austin, Jacob, et al. "Structured denoising diffusion models in discrete state-spaces.".
[5] Lou, Aaron, et al. "Discrete diffusion language modeling by estimating the ratios of the data distribution."

---

> ### Comment · Reviewer_wXn1 · 2024-08-07
> **where is  individual response?**
>
> "In addition to the individual responses we provide directly to each of your comments"

---

> > ### Author Response · Authors · 2024-08-08
> > **Individual responses are now posted**
> >
> > We apologize for the delay. All individual responses have now been posted as “Official comments”. Thank you again for your time and consideration.

---

> ### Author Response · Authors · 2024-08-12
> **Updated results on text8 and comparison to non-AR baselines**
>
> On the text8 dataset, we managed to train our MDLM model for only 800K steps due to resource constraints, unlike the baselines, which were trained for 1M steps. Despite this, we **outperformed all non-autoregressive baselines**, such as the flow-based models (IAF/SCF) [1] and a Bayesian Flow Network (BFN) [2], while matching the previous state-of-the-art, SEDD, in just **80%** of the training steps.
>
> | Method | BPC($\downarrow$) | Train steps |
> | --- | --- | --- |
> | IAF/SCF | 1.88 | 1M |
> | AR Argmax Flow | 1.39 | 1M |
> | BFN | 1.41 | 1M |
> | D3PM | 1.45 | 1M |
> | SEDD | 1.39 | 1M |
> | MDLM | **1.39*** | **0.8M*** |
> | AR | 1.23 | 1M |
>
> *We provide numbers for a partially trained MDLM model at 800K steps. All baseline models were trained for 1M steps. Due to resource constraints, we were unable to train our model to 1M steps during the rebuttal period.
>
> ---
>
> ****References:****
>
> [1] Ziegler, Zachary, and Alexander Rush. "Latent normalizing flows for discrete sequences." International Conference on Machine Learning. PMLR, 2019.
>
> [2] Graves, Alex, et al. "Bayesian flow networks." arXiv preprint arXiv:2308.07037 (2023).

---

### Author Response · Authors · 2024-08-09
**Please refresh the page if the equations are not rendering**

Please **refresh the page** if the equations are not rendering.

---

### Comment · Area_Chair_ccaN · 2024-08-13

Dear Reviewers, Authors,

The discussion period will be ending soon. If you have any additional questions (reviewers) or any additional clarifications (authors) please post them soon.

Dear authors, do you think you can address Reviewer wXn1's concerns regarding the discussion on related works ?

---

> ### Author Response · Authors · 2024-08-13
> **Response to Area Chair ccaN**
>
> Dear AC, Thank you for following up on our discussions. We were actively working on Reviewer wXn1's latest comment and just posted our response.
>
> To the reviewers, please let us know if there are any remaining open questions or concerns. Thank you again for you time and feedback.

---

### Decision · Program_Chairs · 2024-09-25

**Decision:**

Accept (poster)

**Comment:**

The paper proposes a model called Masked Diffusion Language modeling that is a special case of D3M. The authors propose several new contributions that lead of really strong results, bringing diffusion models close to AR models for sequence modeling. Specifically, they propose a specific parameterization of the predictive distribution (which they call SUBS), that allow for better lower bounds in the ELBO objective, because the objective ends up having fewer terms (eqn 8) compared to what a more vanilla formulation from D3M would have had (eon 9). In addition the authors lay out other contributions, related to better implementation, vocabulary choice, ignoring of the time steps in denoising, etc that they claim lead to better results that would have been expected from prior work.

Results are shown on language modeling and human genome sequence modeling. The reviewers were generally positive -- although there was some push back against whether or not the method was indeed "simple", and also others asked for more detailed comparisons to other methods across different scales of the data, and of the model.  I hope the authors will take these points into consideration when submitting their final version, for maximal impact.

While I was perusing the paper, I also happened to find some typos in equations (unless I misunderstood some of the details of the paper) that I am listing here -- if they are indeed correct, I would hope that the authors will make the corrections for better readability:

1. Line 86 might be incorrect. I think it should be: Cat(z_t; \alpha_{t|s} z_s + (1-\alpha_{t|s}) \pi)
2. Line 98 mixes probabilities, and probabilistic variables, making it inconsistent with the rest of their notations. It should be:  q(z_t|x_0) = Cat(z_t; \alpha_t x_0 + (1-\alpha_t) m);
3. Equation on line 103 uses s, t with latent variables, inside the chain rule. Instead of p_\theta(z_s|z_t) it should be p_\theta(z_{i-1} | z_i).
4. Line 181 — the range of time might be wrong. I think it should have been \tilde{x}^{L’:L} , not \tilde{x}^{L':L-L'} ? And the same for  lines 184, z_1 and \tilde{x} have errors in the sequence limits: L’:L-L’ should be L’:L again, possibly.
5. Equation 19 -- z_s, z_t should be x_s, x_t, to be consistent with p_\theta(x_s|x_t). The variable names are being switched around ?